# Tunable hydrogel-based micropillar arrays for myelination studies

Soufian Lasli[1], Claire Vinel[1], Ayushi Agrawal[1], Yousef Javanmardi [1], Paola Pedarzani [2], Beatriz Garcia Diaz [3], Juan Antonio Garcia-Leon[4,5], Boris Djordjevic[1], Ian J. White[6], Graham K. Sheridan [7]✉, William D. Richardson [8]✉ & Emad Moeendarbary [1,9]✉

Oligodendrocytes enable rapid central nervous system signaling by myelinating axons. Here, to model key biomechanical cues regulating myelination, we developed a tunable hydrogel-based micropillar array system that mimics the three-dimensional architecture and softness of axons. This platform supports the long-term culture of oligodendrocytes and robust formation of multilayered compact myelin by rodent and human oligodendrocytes. Using confocal and transmission electron microscopy, we observed a strong linear correlation between immunostained myelin thickness and the number of myelin wraps, enabling high-content quantification of myelination. Systematic variation of pillar stiffness, diameter and surface chemistry within pathophysiological ranges revealed that both mechanical and geometric properties of axon-like substrates critically regulate oligodendrocyte differentiation and myelin wrapping. Importantly, we demonstrate that pharmacological agents exhibit stiffness-dependent effects on myelination, suggesting that overly rigid in vitro models may yield false-positive drug hits. This platform offers a physiologically relevant, high-throughput assay for dissecting oligodendrocyte biology and discovering remyelinating therapies for diseases such as multiple sclerosis.

Demyelinating diseases of the central nervous system (CNS) are characterized by damage or loss of the myelin sheath formed by oligodendrocytes (OLs), disrupting neuronal signaling[1] and leading to sensory, motor and cognitive dysfunctions[2]. While OLs and their precursor cells (OPCs) retain the capacity to regenerate myelin in the adult brain[3], our understanding of the molecular and physical factors regulating CNS myelination remains limited[4]. OL development and myelination are tightly regulated in a spatially and temporally controlled manner by intrinsic and extrinsic cues, such as growth factors, kinases and extracellular matrix (ECM) molecules that modulate gene expression and cytoskeletal dynamics[5]. Mechanobiological cues from neurons and the ECM are key regulators of myelination, as OL mechanosensitivity to stiffness influences process extension and wrapping[6,7]. Importantly, OLs can also produce compact myelin on paraformaldehyde (PFA)-fixed axons, suggesting that myelination can occur independent of axonal signals when the mechanical environment is permissive[8]. In parallel,

[1]Department of Mechanical Engineering, University College London, London, UK. [2]Research Department of Neuroscience, Physiology and Pharmacology, University College London, London, UK. [3]Unidad de Gestión Clínica de Neurociencias, IBIMA, Hospital Regional Universitario de Málaga, Malaga, Spain. [4]Departamento de Biologia Celular, Genetica y Fisiologia, Instituto de Investigación Biomédica de Málaga y Plataforma en Nanomedicina-IBIMA Plataforma BIONAND, Facultad de Ciencias, Universidad de Malaga, Malaga, Spain. [5]CIBER de Enfermedades Neurodegenerativas, Instituto de Salud Carlos III, Madrid, Spain. [6]Laboratory for Molecular Cell Biology, University College London, London, UK. [7]School of Life Sciences, University of Nottingham, Nottingham, UK. [8]Wolfson Institute for Biomedical Research, University College London, London, UK. [9]BioRecode Ltd., London, UK. ✉e-mail: graham.sheridan@nottingham.ac.uk; w.richardson@ucl.ac.uk; e.moeendarbary@ucl.ac.uk

neuronal activity can direct myelination by modulating OL selection of axons[9]. Understanding these diverse inputs is essential for uncovering how myelin forms, is maintained and is damaged in conditions such as multiple sclerosis and Alzheimer's disease[10].

To dissect these complex mechanisms, in vitro models offer controlled, reductionist systems that decouple extrinsic signals from intrinsic OL behavior[11]. While more advanced platforms have replaced traditional flat substrates, they often fall short in replicating the full biophysical, biomechanical and biochemical complexity of the CNS microenvironment. Although OLs are mechanosensitive, with ECM stiffness and axonal properties regulating OL differentiation and the concentric wrapping of myelin, most previous platforms fail to mimic the softness and topography of the CNS (Extended Data Table 1).

Early pioneering work investigating the role of biophysical cues in promoting myelination[12] includes a landmark study by Lee et al.[13], who used electrospun polystyrene nanofibers to examine how axon diameter influences CNS myelination. Their aim was to decouple geometric properties (that is, diameter) from axonal membrane signals. Such systems have been instrumental in demonstrating that OLs can discriminate between axons based on their caliber[13–15], preferentially myelinating fibers larger than 0.5 μm and rarely those below 0.4 μm (ref. 13). While these nanofiber-based models offer precise control over geometry, a key limitation is their inability to recapitulate the soft mechanical properties of neuronal tissue. Commonly used artificial fibers (such as glass, polystyrene, poly-L-lactic acid or polycaprolactone) have stiffnesses above 1 GPa (Extended Data Table 1), in contrast to the ~5 kPa elasticity of native axons, as measured by atomic force microscopy (AFM)[16]. This mechanical mismatch may lead to nonphysiological OL responses and compromises the relevance of these systems for mechanobiology[17].

Although more recent studies have introduced artificial axon platforms with improved mechanical features[14,18,19], they typically lack simultaneous control over geometry (diameter and spacing) and stiffness within physiologically relevant ranges (Extended Data Table 1). Indeed, precise control of interaxonal distance is particularly important, as individual OLs extend multiple processes to wrap several axons in parallel, and spatial arrangement influences both process targeting and myelin architecture[20]. These platforms often rely on stereolithography-based fabrication techniques that require specialized equipment and technical expertise (Extended Data Table 1). Importantly, most do not demonstrate compact multilayered myelin or fail to support human OL culture, limiting their physiological relevance and translational utility (Extended Data Table 1).

Here, we present a fully hydrogel-based micropillar platform that addresses these limitations by enabling independent and tunable control of axon-mimetic diameter, spacing and stiffness within a physiologically relevant range. Using polyacrylamide, a gold-standard hydrogel in mechanobiology that has been commonly used in two-dimensional (2D) OL differentiation and myelination studies[6,7,21,22], we engineered vertical micropillars that mimic axonal properties and support OL attachment, differentiation and multilayered myelin formation. Our approach offers substantial advantages over traditional electrospun fiber scaffolds in terms of fabrication simplicity, imaging accessibility, scalability and biological relevance to human systems. We demonstrated that OLs from both primary rat and human sources form concentric, multilayered myelin around our hydrogel-based pillars, recapitulating key structural features of compact CNS myelin. We further combined fluorescence microscopy with transcriptomic profiling to identify molecular pathways involved in OL differentiation and myelination within this controlled three-dimensional (3D) environment. The ability to fine-tune both biophysical and biochemical cues within a soft, brain-like matrix makes this model highly relevant to human biology and disease. Altogether, our system offers a cost-effective and physiologically relevant tool for dissecting the mechanisms of CNS myelination and accelerating drug discovery in demyelinating disorders.

## Results

### Design and fabrication of hydrogel-based micropillar arrays
Leveraging the advantages of polyacrylamide hydrogels, we have developed biomechanically optimized micropillar arrays for CNS myelination assays. The vertical freestanding cylindrical micropillars mimic neuronal axons around which OLs can wrap myelin. Using photolithography, a conventional microfabrication technique, ultraviolet (UV) light was used to transfer desired micropatterns (array of circles) onto a silicon wafer (Fig. 1a). The diameter ($D$) and interpillar distance ($d$) of circles can be adjusted with submicrometric resolution. After photolithography, the patterned circles were precisely and smoothly etched at a specific depth to produce a reusable silicon wafer master mold (Fig. 1b). By placing polydimethylsiloxane (PDMS) spacers around the micropatterned areas, a polyacrylamide solution (containing fluorescein isothiocyanate (FITC)-labeled molecules) is poured, degassed under vacuum and allowed to polymerize on square thin coverslips (Fig. 1c,d). After peeling off the coverslips from the wafer, the resulting hydrogel micropillar arrays can be visualized using either scanning electron microscopy (SEM; dried condition, Fig. 1e) or a fluorescent upright confocal microscope (hydrated, Fig. 1f). Hydrogel swelling, when immersed in aqueous solutions, needs to be taken into consideration when designing the arrays to reach the desired dimensions. We noticed a 26% ± 4% (mean ± s.d.) increase in the micropillar diameter between the mold and the actual micropillar sizes when immersed in phosphate-buffered saline (PBS) (Fig. 1g and Extended Data Fig. 1). For cell culture, custom-made PDMS wells were glued closely around the pillars (Fig. 1h,i), which enables the use of a small cell number and low media volume. By simply adjusting the concentration of the monomer (acrylamide) and the crosslinking agent (bis-acrylamide), we fabricated ultrasoft (0.52 ± 0.02 kPa), soft (5.1 ± 0.1 kPa), intermediate (20.2 ± 1.9 kPa) and stiff (54.9 ± 2.8 kPa) polyacrylamide micropillars (Fig. 1j). Because OLs are mechanosensitive[23], it is important to be able to replicate the biomechanical properties of brain tissue (0.1–1 kPa)[24] or individual neuronal axons (5 kPa)[16]. Hydrogels with stiffnesses of 20 and 50 kPa were included to represent supraphysiological conditions, enabling the assessment of OL responses to substrates overly stiffer than native brain tissue. This allowed comparisons across a broader mechanical spectrum to better understand stiffness-dependent cellular behavior.

### OLs wrap and produce compact myelin around hydrogel micropillars
Next, we cultured O4+ OL lineage cells (their identity as premyelinating OLs was confirmed; Extended Data Fig. 2a–c) from rat cerebral cortex for up to 2 weeks. We divided the OL culture into three distinct phases for analysis: (1) proliferation at day 2, (2) differentiation at day 7, and (3) myelination at day 14 (Fig. 2a). By day 7, the resulting cell population displayed a gene expression profile characteristic of myelinating OLs, with minimal expression of markers associated with other brain cell types (Extended Data Fig. 2d). Compared with electrospun fibers, one of the distinct advantages of micropillar arrays is the ease and speed with which myelinated structures can be imaged and quantified. Confocal microscopy (25× water-dipping objective, numerical aperture (NA) >0.95) z-stacks spanning the hydrogel base and micropillars were acquired, and maximum-intensity projection images were generated either across the full height (to capture cell bodies and count nuclei) or excluding the base (to quantify myelin). Z-stacks with a step size of 2 μm were used to acquire large field-of-view images (Fig. 2b,c), while the z-step size was reduced to 0.5 μm (Fig. 2d) in a subset of pillars (given the longer imaging acquisition time required) to visualize the fine details of myelin around the pillars and its 3D reconstruction. Two key parameters were used to assess OL wrapping efficiency: the wrapping score (Fig. 2e) and the number of fully wrapped pillars per cell (Fig. 2f). The wrapping score (0–3) reflects the percentage of myelin coverage around the pillars, with a score of 0 indicating no wrapping. By

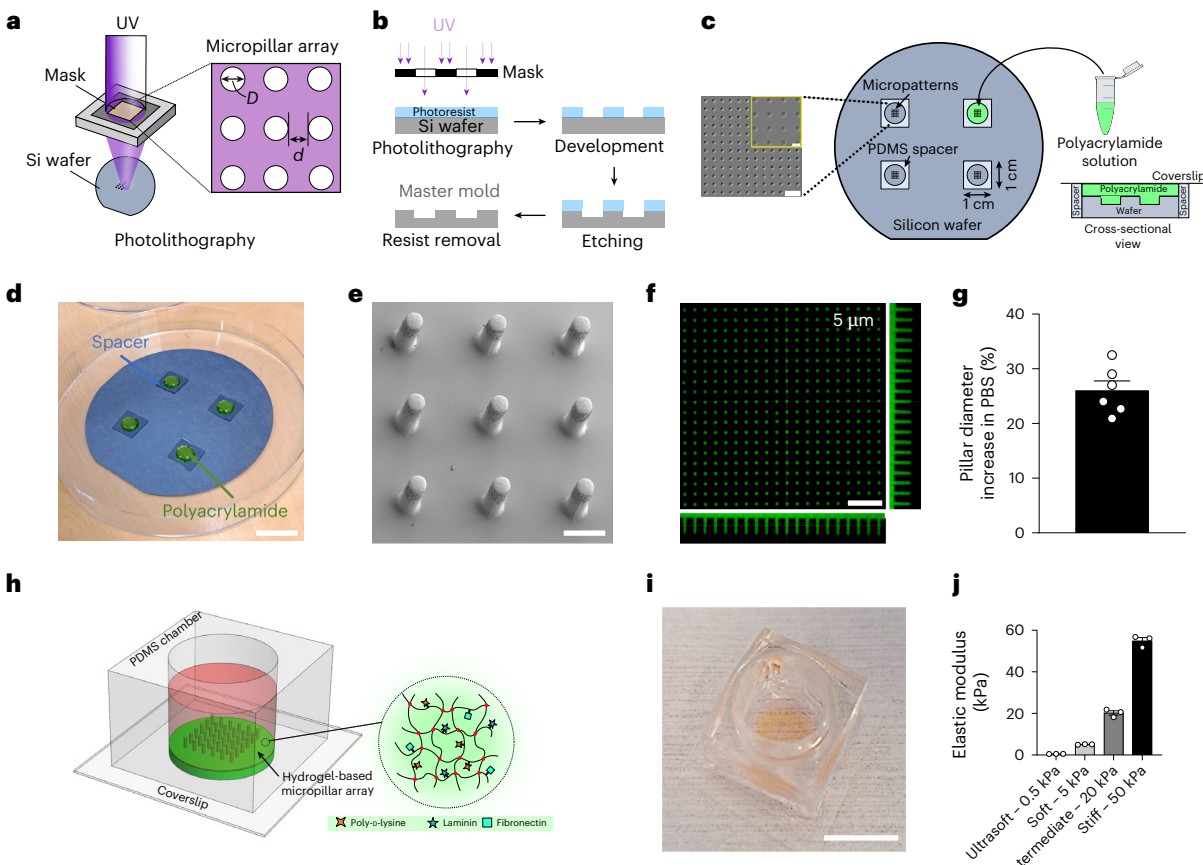

**Fig. 1 | Fabrication of hydrogel-based micropillar arrays for myelination assays. a**, Micropatterns are transferred onto a silicon wafer using microfabrication techniques in a cleanroom. **b**, Patterns are transferred on a photoresist-coated wafer using UV light, developed and etched to produce a master mold for replica molding. **c**, Outside the cleanroom, PDMS spacers are placed around the patterned area, and a polyacrylamide solution is poured, degassed under vacuum and mixed with TEMED to initiate polymerization. A coverslip is placed on top to seal the system, which is then peeled off the mold with the gel after polymerization. **d**, Photograph of the silicon wafer with spacers and polyacrylamide solution. Scale bar, 3 cm. **e**, SEM image of 18-µm-diameter pillars with an interpillar distance of 15 µm and a height of 100 µm

(when immersed in liquid). Scale bar, 20 µm. **f**, Confocal image of hydrated FITC-labeled pillars (5 µm diameter, 10 µm spacing, 24 µm height). Scale bar, 50 µm. **g**, Swelling of micropillars in PBS compared with mold dimensions; data are mean ± s.d. from $n = 6$ molds with different geometries. **h**, Schematic of the final platform for cell culture, with a PDMS chamber enclosing the gel, allowing coating with proteins such as PDL, laminin or fibronectin. **i**, Photograph of the assembled micropillar platform. Scale bar, 1 cm. **j**, Stiffness of micropillars measured by AFM: 0.5 kPa (ultrasoft), 5 kPa (soft), 20 kPa (intermediate) and 50 kPa (stiff); data are mean ± s.d., $n = 3$ gels with 9 measurements per gel. Schematics in **a**–**c** and **h** created in Inkscape.

contrast, a score of 3 indicates a fully wrapped pillar, as indicated with a full ring (Fig. 2e). Unlike Schwann cells, OLs can myelinate multiple axons simultaneously[20]. Accordingly, we quantified the average number of fully wrapped pillars per OL by dividing the number of pillars scoring 3 by the total number of cells (Fig. 2f). Unless otherwise specified, we conducted all experiments using poly-D-lysine (PDL)-coated pillars.

Next, we showed the compatibility of our platform with electron microscopy (EM) techniques (Fig. 2g–i). The periodic nanostructure of myelin, consisting of concentric multilayers surrounding axons, is critical for rapid signal conduction[25]. As a result, it is essential to have imaging techniques that can effectively visualize and characterize this multilayered structure. SEM showed a clear 3D overview of OL-mediated wrapping of pillars (Fig. 2g). The vertical alignment of micropillars enables the efficient acquisition of high numbers of horizontal cross-sections of myelin wraps for transmission electron microscopy (TEM) analysis in contrast to electrospun fibers, where alignment inconsistencies and fiber entanglement complicate characterization. Although the pillars may shrink due to hydrogel dehydration during sample preparation, we observed clear and compact multilayered myelin around our pillars with a nanostructure similar to that found in vivo (Fig. 2h,i). Our analysis showed that over 50% of pillars displayed

multilayered myelin (52 ± 9%; mean ± s.e.m.) with an average thickness of 46 ± 3 nm and average number of myelin sheath of 3.6 ± 0.2 (Fig. 2j). The degree of myelin compaction was consistent across different pillars indicated by the high correlation (average $R^2 = 0.94$) between the number of myelin sheaths and the thickness of myelin (Fig. 2k).

## Effects of topography and geometrical features

To assess the effect of micropillar topography, primary rat OPCs were cultured and differentiated over 10 days on either flat polyacrylamide hydrogels or micropillar arrays (5 µm diameter and 10 µm spacing), followed by transcriptomic analysis of mature OLs. OLs grown on micropillars exhibited over 800 differentially expressed genes (DEGs) (DESeq2, Wald test, adjusted $P < 0.05$, Benjamini–Hochberg correction), with 462 upregulated and 414 downregulated compared with flat conditions (Extended Data Fig. 3a–c).

Gene set enrichment analysis of all upregulated genes (Fig. 3a) confirmed enrichment of signaling pathways related to ECM remodeling (ECM, collagen-containing ECM and cell junction organization) and glial lineage progression (neurogenesis and cell differentiation). Conversely, gene set enrichment analysis of downregulated genes (Fig. 3a) showed enrichment for pathways associated with cell division

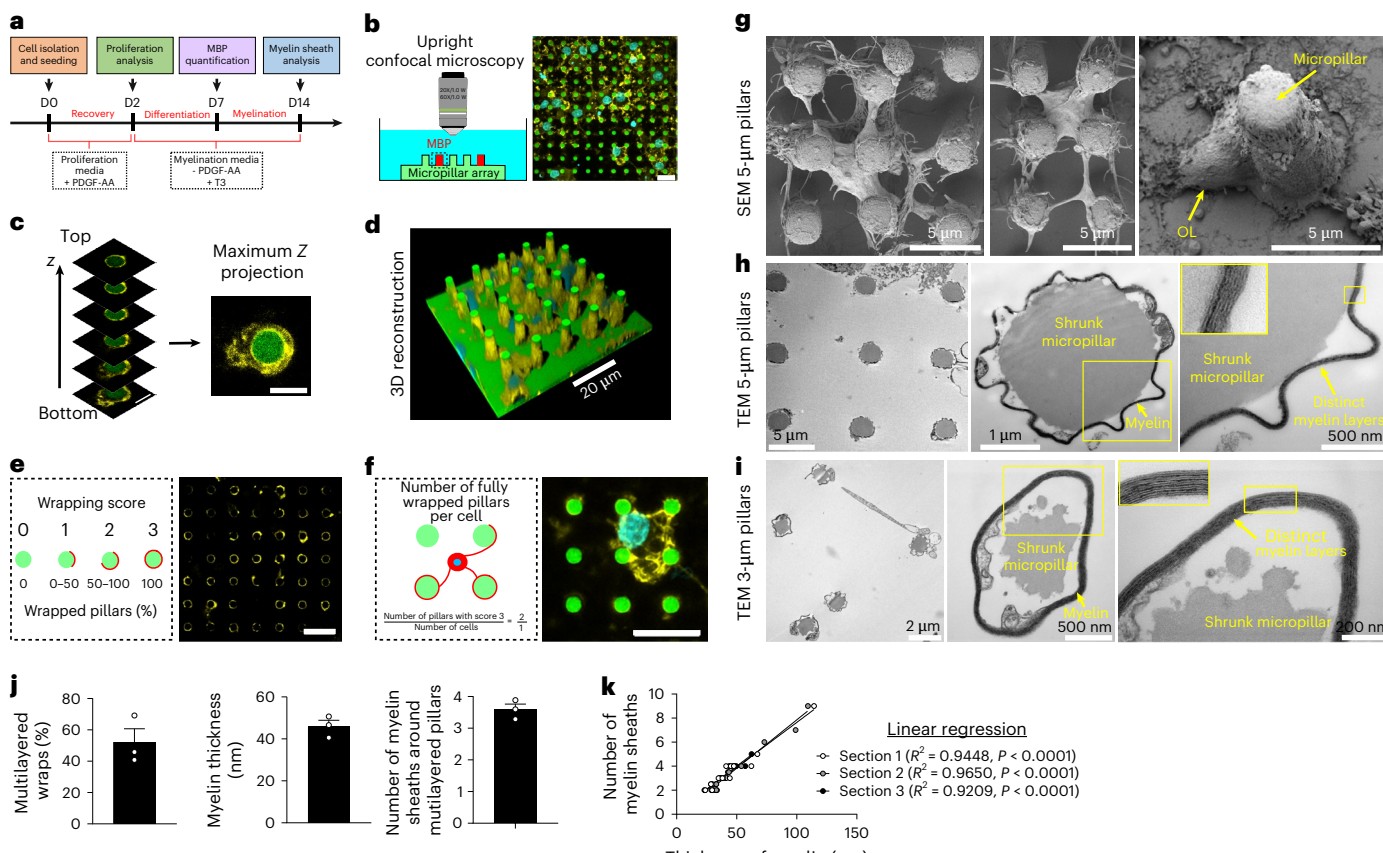

**Fig. 2 | Experimental workflow and validation of multilayered myelin formation. a**, Timeline for rat OL culture. D, day. **b**, Immunostained samples were imaged using an upright confocal microscope (25× water-dipping objective, NA >0.95) for large field-of-view acquisition with good resolution. **c**, Z-stacks (2 μm step size) spanning pillar height (excluding the gel base) were acquired and processed as maximum-intensity projections. Scale bar, 5 μm. **d**, High-resolution 3D reconstruction of a 5 × 5 micropillar field using a 0.5-μm z-step for fine analysis of myelin deposition. **e**, Wrapping score quantifies the degree of myelin coverage per pillar (0–3 scale). **f**, The number of fully wrapped pillars (score 3) per cell was calculated by dividing the total number of fully wrapped pillars by the number of OLs in the field. **g–i**, Representative images of three independent experiments:

SEM image showing OL wrapping on 5-μm pillars after 7 days of culture (scale bar, 5 μm) (**g**); TEM images of compact myelin sheaths around 5-μm (**h**) and 3-μm (**i**) pillars after 14 days of culture. Note that hydrogel shrinkage occurs due to dehydration during sample preparation, but myelin nanostructure is preserved. **j**, Percentage of pillars with multilayer myelin per field (left), myelin thickness (middle) and number of sheaths per pillar (right) based on TEM imaging. **k**, Strong correlation (average $R^2$ = 0.94) between myelin thickness and number of myelin layers. Statistical significance was assessed by Pearson correlation. Data are mean ± s.e.m., quantified from 3 ultrathin sections, each containing 120 pillars, and 13–16 high-resolution pillars quantified per section. Schematics in **a–c**, **e**, and **f** created in Inkscape.

(kinetochore, mitotic spindle and cell cycle) and high cellular dynamism (ATP binding/hydrolysis activity and microtubule/cytoskeletal motor activity). Analysis of the top 30 DEGs (Extended Data Fig. 3d) revealed that most were significantly upregulated in OLs cultured on micropillar hydrogels (Wald test, adjusted $P < 0.05$)). These genes were associated with key biological processes, including cellular metabolism (*Pfkl*, *Ckm* (also known as *CPK-M*), *Cox4i2*, *Aldh1l2* and *Mgarp*), ECM remodeling (*Bcan* and *Fbln2*) and neural development and differentiation (*Wnt4*, *Vegfa* and *Nkx6-2*). Gene Ontology (GO) analysis further indicated enrichment for cell division and drug response pathways (Extended Data Fig. 3e). Key genes associated with OL maturation and myelination were also upregulated, including *Mog*, *Mag*, *Mbp* and *Gpr62*. Additional genes involved in cytoskeletal and lipid remodeling, critical for myelin sheath formation, such as *Ank3*, *Stbn1/4*, *Tubb3*, *Arpc2* and *Fa2h*, were also elevated. This included increased expression of *Nfas* and decreased expression of *Lamb1* and *Lama*, which have previously been associated with OL maturation (Fig. 3b). Transcriptomic clustering further revealed that OLs cultured on micropillars aligned with human mature OL profiles, whereas those on flat hydrogels resembled OPCs and immature OLs (Extended Data Fig. 3f).

Having demonstrated that micropillars enhance OL differentiation and maturation, we next investigated the effect of pillar diameter

and interpillar distance (Fig. 3c–f). Previous research has shown that the distribution of axon diameters in human cortical white matter ranges from 0.1 to 10 μm, with most calibers being less than 1 μm (ref. 26). Fibers with a diameter of less than 0.2 μm do not get myelinated in vivo, whereas the threshold is 0.4 μm in vitro. Axons with diameters ranging from 0.2 to 0.8 μm can be found myelinated or unmyelinated. However, thick axons with diameters larger than 1 μm are preferentially myelinated in vivo and are more often fully myelinated in vitro[27,28]. Taking these ranges into account, we tested three different micropillar diameters (3, 5 and 10 μm) while keeping the pillar height constant (~23–24 μm) (Fig. 3c–f). Furthermore, interpillar distance was varied (three different interpillar distances of 5, 10 and 15 μm) to mimic different axonal spacing within neural tissue and to investigate how pillar density influences OL process extension and myelination dynamics. For ease of comparison between all conditions, we divided the micropillar platform into smaller 4 × 4 arrays. Each small array (repeated four times) included micropillars with one of the three different interpillar distances and a flat region (without pillars) of polyacrylamide substrate (Fig. 3c). The number of cell bodies per field (250 × 250 μm²) was determined by counting Hoechst-positive nuclei at day 2 of the culture (Fig. 3d). We observed significantly (Supplementary Table 1) higher cell densities in micropillar regions

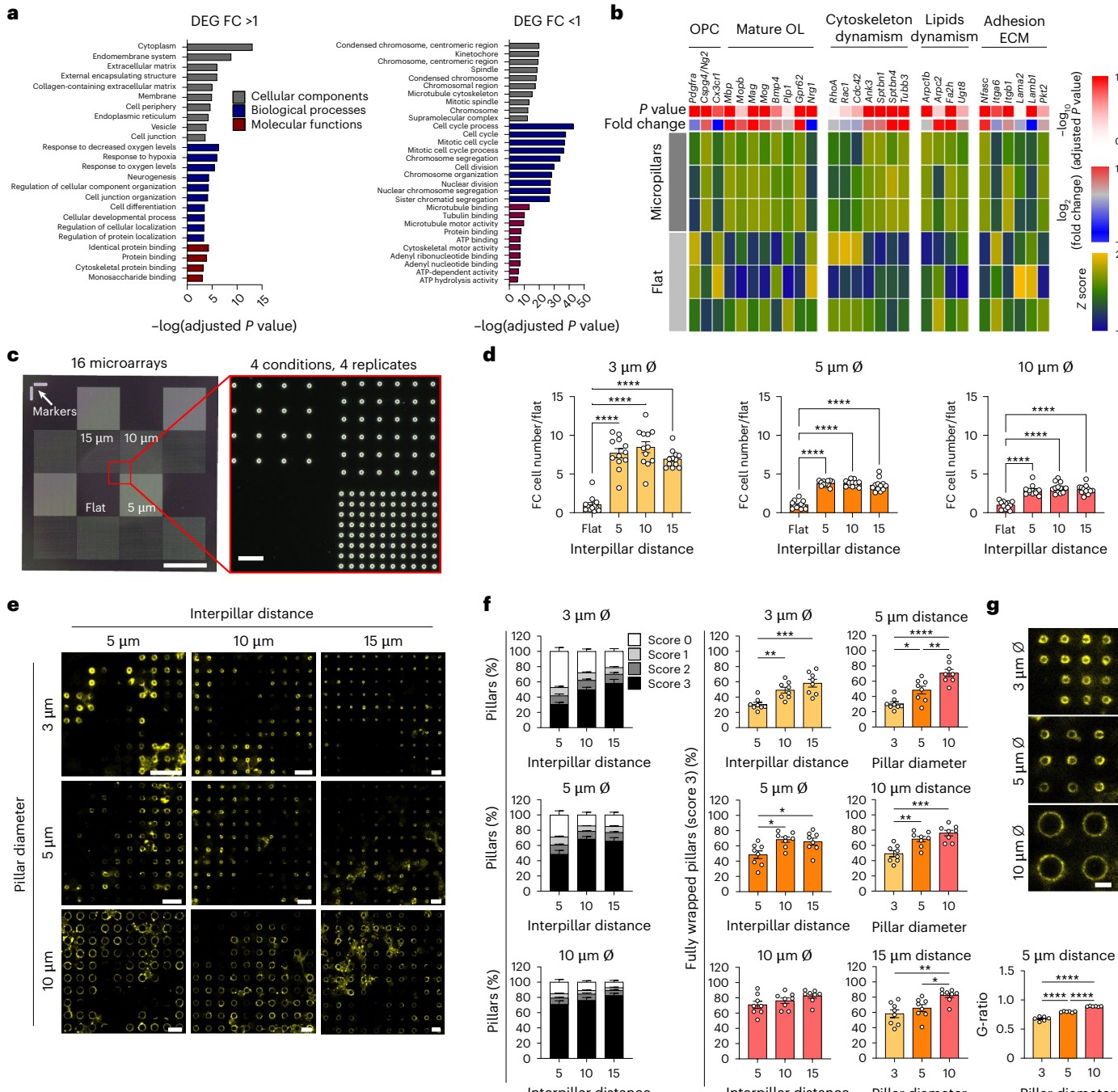

**Fig. 3 | Impact of geometrical features on OL behavior. a**, RNA sequencing analysis of rat OLs cultured on 50-kPa micropillars (5 µm diameter, 10 µm spacing) versus flat hydrogels identified significantly enriched GO terms among upregulated (fold change (FC) >1, left) and downregulated (FC <1, right) DEGs. **b**, Heatmap showing the differential expression (FC) and significance (*P* value) of genes specifically expressed in OPCs and mature OLs, particularly those involved in cytoskeleton and lipid dynamics, as well as ECM adhesion proteins, in OL cultures on micropillars compared with flat hydrogels. Statistical significance was assessed using the Wald test, two-sided, and the resulting *P* values were adjusted for multiple testing using the Benjamini–Hochberg procedure (**a**,**b**). **c**, Micropillar master mold with optimized 16-subarray layout enabling systematic variation of diameter (3, 5 and 10 µm) and spacing (flat, 5, 10 and 15 µm). Scale bars, 1 mm (left) and 20 µm (right). **d**, Fold change in cell number normalized to flat surfaces after 2 days of culture (end of recovery period). Data are quantified from 12 fields from 3 replicates. **e**, Representative maximum-intensity projection images of 10 × 10 pillar arrays immunostained for MBP. Scale bar, 15 µm. **f**, Percentage of pillars with corresponding wrapping score (0–3) (left) and score 3 only (middle and right) across different interpillar distances (middle) and pillar diameters (right). **g**, The *g*-ratio analysis of myelin on pillars of 3, 5 and 10 µm diameter (5-µm spacing). Scale bar, 5 µm. Data are mean ± s.e.m., quantified from eight fields of view from two replicates. All statistics are assessed by one-way ANOVA with Tukey's multiple comparisons test, two-sided, *$P < 0.05$, **$P < 0.01$, ***$P < 0.001$, ****$P < 0.0001$. The exact *P* values are provided in Supplementary Table 1.

compared with flat areas, regardless of pillar diameter. No significant differences (Supplementary Table 1) in cell number were found across different interpillar distances when the pillar diameter was held constant (Fig. 3d). These findings indicate that micropillar arrays enhance

OL cell density independently of their specific geometry (within the tested range), suggesting that spatial constraints may support OL growth and differentiation, as previously reported[8], and consistent with our transcriptomic analysis.

Next, we investigated the effect of micropillar geometry on OL differentiation using myelin basic protein (MBP) as a marker of differentiated OLs at day 7 (Fig. 3e). For each set of arrays, the percentage of pillars that scored 1 or 2 mostly did not change (Supplementary Table 4) when the interpillar distance or pillar diameter were varied (Extended Data Fig. 4). However, the proportion of pillars that scored 3 increased as the interpillar distance increased for 3-µm and 5-µm pillars (Fig. 3f). Interestingly, this trend was not observed for 10-µm pillars, with the vast majority of pillars exhibiting a score of 3 (70.9% ± 7.7%, 76.2% ± 5.9% and 82.5% ± 4.9% (mean ± s.e.m.) for the interpillar distances of 5, 10 and 15 µm, respectively, Fig. 3f). This indicates that decreasing the interpillar distance for thin pillars (for example, 3 µm) reduces the wrapping efficiency and is probably due to an excess of available pillars relative to the number of OLs (the pillar:cell ratio ranged from 1 for 10-µm pillars when $d = 15$ µm to 33 for 3-µm pillars when $d = 5$ µm) (Extended Data Fig. 4c). Independent of the interpillar distance, the proportion of pillars scoring 3 increased as the diameter of pillars increased (Fig. 3f), recapitulating a key feature of in vivo myelination where thicker axons tend to have higher level of myelination[29]. More interestingly, the estimated $g$-ratio for different pillar diameters (0.67 ± 0.03, 0.79 ± 0.01 and 0.88 ± 0.006 (mean ± s.d.) for 3-, 5- and 10-µm pillars, respectively) was comparable to the range observed in the CNS (0.72–0.81)[30] and increased with pillar diameter (Fig. 3g).

## Perturbation of ECM biomechanical properties

The ECM provides biochemical and mechanical cues during oligodendrogenesis[31,32]. Although most studies examined ECM cues using 2D in vitro models[6,7,22,33], incorporating 3D axon-like structures is essential to more accurately mimic the in vivo environment. We therefore assessed OL wrapping efficiency across variations in pillar stiffness and diameter (Fig. 4a–c), surface coating (Fig. 4d–f) and their combined effects (Fig. 4g–i).

Previous studies have shown that OLs respond to mechanical signals, such as substrate stiffness, where soft substrates maintain the progenitor state of OLs[6,7,21]. Therefore, we fabricated pillars with three different stiffnesses: ultrasoft (0.5 kPa) replicating the physiological range of brain tissue elasticity[24], soft (5 kPa) mimicking the stiffness of individual axons[16], and stiff (50 kPa) as a control but yet within the physiological stiffness range of other body tissues[24]. OPCs were cultured and differentiated for 7 days on these pillars and then stained for MBP to assess myelination (Fig. 4a). The number of OLs did not differ across stiffnesses (Fig. 4b). For 5-µm pillars, no statistical difference in myelination was observed between soft and stiff pillars, although myelination was significantly (Supplementary Tables 2 and 5) reduced on ultrasoft ones (Fig. 4b,c and Extended Data Fig. 5a). By contrast, for 10-µm pillars, myelination increased with stiffness, with statistically significant differences (Supplementary Tables 2 and 5) observed between all three stiffness conditions (Fig. 4b,c and Extended Data Fig. 5a). These results highlight the importance of carefully adjusting the mechanical properties of axon-like structures in vitro, as both the diameter and the stiffness of the axon can independently regulate myelination.

To study the effect of biochemical cues on myelination, we coated our pillars with three different coatings: PDL (a synthetic and nonspecific adhesion promoter used as a control), laminin (a binding partner of laminin-binding integrins that has been shown to promote OL differentiation[14,22]) and fibronectin (which has been shown to promote OL proliferation and survival[22]) (Fig. 4d–f). While the different coatings did not influence cell numbers (no statistical difference), laminin promoted myelination, measured as an increase in the percentage of fully wrapped pillars (score 3) when compared with PDL coating (Fig. 4e). No statistical differences in the percentage of fully wrapped pillars (score 3) were observed between PDL and fibronectin coatings (Fig. 4e). However, OLs cultured on fibronectin-coated pillars were able to fully wrap more pillars per cell than in the PDL-coated pillar control group

(Fig. 4e). No differences in the wrapping scores 0, 1 and 2 were observed for the various coating conditions (Fig. 4f and Extended Data Fig. 5b). These data suggest that laminin and fibronectin coatings can influence OL myelination.

Lastly, we investigated the combined effect of stiffness and coating (Fig. 4g–i). We observed no differences in cell number between conditions (Fig. 4h). Laminin increased the percentage of fully wrapped pillars (score 3) but without changing the number of fully wrapped pillars per cell (Fig. 4h). Stiffer gels enhanced both parameters independently of the surface coating (Fig. 4h). The detailed scoring system is depicted in Fig. 4i and Extended Data Fig. 5c. These results further emphasize the importance of considering and tuning axon-like structures' stiffness and coating when studying CNS myelination.

## Tunable micropillars for compound testing

To evaluate the predictive accuracy of our tunable platform, we first tested well-characterized pro-myelinating drugs, including benztropine and clemastine[34], on different stiffness conditions (5 kPa versus 20 kPa) (Fig. 5a–c and Extended Data Fig. 6a). As previously reported[19], we found that stiffness influences drug response. No significant differences (Supplementary Table 3) were observed in cell number across conditions (Fig. 5b), while both drugs enhanced OL wrapping efficiency compared with control groups for both 5-kPa and 20-kPa conditions (Fig. 5b,c and Extended Data Fig. 6a). Interestingly, the magnitude of enhanced myelination due to drugs was attenuated on softer pillars that better simulate in vivo axon mechanical properties.

Moreover, we tested three additional drugs on OLs cultured on soft pillars (5 kPa), including two that were previously investigated for their pro-myelination effects. We tested a pro-myelinating agent GSK239512[35,36] (a histamine H3 receptor antagonist) that was identified through high-content screening of ~1,000 compounds. While GSK239512 showed modest positive effects on myelination in a phase 2 clinical trial (NCT01772199) for relapsing–remitting multiple sclerosis, its practical application was limited by side effects[36]. In addition, we tested simvastatin, a cholesterol-lowering drug evaluated for its potential pro-myelination effects in multiple sclerosis. Preclinical studies reported contradictory findings[37], yet a phase 2 clinical trial (NCT00647348) in secondary progressive MS showed encouraging results[38]. While simvastatin exhibits neuroprotective properties, its role in myelination remains uncertain. Despite their mixed clinical outcomes, both drugs significantly (Supplementary Tables 3 and 6) promoted rodent OL myelination in our assay (Fig. 5d–f and Extended Data Fig. 6b). Finally, we tested wiskostatin[39], an inhibitor of neuronal Wiskott–Aldrich syndrome protein (N-WASP), which regulates filopodia extension and process formation (in OPCs, OLs and Schwann cells) that are key processes in axon ensheathment and myelination[40]. In our assay, wiskostatin caused a dose-dependent decrease in the number of fully wrapped pillars (Fig. 5g–i and Extended Data Fig. 6c), further validating the high sensitivity of our assay to myelination modulators.

## Human OL cell myelination

Despite a well-conserved cellular architecture of the cerebral cortex between mouse and human, recent studies showed substantial disparities between homologous human and mouse brain cell types, highlighting the importance of employing human-based cells[41,42]. Here, we demonstrate that our platform supports the culture of human-derived OLs, offering a valuable tool for investigating human-specific myelination processes and modeling neurological disorders involving OL dysfunction. We cultured lateral ganglionic-derived human fetal neural precursor cell (hfNPC)-derived OPCs onto stiff 50-kPa micropillar arrays with $D = 5$ µm, $d = 6$ µm and $h = 17$ µm (Fig. 6a). After differentiation of hfNPCs into OPCs and expansion in cell culture flasks, cells were collected, seeded and cultured on micropillars for 2–3 days in proliferation medium (N medium) which was subsequently switched to differentiation medium. After 9 days of culture, cells were fixed and

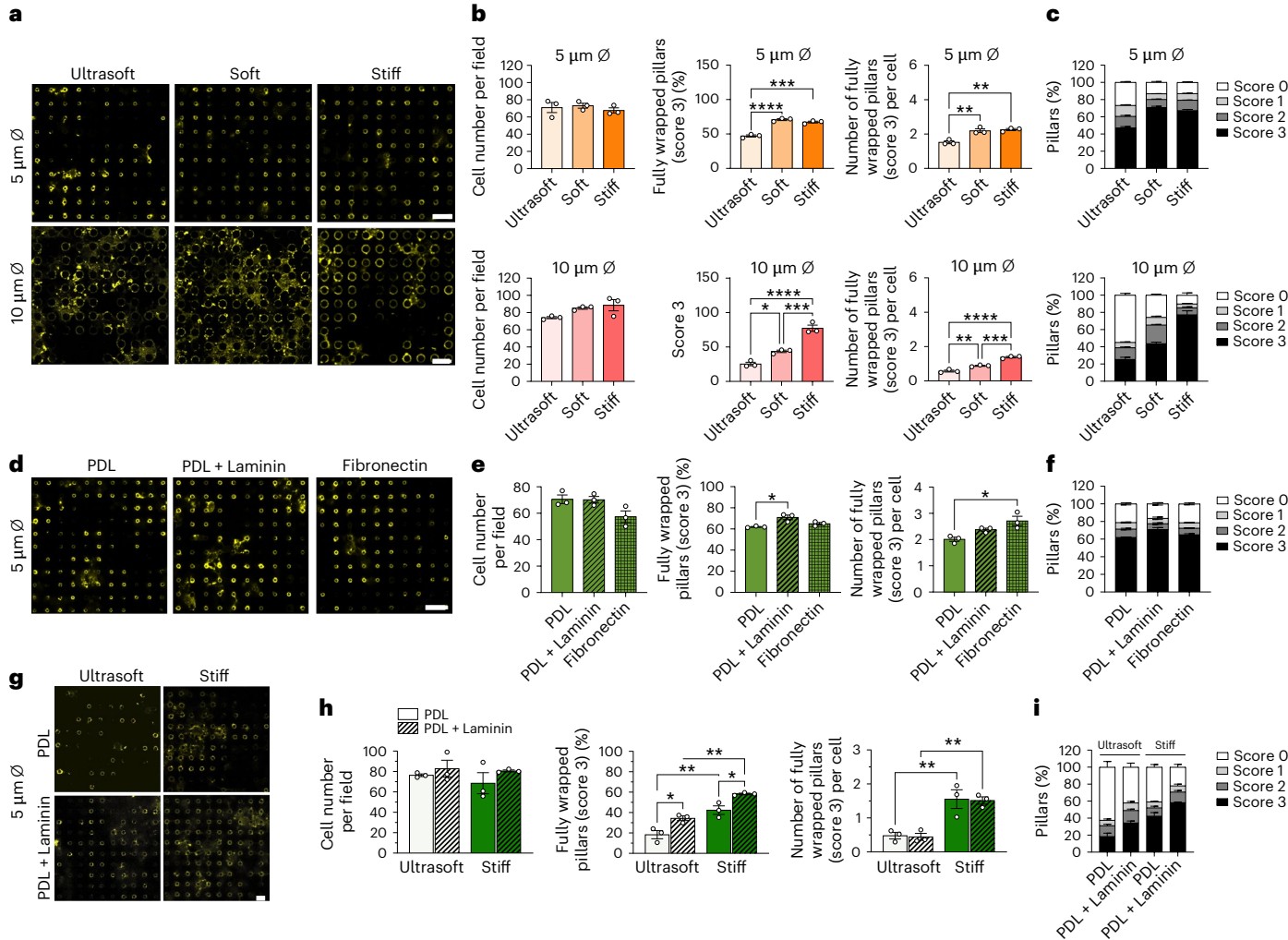

**Fig. 4 | Impact of biochemical and biomechanical cues on OL differentiation and myelination. a**, Representative images of OLs cultured for 7 days on micropillars with varying stiffness (0.5, 5 and 50 kPa) and diameters (5 or 10 μm), stained for MBP. Scale bars, 20 μm. **b**, Quantification of OL number (left), percentage of fully wrapped pillars (middle) and number of fully wrapped pillars per cell (right) for different stiffness and diameter combinations. **c**, Distribution of wrapping scores (0–3) for each stiffness and diameter. **d**, Representative images of OLs cultured on pillars coated with PDL, laminin or fibronectin. Scale bar, 20 μm. **e**, Quantification of OL number (left), percentage of fully wrapped pillars (middle) and number of wrapped pillars per cell (right) for each coating condition. **f**, Distribution of wrapping scores (0–3) across different coatings.

**g**, Representative images of OLs cultured on pillars with different combinations of stiffness (0.5 and 50 kPa) and coating. Scale bar, 10 μm. **h**, Quantification of OL number (left), percentage of fully wrapped pillars (middle) and number of wrapped pillars per cell (right) across combined conditions. **i**, Distribution of wrapping scores (0–3) for the stiffness and coating combinations. Data are mean ± s.e.m., and experiments were performed in at least three technical replicates from one biological sample of pooled cells; all statistical significances were assessed using one-way ANOVA with Tukey's multiple comparisons test, two-sided, *P < 0.05, **P < 0.01, ***P < 0.001, ****P < 0.0001. The exact P values are provided in Supplementary Table 2.

stained for MBP (Fig. 6b). The hfNPC-derived OLs showed a network of long connected processes compared with rat OLs (Fig. 6b). With a count of 49.3 ± 2.9 cells per field (Fig. 6c), we found that 52.5% ± 3.4%, 13.7% ± 1.5%, 16.8% ± 1.7% and 17.1% ± 3.6% (mean ± s.e.m.) of the pillars scored of 3, 2, 1 and 0, respectively (Fig. 6d). We observed that, on average, each OL fully wrapped (score 3) 2.34 ± 0.05 pillars (Fig. 6d). Cells were also cultured for 14 days to assess the nanostructure of myelin around pillars with TEM. Noncompact myelin wrapping was observed (Fig. 6e), indicating initiation of myelination by human fetal OLs. The absence of compact, multilayered myelin suggests that additional extrinsic cues or prolonged maturation may be required to achieve fully structured myelin in fetal cells.

To further validate the versatility of our platform, we tested an additional source of human OLs: human pluripotent stem (hPS) cell-derived O4⁺ cells. hPS cells were first differentiated into OLIG2⁺ progenitors and subsequently into O4⁺ OL lineage cells. These cells were expanded, cryopreserved and stored in liquid nitrogen until use. Upon

thawing, the cells were seeded onto stiff micropillar arrays ($D = 5$ μm, $d = 6$ μm and $h = 17$ μm) and maintained in proliferation medium for 2–3 days before switching to a myelination-promoting medium (Fig. 6f). After 9 days of culture, cells were fixed and stained for MBP (Fig. 6g) showing an average of 85.7 ± 4.9 (mean ± s.e.m.) cells per field (Fig. 6h). Quantification showed that 71.4% ± 2.3% of pillars were fully wrapped (score 3), with an average of 1.89 ± 0.07 fully wrapped pillars per OL (Fig. 6i). SEM at day 9 revealed that OLs extended numerous long processes (Fig. 6j) while TEM performed at day 14 confirmed the presence of compact multilayered myelin sheaths (Fig. 6k).

## Discussion

Our hydrogel-based axon-mimetic platform presents a substantial advance in CNS myelination modeling by integrating five key capabilities: physiologically relevant biomechanical and physical replication, ultrastructural validation of myelin, human OL compatibility, transcriptomic insight and broad experimental accessibility. Alternative

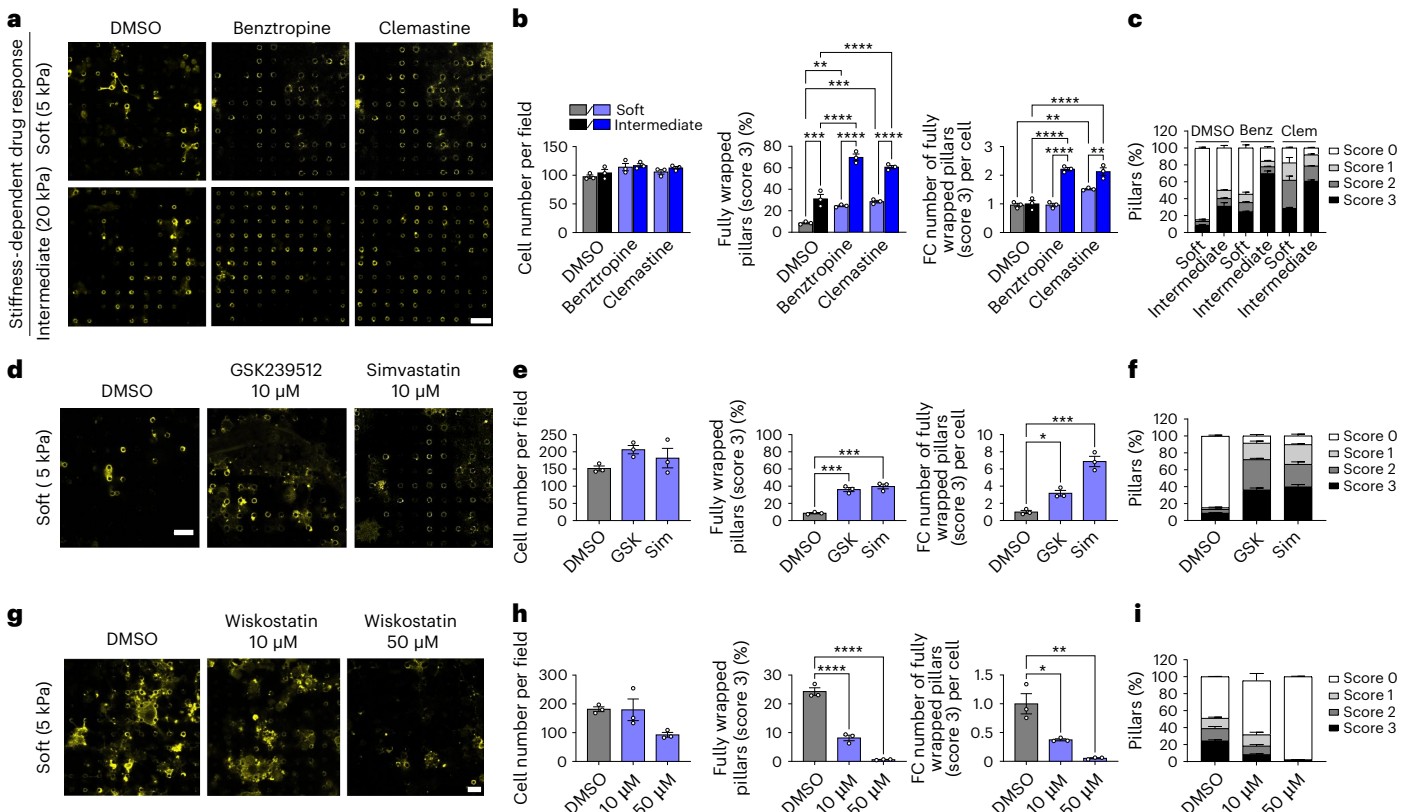

**Fig. 5 | Micropillar assay enables stiffness-dependent compound screening.**
**a**–**c**, OLs on soft (5 kPa) and intermediate (20 kPa) pillars were treated with DMSO, benztropine (benz; 1 µM) or clemastine (clem; 10 µM), stained for MBP as representative images (**a**; scale bar, 20 µm) and quantified for cell number (**b**, left), percentage of fully wrapped pillars (score 3) (**b**, middle), fold change of the number of wrapped pillars per cell standardized on DMSO (**b**, right) and wrapping score distribution (0–3) across conditions (**c**). **d**–**f**, OLs on soft (5 kPa) pillars were treated with DMSO, GSK239512 (10 µM) or simvastatin (10 µM), stained for MBP (**d**; scale bar, 20 µm), and analyzed as in **b** and **c** (**e** and **f**).

**g**–**i**, OLs on soft (5 kPa) pillars were treated with DMSO or wiskostatin (10 or 50 µM), stained for MBP (**g**; scale bar, 15 µm) and quantified as in **b** and **c** (**h** and **i**). For all conditions, rat OLs were cultured on micropillars (5 µm diameter, 22 µm height, 10 µm spacing). Data are the mean ± s.e.m. from at least three technical replicates from one biological sample of pooled cells. All statistical significances were assessed using one-way ANOVA with Tukey's multiple comparisons test, two-sided, *$P < 0.05$, **$P < 0.01$, ***$P < 0.001$, ****$P < 0.0001$. The exact $P$ values are provided in Supplementary Table 3.

approaches like electrospun fiber platforms require specialized equipment, are time-intensive, complicate OL process quantification owing to fiber entanglement and cell body overlap in confocal images, and patterning fibers with high reproducibility remains challenging[34]. While Bechler et al.[15] demonstrated compact myelin formation on electrospun poly-L-lactic acid fibers, their use of rigid materials and limited control over geometry highlight inherent limitations. Ong et al.[43] developed a tunable-stiffness fiber platform, but their fibers remained in the MPa–GPa range and required complex electrospinning. Espinosa-Hoyos et al.[14] and Jagielska et al.[18] approached physiological stiffness (0.4–140 kPa) using 3D-printed poly(HDDA-co-starPEG) but were limited by resolution constraints of projection microstereolithography, restricting pillar diameter to 16-µm and over 20-µm interpillar distance for soft pillars (<1 kPa), far larger than most CNS axons. Carvalho et al.[44] used a mold-based fabrication to generate ultrastiff (~400 kPa) PDMS micropillars of 1–5 µm diameter but without addressing the limitations of PDMS[45] as a cell culture substrate.

Our system, using a well-established hydrogel and a reusable mold fabricated via standard photolithography, allows high-resolution control of geometry and spacing across the array, enabling simultaneous testing of diverse microenvironments with subcellular axon-like geometry (1–15 µm diameter, 5–15 µm interaxonal distance) and CNS-relevant mechanical properties (500–5,000 Pa) supporting ECM coating and optimal OL adhesion. However, it is important to consider that the creation of high-aspect-ratio (>1:10) ultrasoft pillars (<1 kPa)

can pose challenges as they are more susceptible to pillar collapse or damage during the demolding process. Despite this limitation, polyacrylamide micropillars remain favourable when studying myelination, as soft micropillars (5 kPa) were shown to enhance myelin wrapping, are less prone to damage compared with ultrasoft pillars and more closely match the stiffness of axons[16]. OLs cultured on micropillar arrays exhibited transcriptional profiles that align with human mature brain-resident OLs, while those on flat gels remained closer to OPCs. This reinforce the link between biophysical architecture and OL differentiation and highlight our platform's capacity to study mechanobiology in tandem with molecular maturation.

Polyacrylamide is well suited for studying myelination, offering tunable mechanics, ligand coating, porosity for controlled release of bioactive molecules (for example, neuregulin-1) and the potential to be ionically or electrically conductive to investigate the impact of electrical activity. These properties are essential for an in vitro assay to capture key complexities affecting the degree of myelination and to replicate the heterogeneity of the CNS regions, as they have different ECM components[46], stiffness[47,48] and axon geometry[49]. While stiffer substrates (50 kPa) promoted myelination, our focus remains on mimicking in vivo conditions over optimizing artificial output. Indeed, by examining how substrate stiffness affected drug response, we confirmed that increased stiffness alone can promote OL differentiation and myelination[19], potentially masking the effects of candidate compounds. This underscores a key limitation of overly rigid in vitro

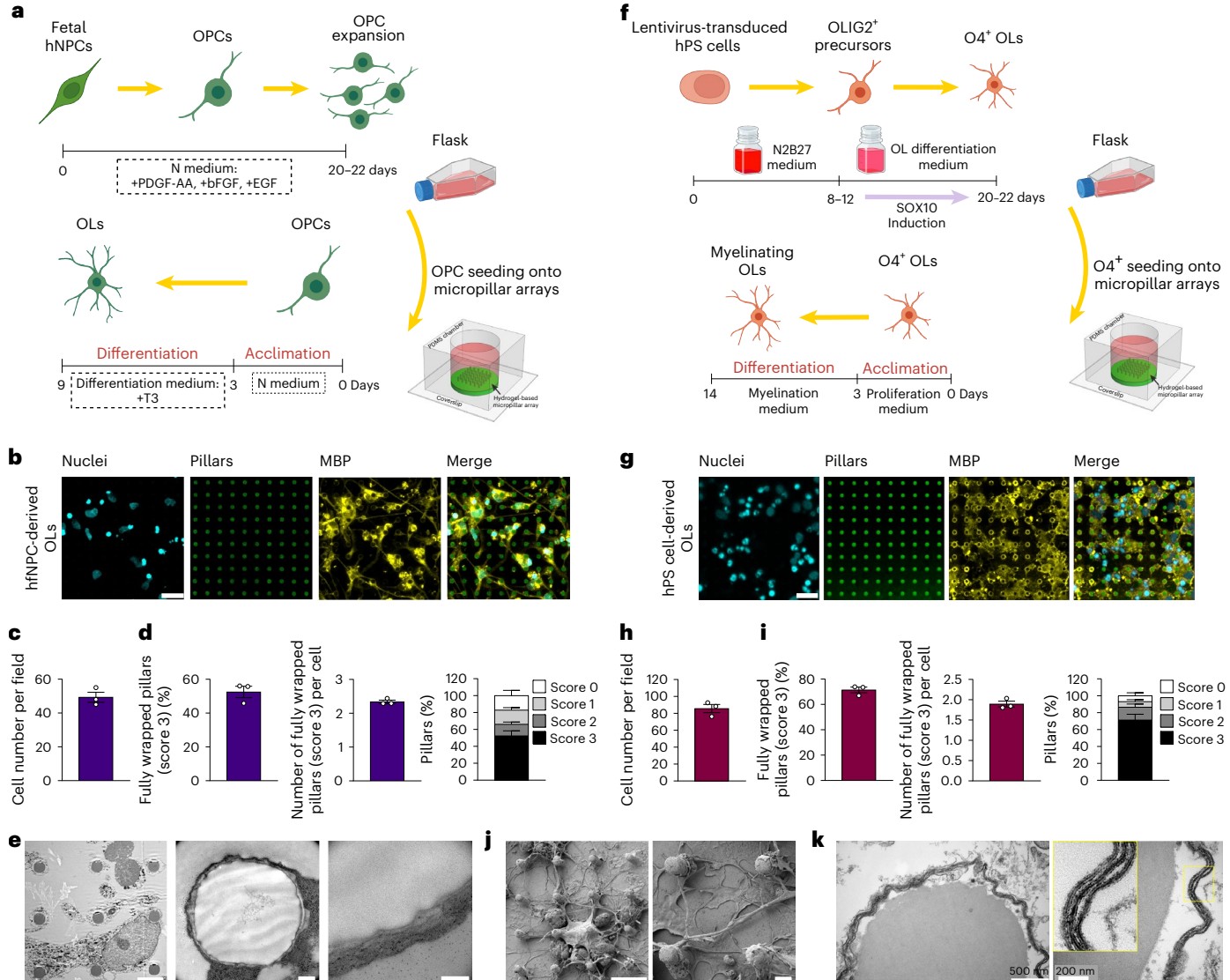

**Fig. 6 | Validation of micropillar myelination by human OLs. a–e,** Human fetal OLs cultured on micropillars: schematic and timeline of fetal human OPC culture (**a**); maximum *Z* intensity projection images of pillars wrapped by human fetal OLs after 9 days of culture (cyan is nuclei stained by Hoechst, green is pillars stained by FITC, and yellow is MBP) (scale bar, 20 μm) (**b**); average cell count per field (15 × 15 micropillars) (**c**); percentage of fully wrapped pillars (score 3) (left), number of fully wrapped pillars (score 3) per cell (middle) and percentage of pillars with wrapping scores of 0–3 (right) (data are mean ± s.e.m. with *n* = 3 fields) (**d**); TEM images of human fetal OLs wrapped around micropillars after 14 days of culture (scale bars, 5 μm (left), 500 nm (middle) and 200 nm (right)) (**e**).

**f–k,** hPS cell-derived OLs cultured on micropillar assay: schematic of hPS cell-derived O4+ cell culture and differentiation (**f**); representative images (scale bar, 20 μm) (**g**) and analyses of cell number (**h**) and OL wrapping (**i**); data are mean ± s.e.m. from three technical repeats (**g–i**). **j,** SEM images of hPS cell-derived OLs and micropillars after 9 days of culture. Scale bars, 10 μm (left) and 2 μm (right). **k,** TEM images of hPS cell-derived OLs and micropillars after 14 days of culture. Scale bars, 0.5 μm (left) and 0.2 μm (right). Human OLs were cultured on stiff micropillar arrays with *D* = 5 μm, *d* = 6 μm and *h* = 17 μm. Schematics in **a** and **b** created in BioRender; Vinel, C. https://biorender.com/ycsm1oy (2026).

---

systems, whereas our soft micropillars more accurately reflect CNS axon mechanics and may reduce false positives, thereby improving the reliability of pro-myelinating compound identification. Specifically, we tested GSK239512 and simvastatin as compounds with conflicting data from preclinical models[35] that ultimately failed in clinical trials[36,38]. This divergence underscores the importance of microenvironmental context, particularly stiffness and 3D architecture, in modulating drug responses. It is possible that some compounds previously dismissed based on 2D data may exhibit beneficial effects in more physiologically relevant settings.

To underscore the physiological relevance of our model, beyond conventional immunostaining of myelin markers such as MBP, we conducted detailed TEM analysis to assess myelin ultrastructure. Our 3-μm and 5-μm hydrogel pillars consistently exhibited compact myelin wraps

with no extracellular gaps, and wrap thickness correlated linearly with the number of layers mimicking myelination around axons. Notably, the average number of myelin wraps (~3.5) observed in our assay is lower than the typical seven to nine layers reported for small-diameter axons in the adult rodent cortex[50]. This difference probably reflects an immature myelination state, consistent with developmental stage and in vitro conditions, yet supports the physiological relevance and in vivo fidelity of our model. However, with an average thickness of 13 nm per single lamella, our results fall well within the range reported by Basu et al.[50] for the adult rat cortex (3.9–17.1 nm). Moreover, we observed that 50% of pillars showed multilayered myelin by TEM at D14, closely matching MBP-based quantification (60% fully wrapped at D7), suggesting a promising correlation between ultrastructural and immunostaining-based assessments. This could enable efficient,

high-throughput myelination screening using light microscopy as a surrogate for EM using our validated system. In addition, a major strength of our platform lies in its compatibility with human-derived OLs, including fetal and hPS cell-derived sources. This represents a critical step forward in translational relevance, as most prior systems rely on rodent cells. Our platform enables functional myelination by human OLs, as indicated by multilayered myelin formation by TEM, thereby expanding the utility of this assay for disease modeling and drug screening.

Altogether, this work presents a versatile and physiologically relevant in vitro platform for studying CNS myelination that combines biomimetic geometry, tunable mechanics and compatibility with both rodent and human-derived OLs. Our findings underscore the critical role of physical microenvironment in shaping myelin formation and highlight the importance of modeling both axonal stiffness and geometry to accurately evaluate pro-myelinating therapies. By enabling distinction between true pharmacological effects and stiffness-driven artefacts, the platform improves the predictive value of drug screening assays. The system's scalability, reusability and compatibility with standard fabrication techniques make it broadly accessible across disciplines, while its adaptability to high-throughput formats and potential integration with signaling gradients or bioelectrical cues open further avenues for innovation. Importantly, future applications involving patient-derived induced pluripotent stem cells will allow disease-specific modeling of myelin pathologies and facilitate personalized screening of therapeutic candidates, advancing precision medicine approaches for disorders such as multiple sclerosis and rare leukodystrophies. Together, these features position our platform as a powerful tool for uncovering mechanistic insights and accelerating translational discovery in demyelinating disease research.

## Online content

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

## Methods

### Mold microfabrication

Silicon molds were fabricated in the cleanroom at the London Centre for Nanotechnology. In brief, a thin layer of positive photoresist (S1805) was coated onto a 3- or 4-inch silicon wafer and then exposed to UV light either without a mask (using a Heidelberg DWL66+) or with a photomask using a mask aligner (Quintel mask aligner). The wafer was then developed and mounted on a ceramic carrier using Santovac5 oil for DRIE (STS ICP ASE), with the number of cycles determined by the specific micropillar height. Finally, the resist was removed using dimethyl sulfoxide (DMSO) and cleaned for 5 min using $O_2$ plasma (Diener plasma asher). To ease the demolding process of the polyacrylamide gels, Rain-X solution was applied on the surface of the molds.

### Polyacrylamide gel preparation

Acrylamide (AAm, Sigma; A4058), bis-acrylamide (bis-AAm, Severn Biotech; 20-2500-05) and acrylic acid (AA, Fischer Scientific; 11470570) were mixed together in PBS at different ratios to produce gels with different stiffnesses (Extended Data Table 2). The gel pH was raised to 8 with 10 N NaOH, fluorescently labeled with FITC Dextran (Sigma; FD2000S; 5% v/v from a 1 mg ml$^{-1}$ stock solution) and stored in the fridge as a stock solution. A fresh 10% stock solution of ammonium persulfate (Sigma; A3678) was added to the comonomer solution at a final concentration of 0.1%, and the mixture was purged with nitrogen for 1–2 min to remove oxygen. Thin PDMS spacers were placed on the mold around the micropatterns, and 120 µl of the purged comonomer was dispensed inside. The wafer was placed in a desiccator under vacuum for 5–40 min (depending on the pump's power) to remove all air from the microwells. The mold was then placed in an ultrasonic cleaner (OneCall Farnell; Shesto SI16822) for 1 min if bubbles were trapped. Tetramethylethylenediamine (TEMED; Sigma; 411019) was finally added to the polyacrylamide precursor solution at a working concentration of 0.1%, gently mixed, and immediately covered with cut Lonza GelBond support film sheets (Scientific Laboratory Supplies; LZ54711). The solutions were kept covered for 10–45 min until complete polymerization. The system was then incubated in PBS for 5 min to facilitate the peeling of the gels from the mold. After removing the GelBond supports from the wafer, the gels were washed thoroughly in PBS and stored in the fridge until use.

### Device fabrication and functionalization

Autoclaved PDMS wells were glued (using uncured PDMS) to Gelbond coverslips overnight in the incubator (with PBS to avoid polyacrylamide drying). Devices were then sterilized under UV for 40 min inside a tissue culture hood. Polyacrylamide hydrogels were washed in 0.1 M of 4-morpholineethansulfonic acid (MES) hydrate (Sigma; M5287) with 0.5 M NaCl (SLS; S9625) in deionized water (MES buffer, pH 6) for 10 min. They were then incubated for 30 min in MES buffer (pH 6) supplemented with 0.1 M N-hydroxysuccinimide (Sigma; 130672) and 0.2 M N-(3-dimethylaminopropyl)-N′-ethylcarbodiimide hydrochloride (Sigma; E7750). They were washed thoroughly in PBS and incubated for at least 1 h at 37 °C with the ligand solution. The desired ECM protein: PDL (Sigma; P6407) or fibronectin (Sigma; FC010) was diluted in 50 mM HEPES buffer (pH 8, Fisher Scientific; 15405479) at a concentration of 1 and 0.1 mg ml$^{-1}$, respectively. Laminin (Sigma; L2020) was coated on top of PDL at a concentration of 0.02 mg ml$^{-1}$ (in PBS) for at least 2 h at 37 °C. Polyacrylamide gels were finally rinsed three times with PBS and stored immersed in PBS in the fridge for up to 1 week.

### Primary rat OL isolation

All procedures were performed in accordance with licenses held under the UK Animals (Scientific Procedures) Act 1986 and subsequent modifications, and in accordance with all relevant guidelines and regulations.

Cerebral cortices from two mixed-sex P6–P7 wild-type Sprague-Dawley Crl:CD (SD) rat neonates were isolated, minced and enzymatically dissociated for 30 min at 37 °C in papain solution consisting of 20 U ml$^{-1}$ papain (Sigma; P4762), 0.24 mg ml$^{-1}$ L-cysteine (Sigma; C7352) and 2,500 U DNase I (Sigma; D5025) in HBSS without $Ca^{2+}/Mg^{2+}$ (Thermo Fisher Scientific; 14175095). Digestion was inhibited using 1 mg ml$^{-1}$ ovomucoid trypsin inhibitor (Sigma; T9253) and 1 mg ml$^{-1}$ bovine serum albumin (Sigma; A3311) in PBS. Cells were then mechanically triturated using a 10-ml serological pipette tip, followed by a 1-ml pipette tip to obtain a single-cell suspension. The solution was filtered using two sequential 40-µm (SLS; 352340) cell strainers. The solution volume was brought to 28 ml using PBS containing 10% fetal bovine serum (Thermo Fisher Scientific; 26140079) and 2,500 U DNase I. Cells were centrifuged at 200g, resuspended in 14 ml of PBS with 10% fetal bovine serum and 2,500 U DNase I, and centrifuged again to remove debris.

The immunomagnetic isolation of O4$^+$ OLs was performed using 50-nm superparamagnetic anti-O4 microbeads (Miltenyi Biotec; 130-094-543) according to the manufacturer's instructions, with minor modifications. A fresh solution of magnetic cell sorting (MCS) buffer was prepared by adding 1 mg ml$^{-1}$ D-glucose (Sigma; G8270) and 5 mg ml$^{-1}$ (0.5%) bovine serum albumin (Sigma; A3311) to 0.5 mM EDTA–PBS (Sigma; E8008). Cells were resuspended in 85 µl of MCS buffer and 15 µl of anti-O4 bead solution per $10^7$ cells, incubated for 15 min in the fridge, washed in MCS buffer and centrifuged at 200g for 10 min. Cells were then resuspended in 500 µl of MCS buffer per $10^7$ cells, passed through a MS column (Miltenyi Biotec; 130-042-201) using a MiniMACS separator (Milteny; 130-042-102) bound to a MACS multistand (Miltenyi Biotec; 130-042-303) and washed three times with 500 µl of MCS buffer inside the column to remove unlabeled cells. Finally, the column was removed from the separator and flushed out with cell culture medium using a plunger.

### Primary rat OL culture

Primary rat OLs were counted, seeded at a density of 100,000 cells cm$^{-2}$ and cultured in proliferation medium for the first 2 days. Proliferation medium consisted of Dulbecco's modified Eagle medium (DMEM) (Thermo Fisher Scientific; 31966-021) mixed with neurobasal medium (Thermo Fisher Scientific; 21103049) at a 1:1 ratio supplemented with 1× N2 (Thermo Fisher Scientific; 17502048), 1× B27 (Thermo Fisher Scientific; 17504044), 5 µg ml$^{-1}$ N-acetyl-L-cysteine (NAC) (Sigma; A8199), 2 µg ml$^{-1}$ (5 µM) forskolin (Cambridge Bioscience; SM18), 10 ng ml$^{-1}$ biotin (Sigma; B4639), 10 ng ml$^{-1}$ ciliary neurotrophic factor (CNTF) (Peprotech; 450-13), 1 ng ml$^{-1}$ neurotrophin-3 (NT-3) (Peprotech; 450-03), 10 ng ml$^{-1}$ platelet-derived growth factor (PDGF-AA) (Peprotech; 100-13 A) and 1% penicillin–streptomycin. After 2 days, the proliferation medium was changed to myelination medium by removing PDGF-AA and adding 30 ng ml$^{-1}$ T3 (Sigma; T6397).

### Human OL culture

To generate hfNPC-derived OPCs, hfNPCs were differentiated to OPCs and expanded using poly-L-ornithine (PLO)+laminin-coated dishes during 3 weeks in a modified N medium. N medium consisted of DMEM–F12 (Thermo Fisher Scientific; 31331028) supplemented with 1× N2, 0.25× B27, 5 mM HEPES (Thermo Fisher Scientific; 15630049), 20 µg ml$^{-1}$ insulin (Sigma; I9278), 6 mg ml$^{-1}$ glucose, 20 ng ml$^{-1}$ epidermal growth factor (Peprotech; AF100-15), 10 ng ml$^{-1}$ PDGF-AA and 10 nM XAV 939 (Cambridge Bioscience; 13596-1mg-CAY). Medium was changed every 2–3 days. The OPCs were then seeded at a density of 75,000 cells cm$^{-2}$ on PLO+laminin-coated micropillars in N medium for 2–3 days. Then, medium was shifted to differentiation, which consisted of DMEM–F12 supplemented with 1× N2, 1× B27, 100 ng ml$^{-1}$ biotin (Sigma; B4639), 1 µM cAMP (Sigma; D0260), 25 µg ml$^{-1}$ insulin, 0.5 µM smoothened agonist (Sigma; 566660), 100 µM ascorbic acid (Sigma; A92902), 60 ng ml$^{-1}$ T3 and 1% penicillin and streptavidin.

To generate hPS cell-derived OLs, the human embryonic stem cell H9 line (WiCell) modified with an insertion of the SOX10 CDS in the AAVS1 locus[51] was first differentiated to OLIG2+ cells in N2B27 medium (DMEM/F12 medium supplemented with non-essential amino acids, 2-mercaptoethanol, penicillin–streptomycin, N2 and B27 (without vitamin A) to 1× concentration plus insulin at 25 µg ml$^{-1}$) supplemented with 10 µM SB431542, 1 µM LDN193189 and 100 nM retinoic acid (RA) for the initial 5 days and with further supplementation of 1 µM smoothened agonist for days 5–8. Cells were then differentiated to O4+ OLs through the forced expression of SOX10, via the addition of doxycycline (2 µg ml$^{-1}$), from the AAVS1 locus in OL differentiation medium (N2B27 supplemented with 10 ng ml$^{-1}$ of PDGFaa, 10 ng ml$^{-1}$ insulin-like growth factor-1 (IGF1), 5 ng ml$^{-1}$ hepatocyte growth factor (HGF), 10 ng ml$^{-1}$ NT3 (all from Peprotech), 100 ng ml$^{-1}$ biotin, 1 µM cAMP, 60 ng ml$^{-1}$ T3 and 2 µg ml$^{-1}$ doxycycline (all from Sigma) for approximately 10 days as previously described[51,52]. Cells were then frozen and stored in liquid nitrogen until use. For culture of human O4+ OLs on PLO+laminin-coated micropillar arrays, cells were quickly thawed and seeded at 100,000 cells cm$^{-2}$ in proliferation medium for 2–3 days, with the addition of RevitaCell (1×; Thermo Fisher Scientific; A2644501) for the first 24 h to increase cell survival. Proliferation medium consisted of DMEM–F12 supplemented with 1× N2, 1× B27, 25 µg ml$^{-1}$ insulin, 1× non-essential amino acids (Thermo Fisher Scientific; 11140050), 50 fM 2-mercaptoethanol (Thermo Fisher Scientific; 31350010), 100 ng ml$^{-1}$ biotin, 1 µM cAMP, 2 µg ml$^{-1}$ doxycycline (Sigma; D9891), 1% penicillin–streptomycin, 10 ng ml$^{-1}$ PDGF-AA, 10 ng ml$^{-1}$ IGF1 (Peprotech; AF-100-11), 5 ng ml$^{-1}$ HGF (Peprotech; 100-39H) and 10 ng ml$^{-1}$ NT-3. After that, PDGF-AA and HGF were removed from the medium and 60 ng ml$^{-1}$ T3 was added to induce maturation of OLs (myelination medium).

## Immunocytochemistry

Cells were washed in PBS, fixed for 15 min in 4% PFA (Thermo Fisher Scientific; 28908) at room temperature and washed in PBS three times. Cells were then permeabilized using 0.1% saponin (Sigma; 47036) for 10 min and blocked using 5% normal goat serum (Abcam; ab7481) in PBS (blocking buffer) for 1 h at room temperature. Cells were then incubated with the primary antibody rat anti-MBP (Bio-Rad; MCA409S, clone 12) at a concentration of 1:100 in blocking buffer for 1 h at room temperature. Samples were washed three times in PBS and incubated with the secondary antibody goat anti-rat IgG (Thermo Fisher Scientific; A21094; 1:500 in PBS) for 1 h at room temperature. Cells were finally washed three times in PBS, stained with 1 µg ml$^{-1}$ Hoechst (New England Biolabs; 4082S) for 10 min at room temperature, and stored in the fridge until imaged. Images were acquired with an upright confocal microscope (Leica SP8 or Zeiss LSM 980) using a 25× water-dipping objective. Z-stacks were acquired at 2-µm intervals at random locations within the arrays, images were processed using ImageJ for MBP analysis and 0.5-µm intervals were applied to obtain 3D high-resolution images. Images were acquired using LAS X Life software and processed and analyzed using ImageJ2 (v2.16.0/1.54p). For g-ratio analysis, myelin thickness was measured with ImageJ using the following formula: $g = D/(D + 2t)$, where $D$ is the pillar diameter and $t$ is the myelin thickness.

## AFM measurements

The Young's modulus of polyacrylamide hydrogels has been determined using AFM with the use of a JPK Nanowizard Cellhesion 200 module mounted on an inverted microscope (Olympus FV1000). A 25- or 50-µm polystyrene bead was glued to a cantilever (MLCT, Bruker AFM probes) with an appropriate nominal spring constant $k$ (ranging from 0.01 to 0.6 N m$^{-1}$). The thermal noise method in the JPK integrated software was used to calibrate the cantilever's sensitivity and spring constant in PBS. Hydrogel samples were glued into a Petri dish, and indentation measurements (3 × 3 grid) were performed in PBS. Force–indentation curves were then processed using the integrated JPK data

processing software. In brief, the curves were smoothed, subtracted from the baseline, adjusted to the contact point, corrected for cantilever bending (vertical tip position) and fit to the Hertz model assuming a Poisson ratio of 0.5. The stiffness (Young's modulus) of polyacrylamide was calculated as the average of nine measurements per gel.

## Flow cytometry

After immunomagnetic cell isolation, two fractions (each containing 100,000 cells) were analyzed. The original fraction, collected before the isolation procedure, contained diverse brain cell types, whereas the positive fraction comprised only those retained within the column following MACS. Cells were incubated with anti-O4–phycoerythrin (PE) (Miltenyi Biotec; 130-117-507; 1:100) for 30 min in the refrigerator and then washed. To measure cell viability, 1 µl of 100 µg ml$^{-1}$ of propidium iodide (PI) (Sigma; P4864) was added to the cell suspension and incubated for 1–2 min before analysis with a BD Accuri C6 flow cytometer. FlowJo software was used to analyze the data.

## Drug assays

To test the effect of drugs in our myelination assay system, cells were cultured in proliferation medium for the first 2 days as usual. Then, the medium was switched to the differentiation formulation without T3. Instead, DMSO (Sigma; D2650), benztropine (Generon; 1554) at 1 µM, clemastine (Generon; B1558), GSK239512 (Cambridge Bioscience; T27462), simvastatin (Cambridge Bioscience; CAY10010344) and wiskostatin (Tocris; 4434) at 10 µM were used. All conditions included the same amount of DMSO (1:1,000), and the medium was changed every other day.

## Electron microscopy

Samples were washed in PBS and fixed in 2% PFA/1.5% glutaraldehyde (both EM grade) in 0.1 M phosphate buffer for 30 min at room temperature. After washing in PBS, cells were stained in 1% osmium tetroxide/1.5% potassium ferricyanide for 1 h at 4 °C and washed three times. For SEM, samples were stained with 1% tannic acid in 0.05 M cacodylate buffer for 40 min, washed and dehydrated in a series of ethanol dehydration steps (70%, 90% and 2×100% ethanol) for 10 min each. Finally, critical point drying (using EM CPD 300 from Leica) was applied, and the samples were mounted and coated with gold (10–15 nm thick) before imaging with a Zeiss Gemini 300 using the SE2 detector with a landing energy of 1.5 kV and a working distance of approximately 7 mm. For TEM, samples were stained with thiocarbohydrazide, 2% osmium tetroxide, 1% uranyl acetate and lead aspartate at 60 °C, and dehydrated in a series of ethanol dehydration steps (70%, 90% and 2× 100% ethanol) for 10 min each. Dehydration was completed in propylene oxide (PO), and samples were embedded in a 1:1 mix of PO and epon resin, followed by 100% epon resin overnight. Samples were placed in 100% fresh epon resin and polymerized at 60 °C overnight. Ultrathin sections (70 nm) were obtained with a Leica UC7 ultramicrotome using a Diatome diamond knife. Images were acquired with a Tecnai Spirit microscope at 120 kV, equipped with an Olympus SIS Morada camera.

## RNA sequencing and transcriptomic analysis

Differentiated primary rat OLs cultured on 50-kPa flat or micropillar ($d = 10$ µm, $D = 5$ µm and height of 22 µm) hydrogels were directly lysed with Qiazol, and total RNA was extracted following the manufacturer's instructions (Zymo; #2060). RNA samples were sequenced on an Illumina platform (2 × 150 bp paired-end) to a depth of 20 million paired-end reads per sample after poly(A) selection by Azenta Life Sciences.

Sequence reads were trimmed to remove possible adapter sequences and low-quality nucleotides using Trimmomatic v.0.36. The trimmed reads were mapped to the *Rattus norvegicus* Rnor6.0 reference genome available on ENSEMBL using the STAR aligner v.2.5.2b. The STAR aligner is a splice aligner that detects splice junctions and

incorporates them to help align the entire read sequences. Unique gene hit counts were calculated using featureCounts from the Subread package v.1.5.2. The hit counts were summarized and reported using the gene_id feature in the annotation file. Only unique reads that fell within exon regions were counted.

After extracting gene hit counts, the gene hit counts table was used for downstream differential expression analysis. Using DESeq2, gene expression was compared across the custom-defined sample groups. The Wald test was used to generate $P$ values and $\log_2$ fold changes. Genes with an adjusted $P$ value <0.05 and absolute $\log_2$ fold change >1 were defined as DEGs for each comparison.

A published single-cell transcriptomic dataset (GSE218022) was analyzed using Seurat's SCTransform from raw counts and compared with rlog-transformed bulk transcriptomic data from OLs cultured on flat and micropillar hydrogels. To mitigate large-scale technical differences between the datasets, batch effects were corrected using the removeBatchEffect function from the limma package in R, treating dataset origin (bulk versus small nuclear RNA sequencing) as a batch factor. $Z$ scores were calculated across all samples to standardize the data, and a heatmap of Euclidean distances was generated using the pheatmap package in R.

A GO analysis was performed on the statistically significant set of genes using the GeneSCF v.1.1-p2 software. The rgd GO list was used to cluster the set of genes on the basis of their biological processes and determine their statistical significance. A list of genes clustered by GO was generated.

### Statistics and reproducibility

For statistical analysis, representative fields (250 × 250 μm) were randomly acquired from each gel. Data were repeated with several technical and biological replicates as indicated. Processing and quantification of confocal images were performed using ImageJ. Statistical analysis and plotting were performed in GraphPad Prism v9.0 (GraphPad Software). Data are presented as either mean ± s.d. or mean ± s.e.m. One-way analysis of variance (ANOVA) was used with Tukey's post hoc comparison two-sided to evaluate statistical significance, and $P$ values < 0.05 were considered statistically significant (*$P$ < 0.05, **$P$ < 0.01, ***$P$ < 0.001).

### Reporting summary

Further information on research design is available in the Nature Portfolio Reporting Summary linked to this article.

### Data availability

The authors declare that all data supporting the findings of this study are available within the Article, its Supplementary Information and the public repositories as stated below. The datasets generated in this study are available as raw data in the NCBI Gene Expression Omnibus (GEO) database with SuperSeries number GSE301308, which includes transcriptomic data. Publicly available single-cell transcriptomic data used in the study are available at GSE218022 (neurotypical patients GSM6732893). Source data are provided with this paper.

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

## Acknowledgements

We thank all members of the EM laboratory, L. Ruban and B. O'Sullivan for their help and support. We acknowledge support from the UCL Department of Mechanical Engineering and Yiannis Ventikos for a PhD studentship to S.L. and E.M. This work was supported by the Leverhulme Trust Research Project Grant (RPG-2018-443) to E.M. and G.K.S., EPSRC (EP/W009889/1), BBSRC (BB/V001418/1), UCL's HEIF Knowledge Exchange & Innovation Fund, UCL's ARUK Network Grant to E.M., and the Sumaira Foundation grant TSF_SPARK_2023_03 to J.A.G.L. We also thank M. K. Tiwari, S. Bertazzo and N. Szita for their support.

## Author contributions

S.L., C.V. and E.M. conceptualized the study. S.L., C.V. and E.M. developed the methods. S.L. and C.V. performed the experiments with help from A.A., Y.J. and B.D. B.G.D. and J.A.G.-L. supported human cell cultures. I.J.W. acquired EM images. P.P., G.K.S. and W.D.R. provided scientific guidance and support. S.L., C.V. and E.M. wrote the manuscript with contributions from all authors. E.M. supervised the study.

## Competing interests

E.M. is the founder of BioRecode Ltd. The other authors declare no competing interests.

## Additional information

**Extended data** is available for this paper at https://doi.org/10.1038/s41592-026-03048-3.

**Correspondence and requests for materials** should be addressed to Graham K. Sheridan, William D. Richardson or Emad Moeendarbary.

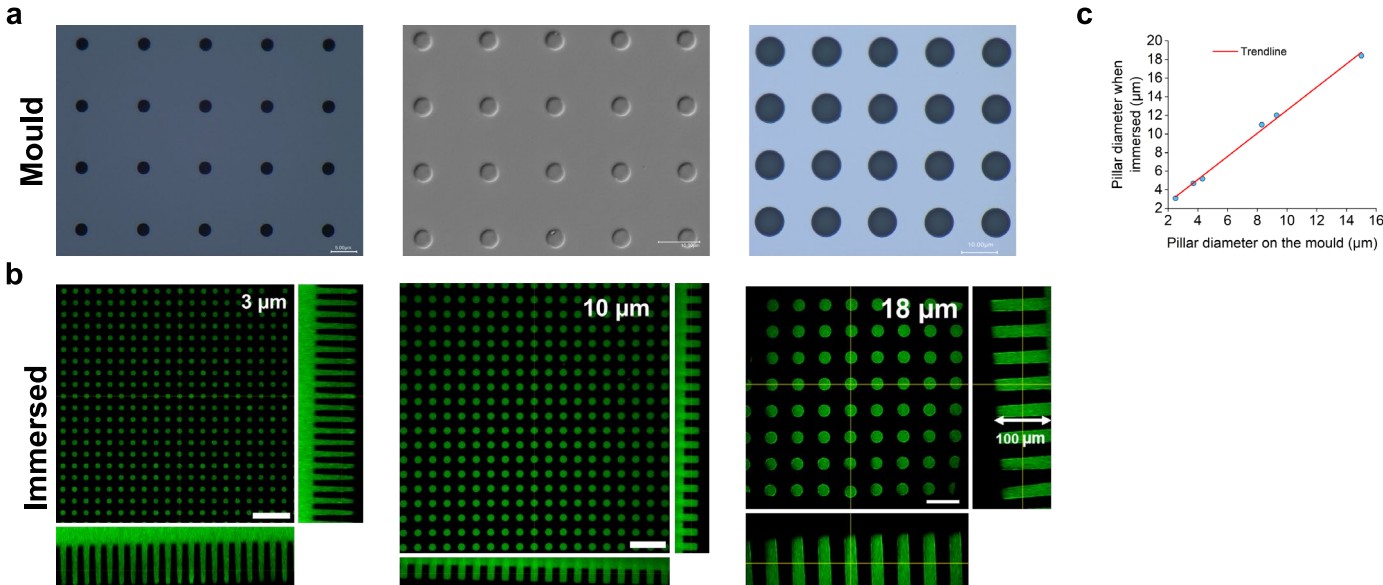

**Extended Data Fig. 1 | Images of micropillars with different geometries.**
**a)** Images of the mould micropattern showing different pillar diameters from 2, 4 and 8 μm. Scale bar = 5 μm (left) and 10 μm (middle and right). **b)** Corresponding confocal images of pillars when immersed in PBS. Scale bars = 20 μm for 3 μm and 50 μm for 10 and 18 μm. **c)** The pillar diameter, when immersed in PBS, is plotted against the diameter from the mould ($D_{immersed} = 1.24 * D_{mould} + 0.15$, $R^2 = 0.996$). Each dot represents a measurement from a different mould (n = 6).

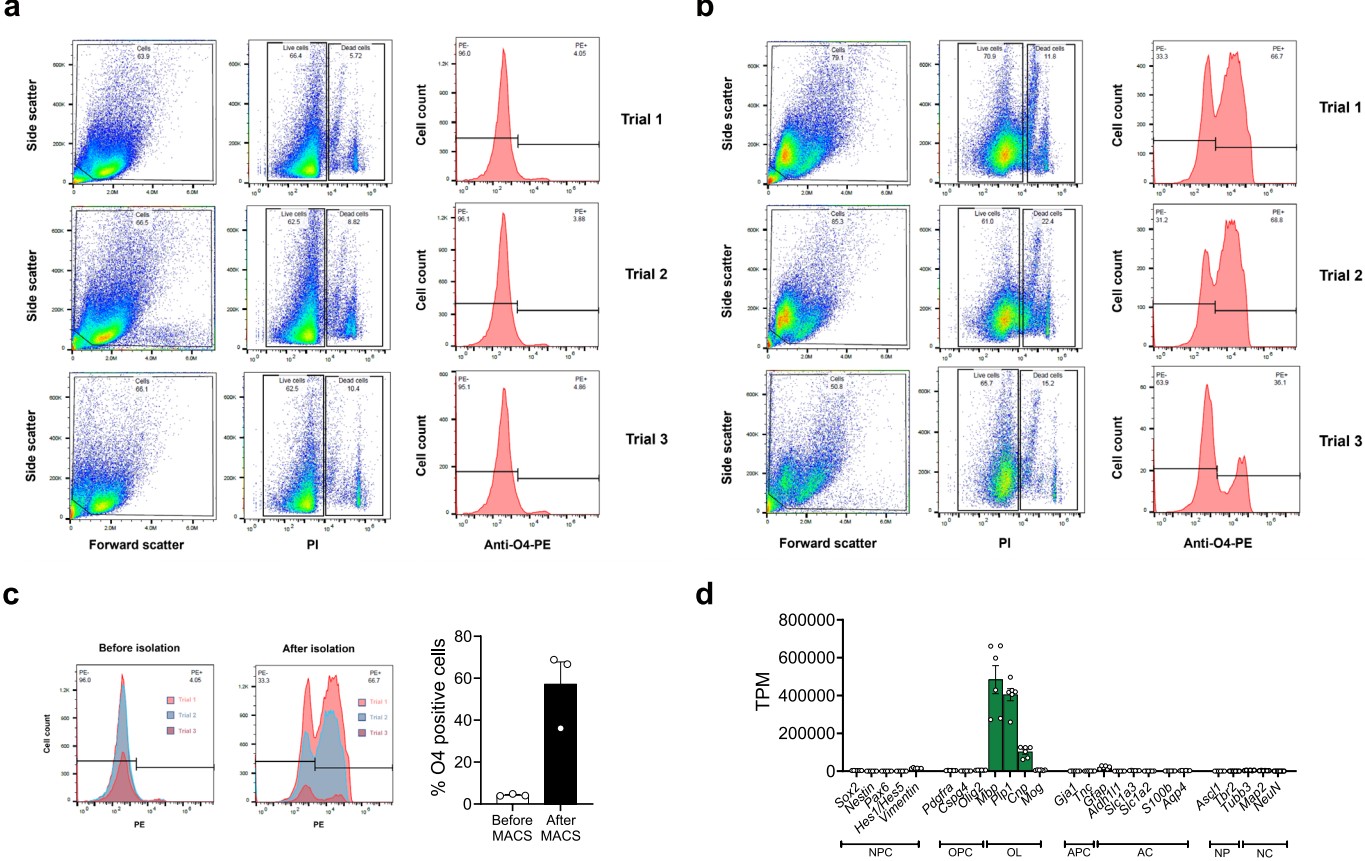

**Extended Data Fig. 2 | OPC purification and OL population characterisation.**
**a-b)** Cell population from rat neonate brain analysed by flow cytometry. Plots
with gating focusing on live cells O4- and O4+ before **(a)** and after **(b)** MACS
selection. Data shown are from 3 independent experiments (Trial 1 to 3).
**c)** Histograms of the relative expression of O4 before and after MACS
selection (left) and quantification (right), results are expressed in percentages

of O4 positive cells in both populations. Data are mean ± s.e.m. from 3
biological replicates. **d)** Expression of cell type-specific signature genes
for neural progenitor cells (NPC), oligodendrocyte progenitor cells (OPC),
oligodendrocytes (OL), astrocyte progenitor cells (APC), astrocytes (AC),
neuronal progenitors (NP) and neurons (NC) in neonate rat oligodendrocytes
cultured on hydrogels. Data are mean ± s.e.m. from 3 technical replicates.

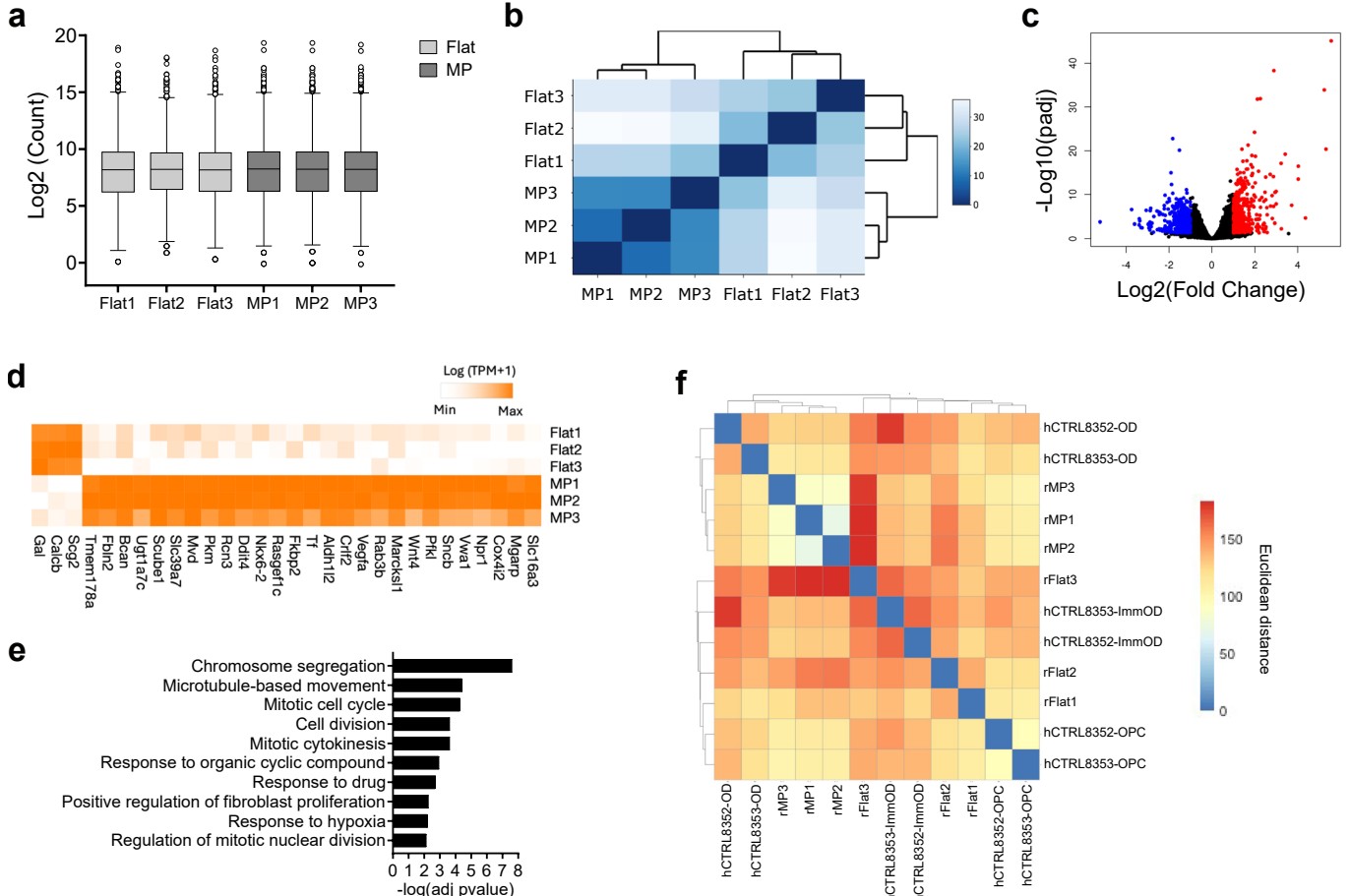

**Extended Data Fig. 3 | Comparative transcriptomic analysis of OL cultures on 2D hydrogels versus 3D micropillars (D = 5 μm and d = 10 μm). a)** Distribution of normalized reads count used to accurately determine differentially expressed genes in three technical replicates of OL cultured on flat or micropillars (MP) hydrogel. Box plots show the median (centre line), the 25th and 75th percentiles (bounds of the box), and the whiskers extend to the most extreme data points within 1.5× the interquartile range from the box. The minima and maxima shown correspond to the smallest and largest values within the whiskers. **b)** Heatmap of the overall similarity among samples assessed by the Euclidean distance between samples. **c)** Volcano plot showing transcriptional change across the flat vs micropillar culture of OLs. Each data point in the scatter plot represents a gene. Results are expressed as log2 fold change (FC) of each gene (x-axis) and the log10 of its adjusted p-value (y-axis). Genes with an adjusted p-value less than 0.05 are coloured, Log2FC > 1 (red, upregulated genes in MP) and Log2FC < 1 (blue, downregulated genes). **d)** Heatmap of top 30 differentially expressed genes

(DEG). Bi-clustering heatmap was used to visualise the expression profile of the top 30 differentially expressed genes sorted by their adjusted p-value by plotting their log2 transformed expression values in samples. **e)** Significantly DEG were clustered by their gene ontology and the enrichment of gene ontology terms was tested using Fisher exact test (GeneSCF v1.1-p2). Gene ontology terms significantly enriched with an adjusted P-value < 0.05 in the differentially expressed gene sets (up to 40 terms). **f)** Heatmap showing the Euclidean distances between samples, calculated from the regularised log transformation (rlog transformed) for the three technical replicates of bulk transcriptomic (flat1, 2, 3 and MP1, 2, 3) and SCTranformed for single cell transcriptomic dataset of OPC, immature OL (ImmOD) and mature OL (OD from the neocortex of two neurotypical individuals (CTRL8352 and 8353) (GSE218022). Statistical significance was assessed using the Wald test two-sided, and the resulting p values were adjusted for multiple testing using the Benjamini-Hochberg procedure (**c**, **e**).

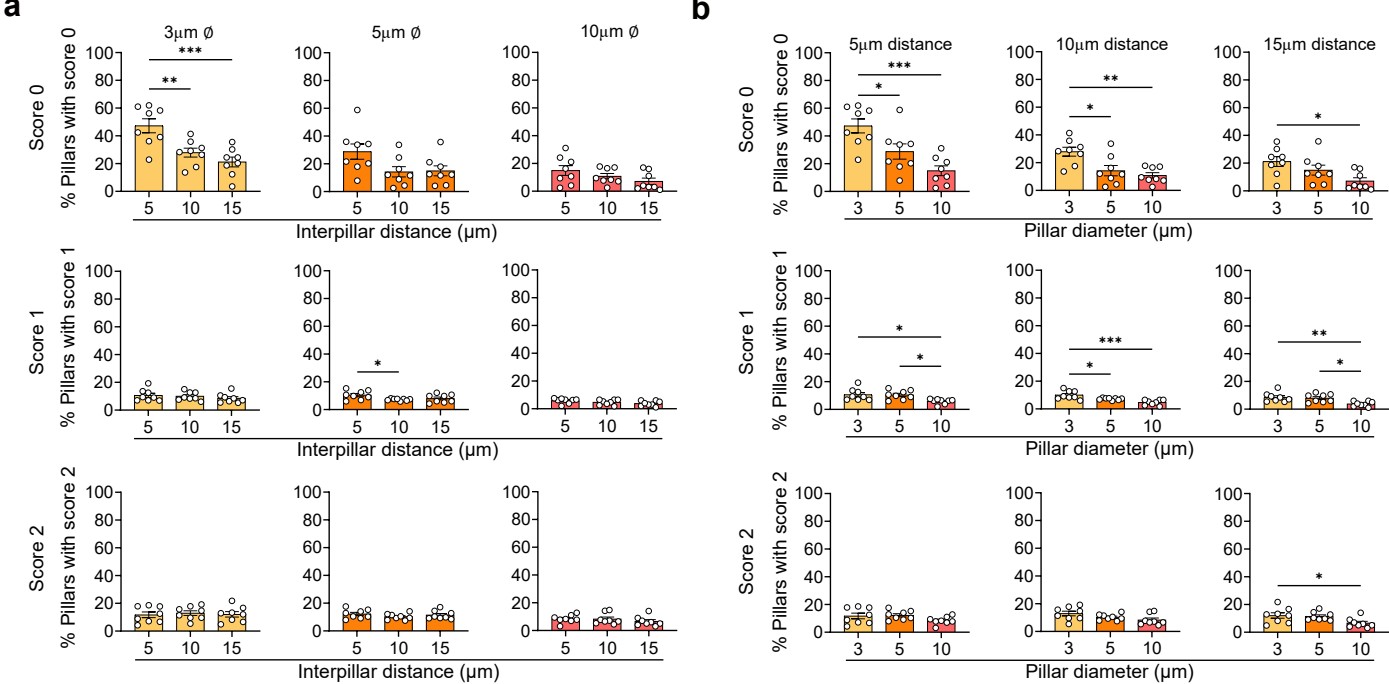

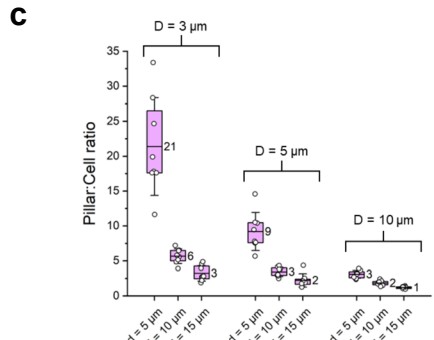

**Extended Data Fig. 4 | Impact of geometrical features on OL wrapping capability. a)** Percentages of 3 µm (left), 5 µm (middle) and 10 µm (right) diameter pillars with interdistances of 5, 10 or 15 µm with score 0 (top), 1 (middle) or 2 (bottom). **b)** Percentages of 5 µm (left), 10 µm (middle) and 15 µm (right) interpillar distances with pillar diameter of 3, 5 or 10 µm with score 0 (top), 1 (middle) or 2 (bottom). **c)** Pillar:Cell ratio for pillars with different geometries.

Box plots show the mean (centre line), the 25th and 75th percentiles (bounds of the box), and the whiskers represent s.d. Data are mean ± s.e.m. **(a, b)** or ± s.d. **(c)** with n = 8 fields from 2 replicates. Statistical significance was determined by one-way ANOVA with Tukey's multiple comparisons two-sided, *p < 0.05, **p < 0.01, and ***p < 0.001, ****p < 0.0001.

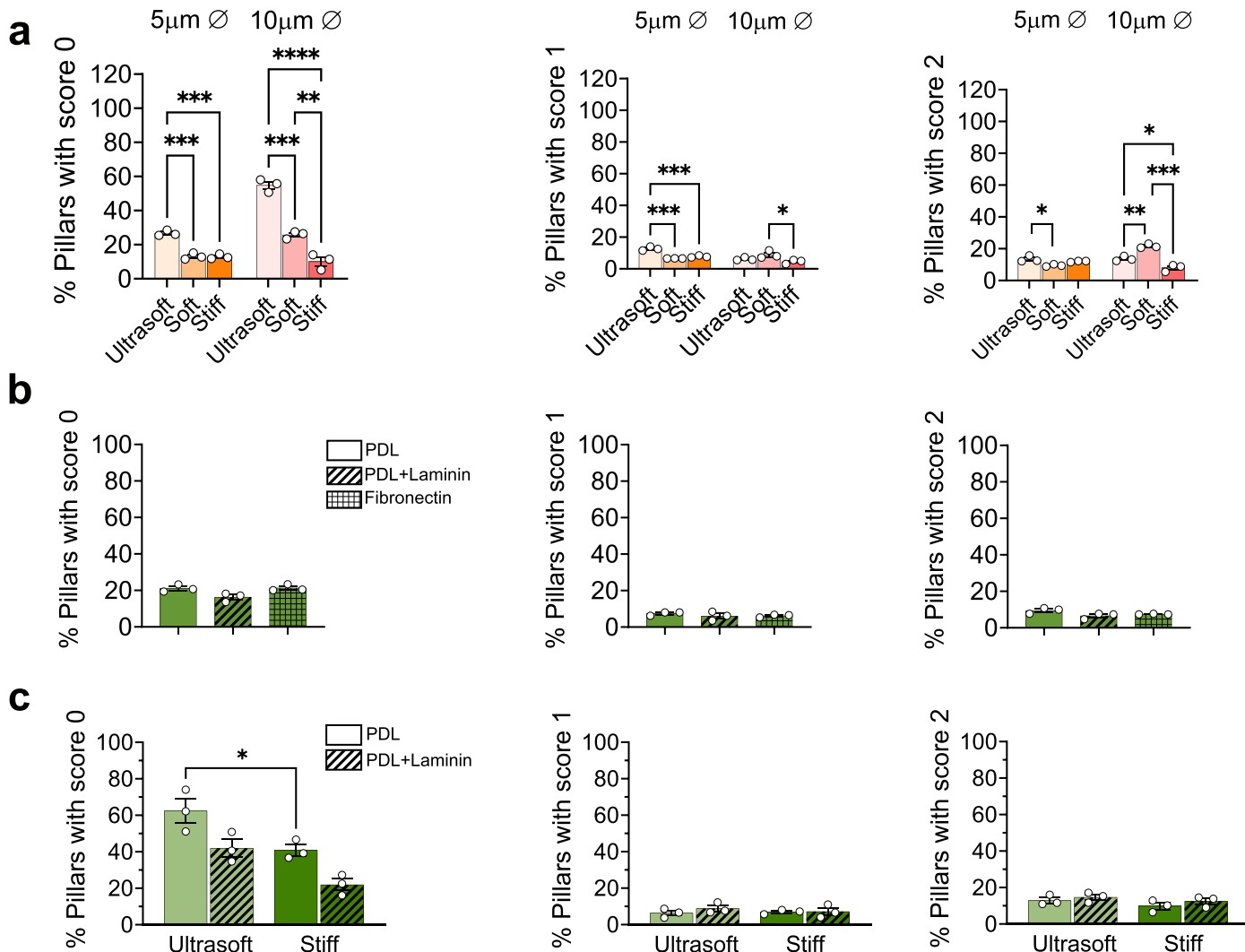

**Extended Data Fig. 5 | Impact of biochemical and biomechanical cues on OL wrapping capability. a-c)** Percentages of pillars with score 0 (left), 1 (middle) and 2 (right) for different combinations of stiffnesses (Ultrasoft-0.5 kPa; Soft-5 kPa and Stiff-50 kPa) and diameters (5 and 10 μm) **(a)**, different coatings (PDL, PDL+laminin and fibronectin) **(b)** and combinations of different stiffnesses (Ultrasoft-0.5 kPa; Stiff-50 kPa) and coatings (PDL and PDL+Laminin) **(c)**. Data are mean ± s.e.m. Each experiment was performed in at least three technical replicates from one biological sample of pooled cells. Statistical significance was determined by one-way ANOVA with Tukey's multiple comparisons two-sided, *p < 0.05, **p < 0.01, and ***p < 0.001, ****p < 0.0001.

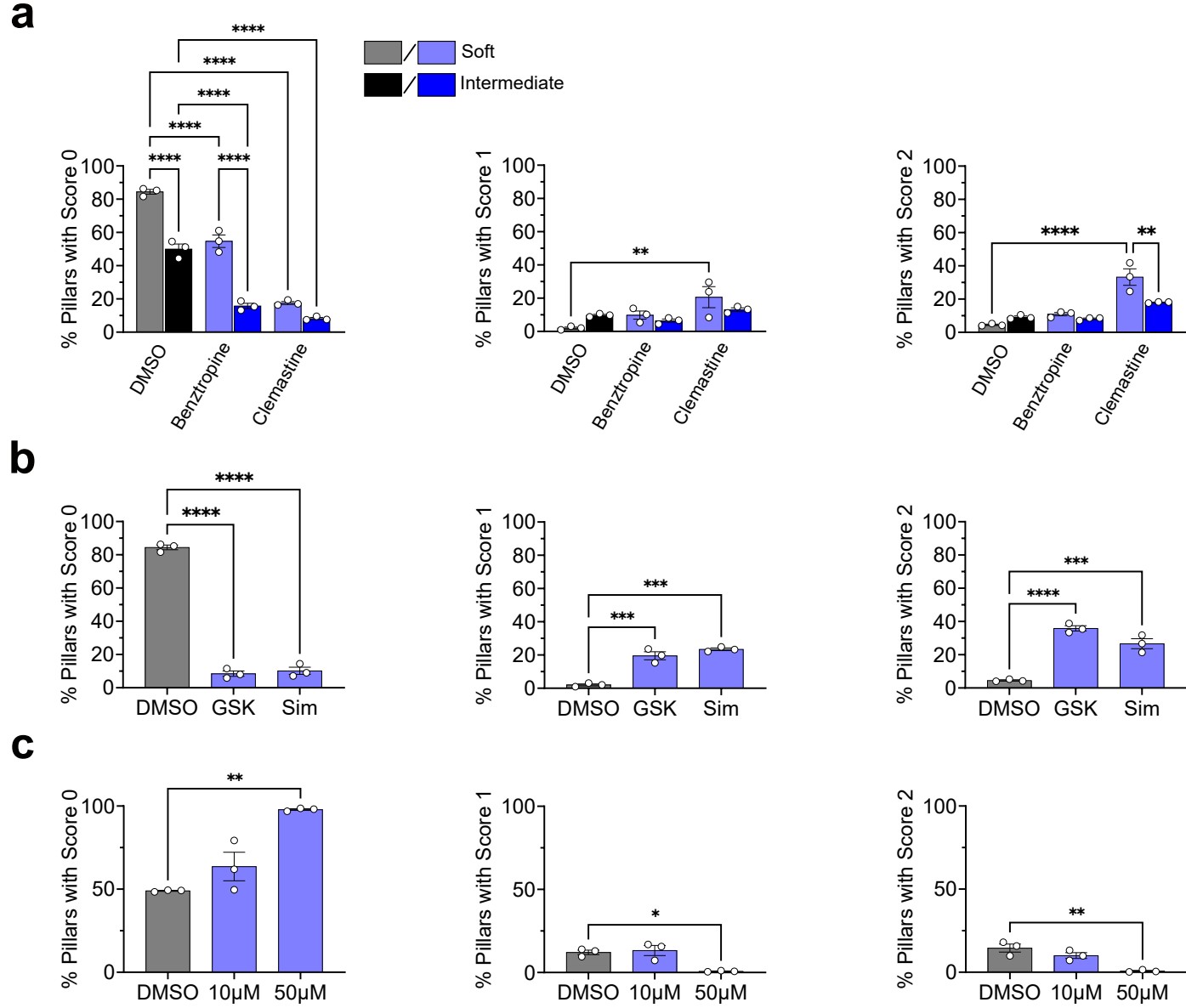

**Extended Data Fig. 6 | Impact of different compounds on OL wrapping capability. a-c)** Percentage of pillars with score 0 (left), 1 (middle) and 2 (right) for OLs treated with DMSO, benztropine (1 µM) or clemastine (10 µM) for different stiffnesses (Soft-5 kPa and intermediate-20 kPa) **(a)**, GSK239512 (10 µM), simvastatin (10 µM) **(b)** and wiskostatin (10 µM) **(c)** for Soft-5 kPa micropillars.

Data are mean ± s.e.m. Each experiment was performed in at least three technical replicates from one biological sample of pooled cells. Statistical significance was determined by one-way ANOVA with Tukey's multiple comparisons, two-sided, *p < 0.05, **p < 0.01, and ***p < 0.001, ****p < 0.0001.

## Extended Data Table 1 | Comparative analysis of existing myelination platforms to date

| Study | Material | Shape | Height/ Length | Stiffness Range | Diameter Range | Thinnest feature at softest condition | Interdistance Spacing | Human OL Compatible | Compact Myelin Formation | High-throughput Compatible | Omics-Compatible | Fabrication Technique | Length of Culture | PMID |
|---|---|---|---|---|---|---|---|---|---|---|---|---|---|---|
| **This work (Lasli et al.)** | Polyacrylamide hydrogel micropillars | Cylindric | 20-25 μm | 0.5-50 kPa | 3 to 10 μm | 5 μm at 0.5 kPa | 5 to 15 μm | Yes (foetal & hPSC-derived) | Yes (confirmed by TEM) | Potentially (array-based design) | Yes | Standard photolithography for reusable mould | 7-14 days | / |
| **Lee et al. (2012)** | Poly-L-lactic acid & polystyrene nanofibres | Cylindric | > 100 μm | > 1 GPa | 0.2–4 μm | N/A | N/A | Not tested | Weak compaction and infrequent (TEM) | Low (fibre entanglement) | Not shown | Electrospinning | 15 days | 22796663 |
| **Mei et al. (2014)** | Fused silica micropillars | Conical | 25 μm | > 1 GPa | 2-50 μm (tip to base) | 2 μm at > 1 Gpa | 50 μm | Not tested | Not shown | Yes | Not shown | Photolithography for single use | 5 days | 24997607 |
| **Bechler et al. (2015)** | Poly-L-lactic acid microfibres | Cylindric | N/A | > 1 Gpa | 0.5–4 μm | N/A | N/A | Not tested | Yes (confirmed by TEM) | Low | Not shown | Electrospinning | 7-21 days | 26320951 |
| **Espinosa-Hoyos et al. (2018)** | PDMS, poly-HEMA & poly(HDDA-co-starPEG) fibres + poly(HDDA-co-starPEG) micropillars | Cylindric | 30-200 μm | 0.4-140 kPa | 5-20 μm | 16μm at 0.4 kPa | ~ 10-30 μm | Not tested | Not shown | Yes | Not shown | Direct ink writing and projection micro-stereolithography | up to 20 days | 29323240 |
| **Ong et al. (2020)** | Polycaprolactone, polylactic acid & gelatin microfibres | Cylindric | N/A | 1 MPa to 3 GPa | 1-2 μm | N/A | Random/mesh | Not tested | Not shown | Low | Not shown | Electrospinning | 3-10 days | 32790058 |
| **Jagielska et al. (2023)** | poly(HDDA-co-starPEG) micropillars | Cylindric | 20 μm | 140 kPa | 8 μm | 8 μm at 140 kPa | ~ 10-15 μm | Not tested | Not shown | Yes | Not shown | Projection micro-stereolithography | 7 days | 37945646 |
| **Yang et al. (2025)** | poly(HDDA-co-starPEG) micropillars | Cylindric | 20 μm | 0.1-13 kPa | 3-12 μm | 3 μm at 0.1 kPa | 23-37 μm | Not tested | Not shown | Yes | Not shown | Projection micro-stereolithography | 7–14 days | 39854563 |
| **Carvalho et al. (2025)** | PDMS micropillars | Cylindric | 10 μm | ~ 0.4-1.2 MPa | 1–5 μm | 1 μm at 0.4 Mpa | 30 μm | Not tested | Not shown | Yes | Not shown | Standard photolithography for reusable mould | 1-7 days | https://doi.org/10.1101/2025.03.16.643578 |

**Extended Data Table 2 | Method of preparation for gels of varying stiffnesses**

| AAm (% w/v) | Bis-AAM (% w/v) | AA (% v/v) | Young's modulus E (kPa) |
|---|---|---|---|
| 3 | 0.06 | 0.3 | 0.5 (ultrasoft) |
| 5 | 0.15 | 0.5 | 5 (soft) |
| 8 | 0.27 | 0.8 | 20 (intermediate) |
| 12 | 0.3 | 1.2 | 50 (stiff) |

**Extended Data Table 2 | Method of preparation for gels of varying stiffnesses**

William D. Richardson
Emad Moeendarbary

# Reporting Summary

## Statistics

For all statistical analyses, confirm that the following items are present in the figure legend, table legend, main text, or Methods section.

| n/a | Confirmed | |
|---|---|---|
| ☐ | ☒ | The exact sample size (*n*) for each experimental group/condition, given as a discrete number and unit of measurement |
| ☐ | ☒ | A statement on whether measurements were taken from distinct samples or whether the same sample was measured repeatedly |
| ☐ | ☒ | The statistical test(s) used AND whether they are one- or two-sided<br>*Only common tests should be described solely by name; describe more complex techniques in the Methods section.* |
| ☒ | ☐ | A description of all covariates tested |
| ☐ | ☒ | A description of any assumptions or corrections, such as tests of normality and adjustment for multiple comparisons |
| ☐ | ☒ | A full description of the statistical parameters including central tendency (e.g. means) or other basic estimates (e.g. regression coefficient) AND variation (e.g. standard deviation) or associated estimates of uncertainty (e.g. confidence intervals) |
| ☐ | ☒ | For null hypothesis testing, the test statistic (e.g. *F*, *t*, *r*) with confidence intervals, effect sizes, degrees of freedom and *P* value noted<br>*Give P values as exact values whenever suitable.* |
| ☒ | ☐ | For Bayesian analysis, information on the choice of priors and Markov chain Monte Carlo settings |
| ☒ | ☐ | For hierarchical and complex designs, identification of the appropriate level for tests and full reporting of outcomes |
| ☒ | ☐ | Estimates of effect sizes (e.g. Cohen's *d*, Pearson's *r*), indicating how they were calculated |

*Our web collection on statistics for biologists contains articles on many of the points above.*

## Software and code

Policy information about availability of computer code

| Data collection | Confocal images were acquired using the LAS X Life software (Leica SP8) or ZEN software (Zeiss LSM 980). |
|---|---|
| Data analysis | Statistical analysis and plotting were performed in GraphPad Prism v9.0 (GraphPad Software). |

For manuscripts utilizing custom algorithms or software that are central to the research but not yet described in published literature, software must be made available to editors and reviewers. We strongly encourage code deposition in a community repository (e.g. GitHub). See the Nature Portfolio guidelines for submitting code & software for further information.

## Data

Policy information about availability of data

All manuscripts must include a data availability statement. This statement should provide the following information, where applicable:
- Accession codes, unique identifiers, or web links for publicly available datasets
- A description of any restrictions on data availability
- For clinical datasets or third party data, please ensure that the statement adheres to our policy

The authors declare that all data supporting the findings of this study are available within the article, its supplementary information files, and the public repositories as stated below. The datasets generated in this study are available as raw data in the NCBI Gene Expression Omnibus database with SuperSeries number GSE301308 which includes transcriptomic data. Publicly available single cell transcriptomic data used in the study: GSE218022 (neurotypical patients GSM6732893).

# Research involving human participants, their data, or biological material

Policy information about studies with human participants or human data. See also policy information about sex, gender (identity/presentation), and sexual orientation and race, ethnicity and racism.

| | |
|---|---|
| Reporting on sex and gender | *Use the terms sex (biological attribute) and gender (shaped by social and cultural circumstances) carefully in order to avoid confusing both terms. Indicate if findings apply to only one sex or gender; describe whether sex and gender were considered in study design; whether sex and/or gender was determined based on self-reporting or assigned and methods used.*<br>*Provide in the source data disaggregated sex and gender data, where this information has been collected, and if consent has been obtained for sharing of individual-level data; provide overall numbers in this Reporting Summary. Please state if this information has not been collected.*<br>*Report sex- and gender-based analyses where performed, justify reasons for lack of sex- and gender-based analysis.* |
| Reporting on race, ethnicity, or other socially relevant groupings | *Please specify the socially constructed or socially relevant categorization variable(s) used in your manuscript and explain why they were used. Please note that such variables should not be used as proxies for other socially constructed/relevant variables (for example, race or ethnicity should not be used as a proxy for socioeconomic status).*<br>*Provide clear definitions of the relevant terms used, how they were provided (by the participants/respondents, the researchers, or third parties), and the method(s) used to classify people into the different categories (e.g. self-report, census or administrative data, social media data, etc.)*<br>*Please provide details about how you controlled for confounding variables in your analyses.* |
| Population characteristics | *Describe the covariate-relevant population characteristics of the human research participants (e.g. age, genotypic information, past and current diagnosis and treatment categories). If you filled out the behavioural & social sciences study design questions and have nothing to add here, write "See above."* |
| Recruitment | *Describe how participants were recruited. Outline any potential self-selection bias or other biases that may be present and how these are likely to impact results.* |
| Ethics oversight | *Identify the organization(s) that approved the study protocol.* |

Note that full information on the approval of the study protocol must also be provided in the manuscript.

# Field-specific reporting

Please select the one below that is the best fit for your research. If you are not sure, read the appropriate sections before making your selection.

☒ Life sciences  ☐ Behavioural & social sciences  ☐ Ecological, evolutionary & environmental sciences

For a reference copy of the document with all sections, see nature.com/documents/nr-reporting-summary-flat.pdf

# Life sciences study design

All studies must disclose on these points even when the disclosure is negative.

| | |
|---|---|
| Sample size | Sample size was determined based on previous studies in the field. |
| Data exclusions | No data were excluded from the analysis. |
| Replication | Experiments were performed using replicates and experiments were reliably reproduced on different days using cells from a different batch of rat pups. |
| Randomization | No particular randomization strategy was implemented. |
| Blinding | Investigators were not blinded to group allocation during data collection. Data analysis was performed by users that was blinded to group allocation. |

# Reporting for specific materials, systems and methods

We require information from authors about some types of materials, experimental systems and methods used in many studies. Here, indicate whether each material, system or method listed is relevant to your study. If you are not sure if a list item applies to your research, read the appropriate section before selecting a response.

## Materials & experimental systems

| n/a | Involved in the study |
|-----|----------------------|
| ☐ | ☒ Antibodies |
| ☐ | ☒ Eukaryotic cell lines |
| ☒ | ☐ Palaeontology and archaeology |
| ☐ | ☒ Animals and other organisms |
| ☒ | ☐ Clinical data |
| ☒ | ☐ Dual use research of concern |
| ☒ | ☐ Plants |

## Methods

| n/a | Involved in the study |
|-----|----------------------|
| ☒ | ☐ ChIP-seq |
| ☐ | ☒ Flow cytometry |
| ☒ | ☐ MRI-based neuroimaging |

# Antibodies

| | |
|---|---|
| Antibodies used | Antibodies were used for fluorescent staining/imaging and are listed in the methods section along with manufacturer.<br>Primary antibody rat anti-MBP (Bio-rad; MCA409S, clone 12) at a concentration of 1:100<br>Secondary antibody goat anti-rat IgG (Thermo Fisher Scientific; A21094; 1:500)<br>Anti O4-Phycoerythrin (PE) (Miltenyi Biotec; 130-117-507; 1:100) |
| Validation | Antibody validations were performed by manufacturers. |

# Eukaryotic cell lines

Policy information about cell lines and Sex and Gender in Research

| | |
|---|---|
| Cell line source(s) | The hESC H9 cell line (WA09, purchased from WiCell Research Institute) was used to produce hPSC-derived OLs. |
| Authentication | Lines were authenticated using SNP profiling. |
| Mycoplasma contamination | All cells tested negative for mycoplasma contamination. |
| Commonly misidentified lines (See ICLAC register) | No commonly misidentified cell lines were used in this study. |

# Animals and other research organisms

Policy information about studies involving animals; ARRIVE guidelines recommended for reporting animal research, and Sex and Gender in Research

| | |
|---|---|
| Laboratory animals | Oligodendrocytes were extracted from a mixture of male and female rats at postnatal day 6 or 7 (see Methods for description of experimental procedures). |
| Wild animals | This study did not involve wild animals. |
| Reporting on sex | The sex of rats is not relevant in the context of this study. No data on sex were collected. |
| Field-collected samples | This study did not involve samples collected from the field. |
| Ethics oversight | All experiments were performed in accordance with the United Kingdom Animal (Scientific Procedures) Act 1986. |

Note that full information on the approval of the study protocol must also be provided in the manuscript.

# Plants

| | |
|---|---|
| Seed stocks | *Report on the source of all seed stocks or other plant material used. If applicable, state the seed stock centre and catalogue number. If plant specimens were collected from the field, describe the collection location, date and sampling procedures.* |
| Novel plant genotypes | *Describe the methods by which all novel plant genotypes were produced. This includes those generated by transgenic approaches, gene editing, chemical/radiation-based mutagenesis and hybridization. For transgenic lines, describe the transformation method, the number of independent lines analyzed and the generation upon which experiments were performed. For gene-edited lines, describe the editor used, the endogenous sequence targeted for editing, the targeting guide RNA sequence (if applicable) and how the editor was applied.* |
| Authentication | *Describe any authentication procedures for each seed stock used or novel genotype generated. Describe any experiments used to assess the effect of a mutation and, where applicable, how potential secondary effects (e.g. second site T-DNA insertions, mosiacism, off-target gene editing) were examined.* |

# Flow Cytometry

## Plots

Confirm that:

☒ The axis labels state the marker and fluorochrome used (e.g. CD4-FITC).

☒ The axis scales are clearly visible. Include numbers along axes only for bottom left plot of group (a 'group' is an analysis of identical markers).

☒ All plots are contour plots with outliers or pseudocolor plots.

☒ A numerical value for number of cells or percentage (with statistics) is provided.

## Methodology

| | |
|---|---|
| Sample preparation | Brain cells were extracted from a rat at postnatal day 6 or 7 (see Methods section of experimental procedures). |
| Instrument | BD Accuri C6 Flow Cytometer. |
| Software | FlowJo Software. |
| Cell population abundance | 100 thousand cells were collected per condition for each experiment. |
| Gating strategy | A first gate was applied to the side-forward scatter plots to remove debris and select the total cell population. A second gating was adjusted to exclude dead cells using side scatter-PI plots. A last gating selected PE positive cells enabling the selection of O4+ cells. Detailed gating is displayed in Extended Data Figure 2. |

☒ Tick this box to confirm that a figure exemplifying the gating strategy is provided in the Supplementary Information.

