## [Peer Review File · Nature Methods]

Tuneable hydrogel-based micropillar arrays for myelination studies

Corresponding Author: Professor Emad Moeendarbary

A version of this paper was originally rejected for publication by Nature Methods, however that decision was reconsidered after appeal by the authors.

Version 0:

Decision Letter:

22nd Nov 2023

Dear Professor Moeendarbary,

Thank you for your patience during this lengthy review process. Your Article entitled "Tuneable hydrogel-based micropillar arrays for myelination studies" has now been seen by three reviewers, whose comments are attached. In the light of their advice we have decided that we cannot offer to publish your manuscript in Nature Methods.

You will see that, while they find your work of some potential interest, the reviewers raise concerns about the advance your methodological approach represents over available methods and about the lack of experiments with human cells. We think that these criticisms are sufficiently important as to prevent publication of your work in Nature Methods.

[Redacted]

I am sorry that we at Nature Methods cannot be more positive on this occasion but hope that you find the reviewers' comments helpful when preparing your paper for submission elsewhere.

Best regards,
Nina

Nina Vogt, PhD
Senior Editor
Nature Methods

Reviewer Comments:

Reviewer #1 (Remarks to the Author):

In the manuscript entitled "Tuneable hydrogel-based micropillar arrays for myelination studies", the authors provide a novel 3D culture environment for myelination by oligodendrocytes using micropillar arrays. The platform is specifically designed to mimic the mechanical properties of soft brain tissue, allowing for the formation of compact myelin around the micropillars that closely resembles the nanostructure of myelinated axons in vivo. The key advantage of this novel platform is that it facilitates high-throughput imaging-based workflows aimed at the accurate quantification of myelin formation under different cell culture conditions. Moreover, coating with different substrates, pillar diameter, inter-pillar distance and pillar stiffness can all be easily tuned to uncouple the roles of mechanical cues and pharmacological signaling in the enhancement of myelination. This is a fascinating study with strong implications for translation. However, there are a number of concerns that have diminished my enthusiasm:

1. While I found the study fascinating and compelling, upon searching the literature, a number of studies have already demonstrated the utility of micropillar arrays for screening compounds for myelination (Mei et al., 2014; Espinosa-Hoyos et al., 2018). In fact, a recent study published in bioRxiv demonstrates that myelin wrapping and drug response by oligodendrocytes are regulated by the stiffness of these micropillars (Yang et al., doi: <https://doi.org/10.1101/2023.08.11.552940>). As such, this study is less about innovation and more about optimization and improvements to current approaches.
2. The use of different substrates is an advance in the context of drug assays. It would be useful to see more compounds tested under these conditions.
3. The EM analysis is a plus and rarely demonstrated. The findings are convincing and compelling. Unfortunately, the prior studies undermine the novelty of the approach and in the end, the manuscript seems a bit incremental for Nature Methods.

Reviewer #2 (Remarks to the Author):

This article extends previous work by making a 3D pillar system for oligodendrocyte processes to wrap around, as a pseudo-myelination assay. The improvement in this system (over previous) is that the pillars can be made different sizes and stiffnesses - and perhaps the latter may be more like real axonal stiffness (which we know can affect myelination). The drugs Benztropine and clemastine were first identified as being potentially interesting in remyelination using a similar micropillar system, and this group find they work at increasing wrapping in their new system too, using rat oligodendrocytes.

This is interesting, and adds something to the previous Mei et al., micropillar system and the microfibre cultures. I feel it is an incremental improvement on these, but I remain unconvinced (as with the previously used systems) that these are really testing myelination, and certainly are not testing what improves remyelination - which is what these authors want.

The EM of the compact myelin-like layers could do with some quantification, as a wrap of a MBP+ oligodendrocyte process around a pillar will score 3, similarly to a more compact myelin ring as seen by EM. I am not clear how many rings actually do have compact-like myelin - even roughly. This would help to convince me (or not). It is a shame that the hydrogels inevitably shrink but there do appear to be layers at least.

The spacing/size experiments are quite interesting as well, but are not going to change how we think about myelination.

The stiffness experiments are perhaps the most interesting for me, but I suppose I need to be convinced that this will improve relevant hits for screening.

The drug effect is perhaps disappointing. Neither trial in human was more than hopeful, and so a screen with drugs on these appropriately stiff pillars that produced other novel hits, perhaps other than or more than clemastine/benzotropine, would help convince me whether it is worth the effort of making these softer pillars.

These are rat cells too - if it worked with human cells, I would have much more increased enthusiasm.

So, I am not convinced that it is sufficiently novel/original/step forward for publication here.

Reviewer #3 (Remarks to the Author):

This is a very clever manuscript, presenting a method to generate a reliable platform for the study of myelination. The need for an experimental system allowing for myelination in vitro assay is undisputed. Several attempts to use nanofibers were plagued by difficulty in execution and imaging and scalability. One of the seminal papers by Mei et al. in 2014 provided an excited solution by proposing a high throughput screening using engineered micropillars, which they termed BIMA. The current study provides an advance to that system, in that they describe the construction of a mold (much simpler design in terms of execution and feasibility), with the possibility to create hydrogel-based micropillars made of acrylamide and therefore tunable to different stiffness levels and thereby allowing the investigation of biochemical and mechanical signals. Another strength of the manuscript is the assessment of the cell: pillar ratio, which allows to ask questions related to number of axons myelinated by a single oligodendrocyte.

Based on these considerations, it is believed that the current manuscript advances the fields and is impactful in providing a simpler approach to the scientific community which could enhance reproducibility and be more readily available for applications.

Major concerns which need to be thoroughly addressed include:

1. The manuscript emphasizes the importance of mechanostimulation and clearly describes that one major advantage of the hydrogel-based micropillars is the possibility to test stiffnesses that more closely resemble the axonal elastic modulus. Yet, lines 175 clearly states that for all the experiments a stiffness of 20 kPa (rather than 5 kPa) was used for all the experiments. This is problematic and it is important to show that the myelination assay is suitable for hydrogel with a stiffness of 5kPa.
2. The cell preparation used is quite confusing. O4 is an antibody which may recognize late progenitors and newly formed

oligodendrocytes. In the manuscript, line 155 describes the cells as isolated from neonatal rat brains and lines 207-208 mentions that all the cells were Ki67Negative (and thereby non proliferative) in all conditions. Yet, in figure 3b there is a clear effect on the number of cells cultured on the hydrogel pillar platform. This is quite hard to follow. A much better characterization of the cell preparation is necessary, possibly with a transcriptional profiling and purity of the preparation. In addition, the results obtained in cells isolated from neonatal brains should be compared with those obtained in cells isolated from the adult brain. 3. Throughout the manuscript it is mentioned that the current platform could be generalized to in vitro assay of drugs prior to clinical trials, yet no human cells are being tested. The use of human iPSCs in this system would be essential to prove the point and make the manuscript very exciting.

Minor concern:

The description of the cell: pillar ratio, while interesting, is highly convoluted. The authors may consider re-writing it.

** For Nature Portfolio general information and news for authors, see <http://npg.nature.com/authors>.

Version 1:

Decision Letter:

22nd Jan 2024

Dear Emad,

Thank you for your letter asking us to reconsider our decision on your Article, "Tuneable hydrogel-based micropillar arrays for myelination studies". After careful consideration we have decided that we are willing to consider a revised version of your manuscript that addresses all the technical concerns brought up by the reviewers.

Please note that you will also have to make a strong case regarding the advance of your approach over existing technology. The reviewers are all experts in the field, and we are not willing to overrule remaining concerns about novelty or advance.

- * include a point-by-point response to our referees and to any editorial suggestions
- * please underline/highlight any additions to the text or areas with other significant changes to facilitate review of the revised manuscript
- * address the points listed described below to conform to our open science requirements
- * ensure it complies with our general format requirements as set out in our guide to authors at www.nature.com/naturemethods
- * resubmit all the necessary files electronically by using the link below to access your home page

Link Redacted

We hope to receive your revised paper within two months. If you cannot send it within this time, please let us know. In this event, we will still be happy to reconsider your paper at a later date so long as nothing similar has been accepted for publication at Nature Methods or published elsewhere.

OPEN SCIENCE REQUIREMENTS

REPORTING SUMMARY AND EDITORIAL POLICY CHECKLISTS

When revising your manuscript, please submit reporting summary and editorial policy checklists.

DATA AVAILABILITY

CODE AVAILABILITY

Please include a "Code Availability" subsection in the Online Methods which details how your custom code is made available. Only in rare cases (where code is not central to the main conclusions of the paper) is the statement "available upon request" allowed (and reasons should be specified).

MATERIALS AVAILABILITY

ORCID

Nature Methods is committed to improving transparency in authorship. As part of our efforts in this direction, we are now requesting that all authors identified as 'corresponding author' on published papers create and link their Open Researcher and Contributor Identifier (ORCID) with their account on the Manuscript Tracking System (MTS), prior to acceptance. This applies to primary research papers only. ORCID helps the scientific community achieve unambiguous attribution of all scholarly contributions. You can create and link your ORCID from the home page of the MTS by clicking on 'Modify my Springer Nature account'. For more information please visit <http://www.springernature.com/orcid>.

Best regards,
Nina

Nina Vogt, PhD
Senior Editor
Nature Methods

Version 2:

Decision Letter:

Our ref: NMETH-A53668B

23rd Sep 2025

Dear Emad,

Thank you for submitting your revised manuscript "Tuneable hydrogel-based micropillar arrays for myelination studies" (NMETH-A53668B). It has now been seen by the original referees and their comments are below. The reviewers find that the paper has improved in revision, and therefore we'll be happy in principle to publish it in Nature Methods, pending minor revisions to satisfy the referees' final requests and to comply with our editorial and formatting guidelines.

TRANSPARENT PEER REVIEW

Nature Methods offers a transparent peer review option for new original research manuscripts. We encourage increased transparency in peer review by publishing the reviewer comments, author rebuttal letters and editorial decision letters if the authors agree. Such peer review material is made available as a supplementary peer review file. **Please state in the cover letter 'I wish to participate in transparent peer review' if you want to opt in, or 'I do not wish to participate in transparent peer review' if you don't.** Failure to state your preference will result in delays in accepting your manuscript for publication.

ORCID

Best regards,
Nina

Nina Vogt, PhD
Senior Editor
Nature Methods

Reviewer #1 (Remarks to the Author):

The authors should be commended for such an extensive revision. The manuscript is far improved and the results are far more compelling as an advance from previous studies.

There are only two comments that would benefit the manuscript. I believe that one key reference is missing that supports the rationale for this approach--the use of hydrogel-based structures. There is a study demonstrating that oligodendrocytes can myelinate fixed (PFA) axons. In this study, I recall that myelination is extensive--far more than on any nanofibers, especially with compact myelin. This suggests that the rigidity of the structure is important for this process. Secondly, it would benefit the table to include the Lee et al., study, as this was the first demonstration of myelination of fibers.

Other than these minor suggestions, I think the manuscript will be a great benefit to the overall community.

Reviewer #2 (Remarks to the Author):

Thank you for the revisions of this paper. I have always liked the concept and design, and now the authors have increased the robustness of their work, by including extra data, being precise with their language but also by comparing their data more with previously published tools with some similarities - the table is rather helpful.

In my view, I think this paper is now ready for publication.

Reviewer #3 (Remarks to the Author):

The authors have substantially improved both the content of the paper (by adding a number of important experiments) and the readability of the text.

I believe that the inclusion of the human data, the transcriptomic data, the more careful quantification of ultrastructural data , address previous concerns regarding novelty and I consider the manuscript as suitable for publication

Version 3:

Decision Letter:

25th Feb 2026

Dear Emad,

I am pleased to inform you that your Article, "Tuneable hydrogel-based micropillar arrays for myelination studies", has now been accepted for publication in Nature Methods. The received and accepted dates will be August 28th, 2023 and February 25th, 2026. This note is intended to let you know what to expect from us over the next month or so, and to let you know where to address any further questions.

Over the next few weeks, your paper will be copyedited to ensure that it conforms to Nature Methods style. Once your paper is typeset, you will receive an email with a link to choose the appropriate publishing options for your paper and our Author Services team will be in touch regarding any additional information that may be required. It is extremely important that you let us know now whether you will be difficult to contact over the next month. If this is the case, we ask that you send us the contact information (email, phone and fax) of someone who will be able to check the proofs and deal with any last-minute problems.

Authors may need to take specific actions to achieve compliance with funder and institutional open access mandates.

If your research is supported by a funder that requires immediate open access (e.g. according to <https://www.springernature.com/gp/open-science/plan-s-compliance> Plan S principles or the <https://www.springernature.com/gp/open-science/us-federal-agency-compliance> NIH public access policy) then you should select the gold OA route, and we will direct you to the compliant route where possible. Because authors warrant under our subscription licensing terms that they haven't committed to licensing any version of their article under a licence inconsistent with the terms of our agreement – including the applicable embargo period – publication under the subscription model isn't suitable for authors whose funders require no embargo.

Best regards,
Nina

Nina Vogt, PhD
Senior Editor
Nature Methods

** Visit the Springer Nature Editorial and Publishing website at www.springernature.com/editorial-and-publishing-jobs for more information about our career opportunities. If you have any questions please click here.**

Open Access This Peer Review File is licensed under a Creative Commons Attribution 4.0 International License, which permits use, sharing, adaptation, distribution and reproduction in any medium or format, as long as you give appropriate credit to the original author(s) and the source, provide a link to the Creative Commons license, and indicate if changes were made. In cases where reviewers are anonymous, credit should be given to 'Anonymous Referee' and the source.

We would like to sincerely thank the reviewers for their thorough and constructive evaluation of our manuscript. Their insightful feedback has substantially improved the clarity, rigour, and impact of our work. In response, we have made several significant revisions. These include the addition of new drug screening data across a range of physiologically relevant pillar stiffnesses, comprehensive ultrastructural quantification of compact myelin formation using TEM, and validation of our platform with human-derived oligodendrocytes. We have also clarified aspects of our cell preparation through flow cytometry and transcriptomic profiling and refined our discussion to emphasise the model's relevance for studying myelination rather than remyelination. These changes are detailed in our responses below and are highlighted in yellow in the revised manuscript.

Response to reviewers' comments

Reviewer #1:

In the manuscript entitled “Tuneable hydrogel-based micropillar arrays for myelination studies”, the authors provide a novel 3D culture environment for myelination by oligodendrocytes using micropillar arrays. The platform is specifically designed to mimic the mechanical properties of soft brain tissue, allowing for the formation of compact myelin around the micropillars that closely resembles the nanostructure of myelinated axons in vivo. The key advantage of this novel platform is that it facilitates high-throughput imaging-based workflows aimed at the accurate quantification of myelin formation under different cell culture conditions. Moreover, coating with different substrates, pillar diameter, inter-pillar distance and pillar stiffness can all be easily tuned to uncouple the roles of mechanical cues and pharmacological signaling in the enhancement of myelination. This is a fascinating study with strong implications for translation. However, there are a number of concerns that have diminished my enthusiasm:

R1.1) *While I found the study fascinating and compelling, upon searching the literature, a number of studies have already demonstrated the utility of micropillar arrays for screening compounds for myelination (Mei et al., 2014; Espinosa-Hoyos et al., 2018). In fact, a recent study published in bioRxiv demonstrates that myelin wrapping and drug response by oligodendrocytes are regulated by the stiffness of these micropillars (Yang et al., doi: <https://doi.org/10.1101/2023.08.11.552940>). As such, this study is less about innovation and more about optimization and improvements to current approaches.*

We thank Reviewer #1 for their positive comments on our study, particularly acknowledging its strong potential for translation from bench to bedside. We believe the improvements we have made over existing micropillar platforms constitute a significant step-change that will significantly benefit the myelin biology research community, which aligns with the high impact scope of Nature Methods. Current in vitro models for myelination require costly, complex, and time-intensive production of single-use micropillars, typically via 3D printing or electrospinning and requiring specialised manufacturing equipment. Our innovative platform democratises the 3D myelination assay space by using hydrogels that are polymerised within a reusable mould. Importantly, polyacrylamide hydrogel is an elastic substrate (the best-known hydrogel in the field of mechanobiology¹ that produces much softer pillars than previously published, leading to four significant advantages:

1) The creation of in vivo-like artificial axons that replicate the combined physical characteristics of central nervous system (CNS) axons, i.e. their geometry (diameter, height, distance), stiffness, and extracellular matrix coatings. As the reviewer correctly highlights, substrate stiffness has been shown to influence myelination. Since stiff micropillars are supra-physiologically rigid, they may enhance myelin wrapping. As discussed in the response to Reviewer #2, this could have unintended consequences such as producing more false positive results when screening for potential clinically relevant pro-myelinating drugs. Therefore, a key innovation with our technique is achieving physiologically relevant pillar stiffnesses (100-10000 Pa) that will greatly improve both our fundamental understanding of myelin formation as well as reduce costly errors in drug screening by attenuating the myelin-enhancing properties of stiff micropillars.

2) The sustainable design of our mould permits indefinite reuse, thus greatly enhancing cost-effectiveness and time-efficiency in experimental design. These important features of polyacrylamide-based micropillars will encourage their widespread adoption by the myelin biology community, which is mainly composed of neurobiologists with little or no expertise in biomaterial science or clean-room and specialised fabrication methods. Lowering the bar of entry into this field of bioengineering is important and necessary to facilitate cross-disciplinary collaboration and accelerate new advancements in myelin biology. Those focused on novel compound screening approaches will find the straightforward imaging pipeline presented here as a significant strength. Whereas researchers more interested in unravelling the mechanisms of myelin formation will appreciate the myelin-promoting properties of 3D pillars versus flat cell culture substrates which enhances the physiological relevance of downstream -omics studies in human oligodendrocytes (OLs).

3) Neither of the referenced studies demonstrated concentric, multilayered myelin around their artificial axons, which is a key criterion for validating an in vitro myelination system. Also neither of studies showed myelination of human cells. In contrast, our platform successfully recapitulated this essential feature using both rodent and human-derived OLs, highlighting a significant and novel advancement.

4) Alternative approaches, such as electrospun fibre platforms, require specialised equipment, involve time intensive procedures, and often suffer from low reproducibility. Regarding Mei et al.'s study, fibre entanglement and cell body overlap complicate OLs process quantification in confocal images². While Bechler et al. demonstrated compact myelin formation on electrospun poly-L-lactic acid fibres, their use of rigid materials and limited control over geometry highlight inherent limitations³. Ong et al. developed a tuneable-stiffness fibre platform, however, their fibres remained in the MPa-GPa range and required complex electrospinning⁴. Espinosa-Hoyos et al.⁵ and Jagielska et al.⁶ approached physiological stiffness (0.4 – 140 kPa) using 3D-printed poly(HDDA-co-starPEG) hydrogels but were limited by resolution constraints of projection micro-stereolithography, restricting pillar diameter to 16 μm for soft pillars (<1 kPa), far larger than most axons. Carvalho et al. used a mould-based fabrication to generate ultrastiff (~400 kPa) PDMS micropillars of 1-5 μm diameter without addressing the PDMS's limitations as a cell culture substrate⁷.

Please note that the manuscript has been updated to highlight the innovations of our assay in a table comparing key features of existing models (Extended Data Table 1) and the

Introduction (lines 63-113) and Discussion (lines 384-497) have been substantially revised accordingly.

Study / Platform	Material	Shape	Height	Stiffness Range	Diameter Range	Thinnest reached for softest condition (Stiffness/Diameter)	Interdistance Spacing	Human OL Compatibility	Compact Myelin Formation	High-throughput Compatible	Omics-Compatible	Fabrication Complexity	Length of Culture	PMID
This work (Lasli et al.)	Polyacrylamide hydrogel	Cylindric	20-25 μm	0.5-50 kPa	3 to 10 μm	0.5 kPa / 5 μm	5 to 15 μm	Yes (fetal & hPSC-derived)	Yes (confirmed by TEM)	Potentially (array-based design)	Yes	Low (standard photolithography for reusable mould)	7-14 days	/
Mei et al. (2014)	Fused silica micropillars	Conical	25 μm	> 1 GPa	2-50 μm (tip to base)	1 GPa / 2 μm	50 μm	Not tested	Not shown	Yes	Not shown	Moderate (photolithography for single use)	5 days	24997607
Bechler et al. (2015)	Poly-L-lactic acid nanofibers	Cylindric	N/A	> 1 GPa	0.5–4 μm	N/A	N/A	Not tested	Yes (confirmed by TEM)	Low (fiber entanglement)	Not shown	High (electrospinning)	14-21 days	26320951
Espinosa-Hoyos et al. (2018)	HDDA-PEG hydrogel	Cylindric	up to 70 μm	0.4-140 kPa	10-20 μm	0.4 kPa / 16 μm	~ 10-20 μm	Not tested	Not shown	Yes	Not shown	Very High (microstereolithography)	\geq 20 days	29323240
Ong et al. (2020)	Polycaprolactone fibers	Cylindric	N/A	0.8-13 kPa	3-15 μm	N/A	Random/mesh	Not tested	Not shown	Low	Not shown	High (electrospinning)	10 days	32790058
Jagielska et al. (2023)	HDDA-PEG hydrogel	Cylindric	20 μm	140 kPa	8 μm	140 kPa / 8 μm	~ 10-15 μm	Not tested	Not shown	Yes	Not shown	Very high (microstereolithography)	7 days	37945646
Yang et al. (2025)	HDDA-PEG hydrogel	Cylindric	20 μm	0.1-13 kPa	3-12 μm	0.1 kPa / 3 μm	20-40 μm	Not tested	Not shown	Yes	Not shown	Very high (microstereolithography)	7–14 days	39854563
Carvalho et al. (2025)	PDMS micropillars	Cylindric	10 μm	1-2 GPa	1–5 μm	1.3 GPa / 1 μm	30 μm	Not tested	Not shown	Yes	Not shown	Low (soft lithography for reusable mould)	7 days	https://doi.org/10.101/2025.03.16.643578

Extended Data Table 1: Comparative analysis of existing myelination platforms to date.

R1.2) *The use of different substrates is an advance in the context of drug assays. It would be useful to see more compounds tested under these conditions.*

We thank the reviewer for this excellent suggestion as it has added a new dimension to this paper and highlighted the importance of fabricating micropillars with physiologically relevant stiffnesses. These additional data are presented in the Figure 5 and Extended Data Figure 6 of the revised manuscript. Please also note that the Results (lines 318-344) and Discussion (lines 424- 440) sections have been updated in the revised manuscript.

Our platform demonstrated strong potential for phenotypic drug screening to assess compounds that enhance myelin formation. Using MBP-based detection of myelin wrapping, we validated the assay with two well-established pro-myelinating agents, clemastine and benztropine (Fig. 5a-c), known to promote myelination in both in vitro and in vivo models. We also included wiskostatin (Fig.5g-i), a cytoskeletal inhibitor⁸, to evaluate the platform's sensitivity to drugs modulating the mechanical aspects of myelin wrapping. Wiskostatin induced a dose-dependent decrease in myelination, confirming the assay's ability to detect subtle cytoskeleton-related effects.

Importantly, we examined the influence of substrate stiffness on drug response (Fig. 5a-c). Our data confirmed that increased stiffness alone can enhance oligodendrocyte (OL) differentiation and myelin formation⁹, potentially overshadowing or masking the effects of candidate compounds. This finding highlights a critical limitation of overly rigid in vitro systems. In contrast, our soft, axon-mimetic micropillars better replicate the mechanical environment of CNS axons and may reduce false positives, thus improving the reliability of pro-myelinating compound identification.

To further explore context-dependent drug responses, we tested GSK239512 and simvastatin (Fig. 5d-f), compounds with conflicting 2D data that ultimately failed in clinical trials. Our results confirmed previous *in vitro* findings for GSK239512's pro-myelinating effects¹⁰. In contrast, simvastatin, although associated with neuroprotective effects in MS patients, also enhanced myelination in our 3D setting, in contrast to reports from 2D culture systems where it impaired OL process formation, possibly via inhibition of Ras and Rho signaling¹¹. This divergence underscores the importance of microenvironmental context, particularly stiffness and 3D architecture, in modulating drug responses. It suggests that some compounds previously dismissed based on 2D data may exhibit beneficial effects in more physiologically relevant settings.

Collectively, our results emphasize the importance of modelling both geometry and mechanics to accurately assess therapeutic potential. The ability of our platform to distinguish genuine drug effects from stiffness-driven artifacts is critical for improving the predictive value of *in vitro* drug screening. Future work will involve testing patient-derived OLs and additional compounds with complex or ambiguous clinical trajectories, further validating the translational relevance of our assay.

Figure 5: Micropillar assay enables stiffness-dependent compound screening.

a-c) OLs on soft (5 kPa) and intermediate (20 kPa) pillars were treated with DMSO, benzotropine (1 μ M), or clemastine (10 μ M), stained for MBP as representative images (**a**), scale bar = 20 μ m, and quantified for cell number (**b**, left), percentage of fully wrapped pillars (score 3) (**b**, middle), number of wrapped pillars per cell (**b**, right), and wrapping score distribution (0–3) across conditions (**c**). **d–f)** OLs on soft (5 kPa) pillars were treated with DMSO, GSK239512 (10 μ M), or simvastatin (10 μ M), stained for MBP (**d**, scale bar = 20 μ m), and analysed as in **b** and **c** (**e,f**). **g–i)** OLs on soft (5 kPa) pillars were treated with DMSO or wiskostatin (10 or 50 μ M), stained for MBP (**g**, scale bar = 15 μ m), and quantified as in **b** and **c** (**h,i**). For all conditions, rat OLs were cultured on micropillars (5 μ m diameter, 22 μ m height, 10 μ m spacing). Data are mean \pm S.E.M. from triplicates; statistical significance by one-way ANOVA with Tukey's test, * p < 0.05, ** p < 0.01, and *** p < 0.001, **** p < 0.0001.

R1.3) *The EM analysis is a plus and rarely demonstrated. The findings are convincing and compelling. Unfortunately, the prior studies undermine the novelty of the approach and in the end, the manuscript seems a bit incremental for Nature Methods.*

We appreciate the reviewer's positive evaluation of our work, particularly highlighting the gold standard method of TEM for demonstrating the formation of compact myelin wrapping around artificial axons in vitro. We also added new TEM images from human-derived OLs and show that PSC-derived OLs also deposit compact multilayered myelin around micropillars (Fig. 6e and k). We acknowledge the concern that prior studies may appear to undermine the novelty of our approach. However, as the reviewer states above, EM analysis is rarely shown when assessing OL or Schwann cell myelin wrapping of artificial axons. Additionally, to our knowledge, the myelination and EM analysis of human OLs in vitro is a novel aspect of our work that has not been previously demonstrated.

To demonstrate the novelty of our findings, we performed in-depth TEM analysis by quantifying myelin wrapping of rat OLs. In our 3 and 5 μm hydrogel pillars, TEM images show compact myelin wraps with no extracellular gaps between tightly packed layers (Fig. 2h and i, from the original manuscript). The thickness of these wraps correlates positively with the number of layers (Fig. 2k). A high correlation supports the conclusion that myelin thickness is tightly controlled, with each additional wrap contributing to overall thickness, whereas a weak correlation would indicate irregularities in myelination, such as defective compaction or abnormal membrane spacing. This controlled growth pattern is a key feature of physiological myelination and suggests that our model closely mimics natural myelination dynamics.

In the rodent cortex, determining the exact number of myelin lamellae (wraps) around small-diameter axons (1-5 μm) is challenging, as it depends on factors such as axon location, function, calibre, and the specific position along the internode between nodes of Ranvier. Nevertheless, several studies provide quantitative values. For example, Snaidero et al. reported between as few as 2 wraps at the edges of the myelin sheath and up to 11 wraps at the thickest region near the oligodendrocyte process in the optic nerve of postnatal day 10 (P10) rats (axon diameters $<1 \mu\text{m}$). In contrast, small CNS axons in adult rat cortex typically exhibit between 7 and 9 myelin layers. In our assay, we found that fully wrapped pillars had an average of 3.5 layers, a value within the lower range of in vivo findings for small axons (Fig. 2j, right). Our TEM analysis also reveals that the average myelin sheath thickness is $45.9 \pm 5.0 \text{ nm}$ (Fig. 2j, middle), corresponding to an average thickness of $12.7 \pm 0.4 \text{ nm}$ per single lamella, consistent with findings from Basu et al. in the adult rat cortex (range: 3.9 to 17.1 nm)⁶⁸.

Additionally, 50% of hydrogel pillars exhibit multilayered myelin wrapping (at D14) (Fig. 2j, left), while MBP staining indicates that 60% of pillars are fully wrapped (at D7) (Fig. 4b). This is consistent with a potential correlation between TEM and MBP-based assessments of myelination, although a direct quantitative comparison would require further dedicated studies. This correlation would enable rapid, high-throughput screening of various conditions using immunostaining, eliminating the need for the more time-consuming EM analysis for each case. By providing a quantitative framework for myelin compaction and layer formation, our study advances in vitro myelination models beyond previous work and offers a platform for investigating myelination dynamics with high precision.

Figure 2j) Percentage of pillars with multilayer myelin per field (left), myelin thickness (middle) and number of sheaths per pillar (right) based on TEM imaging. **k)** Strong correlation ($R^2 = 0.94$) between myelin thickness and number of myelin layers. Data are mean \pm S.E.M., $n = 3$ ultrathin sections, each containing 120 pillars, and 13–16 high-resolution pillars quantified per section.

Moreover, in response to other reviewers' comments, we have significantly enhanced the impact of our work by incorporating new data demonstrating myelination and compact myelin wrapping by OLs derived from human foetal neural progenitor cells (hNPC-OPCs, Fig. 6a-e) and human pluripotent stem cells (hPSC-OPCs, Fig. 6f-k). To the best of our knowledge, no studies to date have shown myelin ensheathment and compaction around 3D micropillars or nanofibers by human OLs. We demonstrate this here using our hydrogel-based pillars to mimic the mechanical environment of the brain. We argue that our innovative assay represents the first-of-its-kind in vitro platform for studying human myelination and drug screening. This is particularly timely, as the FDA recently announced (April 10, 2025) the phasing out of animal testing requirements for monoclonal antibodies and other drugs, highlighting the importance of our in vitro model for the next generation of drug testing.

These new analysis and experiments have been added to the Results (lines 184-187 and 347-383), Material and Methods (lines 599-623) and Discussion section (lines 454-479) of the revised manuscript.

Figure 6: Validation of micropillar myelination by human oligodendrocytes.

a-e) Human foetal OLs cultured on micropillars. **a)** Schematic and timeline of foetal human OPC culture. **b)** Maximum Z intensity projection images of pillars wrapped by human foetal OLs after 9 days of culture (Cyan is nuclei stained by Hoechst, green is pillars stained by FITC and yellow is MBP). Scale bar = 20 μm . **c)** Average cell count per field (15x15 micropillars). **d)** Percentage of fully wrapped pillars (score 3) (left), number of fully wrapped pillars (score 3) per cell (middle), and percentage of pillars with wrapping scores of 0-3 (right). Data are mean \pm S.E.M. with $n = 3$ fields. **e)** TEM images of human foetal OLs wrapped around micropillars after 14 days of culture. Scale bars = 5 μm (left), 500 nm (middle) and 200 nm (right). **f-k)** hPSC-derived OLs cultured on micropillar assay. **f)** Schematic of hPSC-derived O4+ cell culture and differentiation. **g-i)** Representative images and analyses. Scale bar in **(g)** = 20 μm . **j)** SEM images of hPSC-derived OLs and micropillars after 9 days of culture. Scale bars = 10 μm (left) and 2 μm (right). **k)** TEM images of hPSC-derived OLs and micropillars after 14 days of culture. Scale bar = 0.2 μm . Human OLs were cultured on stiff micropillar arrays with $D = 5 \mu\text{m}$, $d = 6 \mu\text{m}$ and $h = 17 \mu\text{m}$. Data are mean \pm S.E.M. from triplicates.

Reviewer #2:

This article extends previous work by making a 3D pillar system for oligodendrocyte processes to wrap around, as a pseudo-myelination assay. The improvement in this system (over previous) is that the pillars can be made different sizes and stiffnesses - and perhaps the latter may be more like real axonal stiffness (which we know can affect myelination). The drugs Benztropine and clemastine were first identified as being potentially interesting in remyelination using a similar micropillar system, and this group find they work at increasing wrapping in their new system too, using rat oligodendrocytes.

This is interesting, and adds something to the previous Mei et al., micropillar system and the microfibre cultures. I feel it is an incremental improvement on these, but I remain unconvinced (as with the previously used systems) that these are really testing myelination, and certainly are not testing what improves remyelination - which is what these authors want.

We thank the reviewer for this important point and the opportunity to clarify the novelty and significance of our work. While our study builds upon prior efforts to engineer in vitro models of myelination, we believe it represents a **substantial conceptual and technical advance**.

Below, we outline seven key reasons why:

1-Technologic innovations: Alternative approaches, such as electrospun fibre platforms, require specialised equipment, involve time intensive procedures, and often suffer from low reproducibility. Fiber entanglement and cell body overlap complicate oligodendrocytes (OLs) process quantification in confocal images¹². While Bechler et al.¹³ demonstrated compact myelin formation on electrospun poly-L-lactic acid fibres, their use of rigid materials and limited control over geometry highlight inherent limitations. Ong et al.¹⁴ developed a tuneable-stiffness fibre platform, however, their fibres remained in the MPa-GPa range and required complex electrospinning. Espinosa-Hoyos et al.¹⁵ and Jagielska et al.⁶ approached physiological stiffness (0.4 – 140 kPa) using 3D-printed poly(HDDA-co-starPEG) hydrogels but were limited by resolution constraints of projection micro-stereolithography, restricting pillar diameter to 16 μm for soft pillars (<1 kPa), far larger than most axons. Carvalho et al. used a mould-based fabrication to generate ultrastiff (~400 kPa) PDMS micropillars of 1-5 μm diameter without addressing the PDMS's limitations as a cell culture substrate⁷.

Our study notably introduces two key innovations compared to existing in vitro platforms. First, we employ reusable mould fabricated via standard photolithography, allowing high-resolution control of geometry and spacing across an array, enabling simultaneous testing of diverse microenvironments. Second, we use polyacrylamide (PAA), a well-established biocompatible porous, and tuneable hydrogel, to fabricate pillars with subcellular axon-like geometry (0.5-5 μm diameter) with central nervous system (CNS)-relevant mechanical properties (~500 Pa) in a fully hydrogel-based setting. PAA also supports extra-cellular matrix (ECM) coating and optimal OL adhesion.

Our assay, with unlimited design, provides the flexibility to replicate these conditions, even on the same array, capturing key spatial and mechanical variations that influence OLs behaviour and myelination. PAA is particularly useful when studying molecular mechanisms of myelination since the pores of the micropillars could be leveraged in future assay iterations to facilitate the sustained release of growth factors or guidance molecules from the gel core. Molecules released by neuronal axons (such as neuregulin-1¹⁶), small molecule drugs, extracellular vesicles or short interfering RNAs (siRNAs) could also be adsorbed into the

micropillars and engineered to be slowly released¹⁷ into the medium and captured via cell-targeted uptake into myelin-forming OLs in order to study the effects of such molecules on myelination. The ability to decouple the role of soluble factors from axonal membrane-attached promoters of myelin wrapping is a key benefit of this in vitro system. PAA is not only mechanically tuneable but can also serve as ionically or electronically conductive materials and the platform could be implemented to study the effect of an electric signal propagation on myelination and vice versa in vitro¹⁸. Importantly, it can be functionalised, conferring an optimum culture environment for OLs culture and differentiation and coated with ECM proteins. These properties are essential for an in vitro assay to consider key complexities affecting the degree of myelination and to replicate the heterogeneity of the brains as ECM components¹⁹, stiffness²⁰ and neurons sizes and spacing are different in different regions of the CNS²¹.

Furthermore, the mechanical and chemical properties of polyacrylamide can be easily manipulated. The micropillar stiffness can range from 0.1–100 kPa and different ECM proteins (e.g. laminin) or axonal adhesion molecules (e.g. integrins) can be coated on the surface of micropillars^{15,22}. However, it is important to consider that the creation of high aspect ratio (> 1:10) ultrasoft pillars (< 1 kPa) can pose challenges as they are more susceptible to pillar collapse or damage during the demoulding process. Despite this limitation, polyacrylamide micropillars remain favourable when studying myelination, as soft micropillars (5 kPa) were shown to enhance myelin wrapping and are less prone to damage compared to ultrasoft pillars. Moreover, axons are generally the stiffest part of the neuronal cells, with stiffness of 5 kPa²³ meaning that soft pillars may better approximate axonal mechanics and thus be more physiologically relevant than ultrasoft pillars for studying myelination-related processes.

2-Biomimetic geometry and mechanics: We found that spatial constraints provided by micropillars enhance OL density compared to flat controls, regardless of pillar spacing (Fig. 2 of the original manuscript). Interestingly, on the same surface area, at similar spacing, increase diameter increases the contact surface of OLs but decrease the number of cells, when compared to flat. These findings indicate that micropillar arrays enhance OLs cell density independently of their specific geometry (within the tested range), suggesting that spatial constraints may support OLs differentiation, as previously reported²⁵, and possibly cell survival. This effect adds a new dimension to the design of culture platforms for OLs biology. Interestingly, while stiffer substrates (50 kPa) improved myelination over softer ones (5 kPa), our focus remains on replicating in vivo conditions rather than optimising artificial myelination.

3-Ultrastructural validation of compact myelin: We confirm the formation of compact myelin by TEM analysis. To underscore the novelty and physiological relevance of our model, beyond conventional MBP staining, we conducted detailed TEM analysis to assess myelin ultrastructure including g-ratio measurements, lamellae count (Fig. 2j and k). Our 3 μm and 5 μm hydrogel pillars consistently exhibited compact myelin wraps with no extracellular gaps, and wrap thickness correlated linearly with the number of layers, hallmarks of controlled, physiological myelination. Notably, the average number of wraps (~ 3.5) aligns with estimates from small-diameter axons in the rodent cortex, supporting the model's in vivo fidelity. Our TEM analysis reveals that the average myelin sheath thickness is 45.9 ± 5.0 nm (Fig. 2j, middle), corresponding to an average thickness of 12.7 ± 0.4 nm per single lamella, consistent with findings from Basu et al. in the adult rat cortex (range: 3.9 to 17.1 nm). Moreover, we observed

that 50% of pillars showed multilayered myelin by TEM at D14, closely matching MBP-based quantification (60% fully wrapped at D7), suggesting a promising correlation between ultrastructural and immunostaining-based assessments. This could enable efficient, high-throughput myelination screening using light microscopy as a surrogate for EM. These ultrastructural readouts offer definitive evidence of myelin compaction, distinguishing our system from previous platforms that often rely solely on surface immunostaining.

Figure 2j) Percentage of pillars with multilayer myelin per field (left), myelin thickness (middle) and number of sheaths per pillar (right) based on TEM imaging. **k)** Strong correlation ($R^2 = 0.94$) between myelin thickness and number of myelin layers. Data are mean \pm S.E.M., $n = 3$ ultrathin sections, each containing 120 pillars, and 13–16 high-resolution pillars quantified per section.

Please note that these new analysis have been added to the Results (lines 183-187), Material and Methods (lines 638-640) and Discussion (lines 454-470) sections of the revised manuscript.

4-Human-derived oligodendrocytes and disease modelling: A major strength of our platform lies in its compatibility with human-derived OLs, including foetal and hPSC-derived sources (Fig. 6). This represents a critical step forward in translational relevance, as most prior systems rely on rodent cells. Our platform enables functional myelination by human OLs, including evidence of multilayered myelin formation by TEM, thereby expanding the utility of this assay for disease modelling and drug screening. This is especially relevant given the growing appreciation for species-specific OL phenotypes and myelination patterns in disorders such as multiple sclerosis, leukodystrophies, and Alzheimer's disease. Additionally, the specific compatibility of hPSC-derived OLs is a major step toward modelling demyelinating-related rare genetic diseases such as Pelizaeus-Merzbacher disease, Metachromatic leukodystrophy, Krabbe disease (Globoid cell leukodystrophy), Adrenoleukodystrophy (X-ALD), Alexander disease and Canavan disease.

Figure 6: Validation of micropillar myelination by human oligodendrocytes.

a-e) Human foetal OLs cultured on micropillars. **a)** Schematic and timeline of foetal human OPC culture. **b)** Maximum Z intensity projection images of pillars wrapped by human foetal OLs after 9 days of culture (Cyan is nuclei stained by Hoechst, green is pillars stained by FITC and yellow is MBP). Scale bar = 20 μm . **c)** Average cell count per field (15x15 micropillars). **d)** Percentage of fully wrapped pillars (score 3) (left), number of fully wrapped pillars (score 3) per cell (middle), and percentage of pillars with wrapping scores of 0-3 (right). Data are mean \pm S.E.M. with $n = 3$ fields. **e)** TEM images of human foetal OLs wrapped around micropillars after 14 days of culture. Scale bars = 5 μm (left), 500 nm (middle) and 200 nm (right). **f-k)** hPSC-derived OLs cultured on micropillar assay. **f)** Schematic of hPSC-derived O4+ cell culture and differentiation. **g-i)** Representative images and analyses. Scale bar in **(g)** = 20 μm . **j)** SEM images of hPSC-derived OLs and micropillars after 9 days of culture. Scale bars = 10 μm (left) and 2 μm (right). **k)** TEM images of hPSC-derived OLs and micropillars after 14 days of culture. Scale bar = 0.2 μm . Human OLs were cultured on stiff micropillar arrays with $D = 5 \mu\text{m}$, $d = 6 \mu\text{m}$ and $h = 17 \mu\text{m}$. Data are mean \pm S.E.M. from triplicates.

Please note that these new experiments have been added to the Results (lines 346-383), Material and Methods (lines 589-623) and Discussion (lines 470-479) sections of the revised manuscript.

5-Mechanobiological and transcriptomic insight: OLs cultured on micropillar arrays exhibit transcriptional profiles aligned with human mature brain-resident OLs, while those on flat gels remain closer to oligodendrocyte progenitor cells (OPCs). Analysis of the top 30 differentially expressed genes (DEGs) (Extended Data Fig. 3d) revealed that most were significantly upregulated in OLs cultured on micropillar hydrogels. These genes were associated with key biological processes, including cellular metabolism (Pfk1, CPkm, Cox4i2, Aldh1l2, Mgarpl), ECM remodelling (Bcan, Fbln2) and neural development and differentiation (Wnt4, Vegfa, Nkx6-2). Gene ontology (GO) analysis further indicated enrichment for cell division and drug response

pathways (Extended Data Fig. 3e). Key genes associated with OL maturation and myelination were also upregulated, including *Mog*, *Mag*, *Mbp*, and *Gpr62*. Additional genes involved in cytoskeletal and lipid remodelling, critical for myelin sheath formation, such as *Ank3*, *Stbn1/4*, *Tubb3*, *Arcp2*, and *Fa2h*, were also elevated. This included increased expression of *Nfas* and decreased expression of *Lamb1* and *Lama*, which have previously been associated with OL maturation (Fig. 3b). Transcriptomic clustering further revealed that OLs cultured on micropillars aligned with human mature OL profiles, whereas those on flat hydrogels resembled OPCs and immature OLs (Extended Data Fig. 3f). Altogether, these data reinforce the link between biophysical architecture and OL differentiation and highlight our platform’s capacity to study mechanobiology in tandem with molecular maturation.

Extended Data Figure 3d) Heatmap of top 30 differentially expressed genes (DEG). Bi-clustering heatmap was used to visualise the expression profile of the top 30 differentially expressed genes sorted by their adjusted p-value by plotting their log₂ transformed expression values in samples. **e)** Significantly DEG were clustered by their gene ontology and the enrichment of gene ontology terms significantly enriched with an adjusted P-value < 0.05 in the differentially expressed gene sets (up to 40 terms). **f)** Heatmap showing the Euclidean distances between samples, calculated from the regularised log transformation (rlog transformed) for the three technical replicates of bulk transcriptomic (flat1, 2, 3 and MP1, 2, 3) and SCTransformed for single cell transcriptomic dataset of OPC, immature OL (ImmOD) and mature OL (OD from the neocortex of two neurotypical individuals (CTRL8352 and 8353) (GSE218022)¹.

Figure 3b) Heatmap shows differential expression (FC) and significance (pvalue) of particularly expressed in OPC, mature OL and involved in cytoskeleton and lipids dynamism and adhesion protein of the ECM in OLs cultures on micropillars compared to flat hydrogels.

Please note that these new experiments have been added to the Results (lines 190-216), Material and Methods (lines 693-725) and Discussion (lines 442-452) sections and are displayed in Figure 3, Extended Data Figure 3 and Extended Data Tables 2-5 of the revised manuscript.

6-Stiffness and drug screening implications: Our platform demonstrated strong potential for phenotypic drug screening to assess compounds that enhance myelin formation. Using MBP-based detection of myelin wrapping, we validated the assay with two well-established pro-myelinating agents, clemastine and benztropine (Fig. 5a-c), known to promote myelination in both in vitro and in vivo models. We also included wiskostatin, a cytoskeletal inhibitor⁸, to evaluate the platform's sensitivity to drugs modulating the mechanical aspects of myelin wrapping. Wiskostatin induced a dose-dependent decrease in myelination, confirming the assay's ability to detect subtle cytoskeleton-related effects (Fig. 5g-i).

Importantly, we examined the influence of substrate stiffness on drug response. Our data confirmed that increased stiffness alone can enhance oligodendrocyte (OL) differentiation and myelin formation⁹, potentially overshadowing or masking the effects of candidate compounds (Fig. 5a-c). This finding highlights a critical limitation of overly rigid in vitro systems. In contrast, our soft, axon-mimetic micropillars better replicate the mechanical environment of CNS axons and may reduce false positives, thus improving the reliability of pro-myelinating compound identification.

To further explore context-dependent drug responses, we tested GSK239512 and simvastatin (Fig. 5d-f), compounds with conflicting 2D data that ultimately failed in clinical trials. Our results confirmed previous in vitro findings for GSK239512's pro-myelinating effects¹⁰. Simvastatin, although associated with neuroprotective effects in MS patients, also enhanced myelination in our 3D setting, in contrast to reports from 2D culture systems where it impaired OL process formation, possibly via inhibition of Ras and Rho signaling¹¹. This divergence underscores the importance of microenvironmental context, particularly stiffness and 3D architecture, in modulating drug responses. It suggests that some compounds previously dismissed based on 2D data may exhibit beneficial effects in more physiologically relevant settings.

Collectively, our results emphasize the importance of modelling both geometry and mechanics to accurately assess therapeutic potential. The ability of our platform to distinguish genuine drug effects from stiffness-driven artifacts is critical for improving the predictive value of in vitro drug screening. Future work will involve testing patient-derived OLs and additional compounds with complex or ambiguous clinical trajectories, further validating the translational relevance of our assay.

Figure 5: Micropillar assay enables stiffness-dependent compound screening.

a-c) OLs on soft (5 kPa) and intermediate (20 kPa) pillars were treated with DMSO, benztropine (1 μ M), or clemastine (10 μ M), stained for MBP as representative images (**a**), scale bar = 20 μ m, and quantified for cell number (**b**, left), percentage of fully wrapped pillars (score 3) (**b**, middle), number of wrapped pillars per cell (**b**, right), and wrapping score distribution (0–3) across conditions (**c**). **d–f)** OLs on soft (5 kPa) pillars were treated with DMSO, GSK239512 (10 μ M), or simvastatin (10 μ M), stained for MBP (**d**, scale bar = 20 μ m), and analysed as in **b** and **c** (**e,f**). **g–i)** OLs on soft (5 kPa) pillars were treated with DMSO or wiskostatin (10 or 50 μ M), stained for MBP (**g**, scale bar = 15 μ m), and quantified as in **b** and **c** (**h,i**). For all conditions, rat OLs were cultured on micropillars (5 μ m diameter, 22 μ m height, 10 μ m spacing). Data are mean \pm S.E.M. from triplicates; statistical significance by one-way ANOVA with Tukey’s test, * $p < 0.05$, ** $p < 0.01$, and *** $p < 0.001$, **** $p < 0.0001$.

Please note that these new experiments have been added to the Results (lines 320-344) and Discussion (lines 429-440) sections and are displayed in Figure 5 and Extended Data Figure 6 of the revised manuscript.

7-Scalability, accessibility, and future directions: By leveraging standard microfabrication tools and commercially available hydrogels, our system is reproducible, cost-effective, and widely deployable. Unlike labour-intensive fibre spinning or complex 3D printing, our method is accessible to both neuroscience and bioengineering labs. The system is reusable, supports high-throughput formats, and can be adapted for future integration of signalling gradients or electrical cues.

To emphasize the innovation of our model, we have implemented the manuscript with a table comparing key features of existing models to ours (Extended Data Table 1).

Study / Platform	Material	Shape	Height	Stiffness Range	Diameter Range	Thinnest reached for softest condition (Stiffness/Diameter)	Interdistance Spacing	Human OL Compatibility	Compact Myelin Formation	High-throughput Compatible	Omics-Compatible	Fabrication Complexity	Length of Culture	PMD
This work (Lasli et al.)	Polyacrylamide hydrogel	Cylindric	20-25 μm	0.5-50 kPa	3 to 10 μm	0.5 kPa / 5 μm	5 to 15 μm	Yes (fetal & hPSC-derived)	Yes (confirmed by TEM)	Potentially (array-based design)	Yes	Low (standard photolithography for reusable mould)	7-14 days	/
Mei et al. (2014)	Fused silica micropillars	Conical	25 μm	> 1 GPa	2-50 μm (tip to base)	1 GPa / 2 μm	50 μm	Not tested	Not shown	Yes	Not shown	Moderate (photolithography for single use)	5 days	24997607
Bechler et al. (2015)	Poly-L-lactic acid nanofibers	Cylindric	N/A	> 1 GPa	0.5-4 μm	N/A	N/A	Not tested	Yes (confirmed by TEM)	Low (fiber entanglement)	Not shown	High (electrospinning)	14-21 days	26320951
Espinosa-Hoyos et al. (2018)	HDDA-PEG hydrogel	Cylindric	up to 70 μm	0.4-140 kPa	10-20 μm	0.4 kPa / 16 μm	~ 10-20 μm	Not tested	Not shown	Yes	Not shown	Very High (microstereolithography)	\geq 20 days	29323240
Ong et al. (2020)	Polycaprolactone fibers	Cylindric	N/A	0.8-13 kPa	3-15 μm	N/A	Random/mesh	Not tested	Not shown	Low	Not shown	High (electrospinning)	10 days	32790058
Jagielska et al. (2023)	HDDA-PEG hydrogel	Cylindric	20 μm	140 kPa	8 μm	140 kPa / 8 μm	~ 10-15 μm	Not tested	Not shown	Yes	Not shown	Very high (microstereolithography)	7 days	37945646
Yang et al. (2025)	HDDA-PEG hydrogel	Cylindric	20 μm	0.1-13 kPa	3-12 μm	0.1 kPa / 3 μm	20-40 μm	Not tested	Not shown	Yes	Not shown	Very high (microstereolithography)	7-14 days	39854563
Carvalho et al. (2025)	PDMS micropillars	Cylindric	10 μm	1-2 GPa	1-5 μm	1.3 GPa / 1 μm	30 μm	Not tested	Not shown	Yes	Not shown	Low (soft lithography for reusable mould)	7 days	https://doi.org/10.1101/2025.03.16.643578

Extended Data Table 1: Comparative analysis of existing myelination platforms to date.

However, we understand the reviewer’s concern about the study of “remyelination” in our assay and have updated the manuscript to focus the discussion on myelination processes rather than “remyelination”.

We acknowledge that in vitro models that attempt to mimic the physical characteristics of axons possess inherent limitations; a trait all complex 3D models share. The primary challenge, as in previous studies, is in creating artificial axons that fully replicate in vivo characteristics. Achieving this could accelerate both our fundamental understanding of the myelination process as well as the pharmacological actions of myelin-repair agents. Therefore, it is crucially important to improve the in vivo-like characteristics of currently available in vitro models of myelination so as to replicate the complex microenvironment of the central nervous system. While in vivo validation of new discoveries from in vitro assays remains essential, biomimetic in vitro assays are key to unravelling and examining the mechanistic roles of various parameters, as well as in testing and manipulating different pathways both individually and collectively. We emphasise that our platform uniquely captures the in vivo mechanical properties of axons and their microenvironment more faithfully than previous assay attempts and demonstrate, for the first time, and that micropillars promote myelin sheath formation by human OLs (Fig. 6). This represents a considerable step forward in basic and translational applications.

The EM of the compact myelin-like layers could do with some quantification, as a wrap of a MBP+ oligodendrocyte process around a pillar will score 3, similarly to a more compact myelin ring as seen by EM. I am not clear how many rings actually do have compact-like myelin - even roughly. This would help to convince me (or not). It is a shame that the hydrogels inevitably shrink but there do appear to be layers at least.

We appreciate Reviewer #2’s suggestion. As mentioned in our response to Reviewer #1 and as suggested here, we performed quantification of TEM images and incorporated the results

in the updated manuscript. We found that compact myelin wraps are comprised of 3-4 layers of myelin, on average, (Fig. 2j, right) and show high correlation with myelin thickness (Fig. 2k), as previously described *in vivo*. Our TEM analysis reveals that the average myelin sheath thickness is 45.9 ± 5.0 nm (Fig. 2j, middle), corresponding to an average thickness of 12.7 ± 0.4 nm per single lamella, consistent with findings from Basu et al. in the adult rat cortex (range: 3.9 to 17.1 nm). Moreover, we found that 50% of the hydrogel pillars exhibit multilayered myelin wrapping by D14 (Fig. 2j, left), while MBP immunostaining indicates that 60% of pillars are fully wrapped as early as D7. These findings suggest a potential correlation between TEM and MBP-based assessments of myelination, although a direct, in-depth, quantitative comparison would require further systematic studies. Establishing such a correlation, for the first time, would facilitate rapid, high-throughput screening of different conditions via immunostaining, reducing the reliance on the more time- and labour-intensive EM analysis.

Figure 2j Percentage of pillars with multilayer myelin per field (left), myelin thickness (middle) and number of sheaths per pillar (right) based on TEM imaging. **k** Strong correlation ($R^2 = 0.94$) between myelin thickness and number of myelin layers. Data are mean \pm S.E.M., $n = 3$ ultrathin sections, each containing 120 pillars, and 13–16 high-resolution pillars quantified per section.

The manuscript has been revised accordingly in the discussion (lines 454-479) and the results section as follows:

“Our analysis showed that over 50% of pillars displayed multilayered myelin ($51.9 \pm 15.12\%$) with average thickness of 45.9 ± 20.7 nm and average number of myelin sheath of 3.58 ± 1.6 (Fig.2j). The degree of myelin compaction was consistent across different pillars indicated by the high correlation ($R^2=0.94$) between the number of myelin sheaths and the thickness of myelin (Fig.2k).”

The spacing/size experiments are quite interesting as well, but are not going to change how we think about myelination.

Our study represents a substantial advance beyond previous myelination models by integrating biomimetic geometry, physiological mechanics, and human relevance in a scalable platform. Unlike prior work, we achieve sub-3 μm hydrogel pillars with tuneable, CNS-relevant stiffness (~ 500 Pa), enabling compact myelin formation validated by ultrastructural TEM, including g-ratio analysis and lamellae quantification. We further demonstrate functional myelination using human hPSC-derived OLs (Fig.6), a key step toward clinical translation. Importantly, transcriptomic profiling reveals that our 3D pillar system enhances OL maturation compared to flat substrates, offering molecular insight into mechanobiological regulation (Fig. 3 and Extended Data Figure 2d-f). Designed for accessibility and reuse, our platform supports high-throughput, physiologically realistic drug screening, addressing key limitations of prior systems and establishing a qualitatively new tool for studying myelination.

Our new data demonstrate the effect of stiffness on myelination depends on pillar geometry (Fig. 4a-c). Specifically, for 5 μm diameter pillars, OLs show no difference in myelination between soft and stiff conditions, whereas ultrasoft conditions reduce the number of fully wrapped pillars. In contrast, for 10 μm diameter pillars, myelination increases with stiffness (Fig. 4a-c). This highlights that stiffness influences myelination in a geometry-dependant manner.

Figure 4 a-c) Impact of pillar stiffness and diameter on OL differentiation and myelination. Data are mean \pm S.E.M. Each experiment was performed in triplicates. Statistical significance was determined by one-way ANOVA with Tukey's multiple comparisons post hoc tests, * $p < 0.05$, ** $p < 0.01$, and *** $p < 0.001$, **** $p < 0.0001$.

As a methodology-focused study, we demonstrate (1) the high sensitivity of our assay, capable of detecting differences in myelination response to geometry and biomechanical changes and (2) the importance of a highly versatile and tuneable platform, allowing the precise modelling of myelination dynamics and disease conditions.

Another important finding from the spacing/size experiments is that OL wrapping exhibited a trend (Fig. 3c-f), and further analysis revealed that pillar diameter (3, 5, and 10 μm) significantly influenced the extent of wrapping (Fig. 3f, right). This suggests that, in vitro, OLs intrinsically favour wrapping thicker pillars, in line with in vivo observations where larger axon calibres are typically associated with increased myelination.

Additionally, the measure of the g-ratio showed that larger pillars (10 μm) produced higher g-ratios, consistent with CNS observations⁴⁰, whereas the g-ratio observed in 5 μm pillars most closely matched the physiological range reported in rodents (0.72 to 0.81⁴¹).

Figure 3f) Percentage of pillars with corresponding wrapping score (0-3) (left) and score 3 only (middle and right) across different interpillar distances (middle) and pillar diameters (right). **g)** G-ratio analysis of myelin on pillars of 3, 5, and 10 μm diameter (5 μm spacing). Scale bar = 5 μm. Data are mean ± S.E.M., n = 8 fields of view for two replicates; statistics by one-way ANOVA with Tukey's test, *p < 0.05, **p < 0.01, and ***p < 0.001, ****p < 0.0001.

Please note, the following addition to the Results section:

“Independent of the interpillar distance, the proportion of pillars scoring 3 increased as the diameter of pillars increased (Fig. 3f, right) recapitulating a key feature of in vivo myelination where thicker axons tend to have higher level of myelination⁴⁰. More interestingly, the estimated g-ratio for different pillar diameters (0.67 ± 0.03 , 0.79 ± 0.01 and 0.88 ± 0.006 for 3, 5 and 10 μm pillars respectively) was in the range found in CNS ($0.72-0.81$)⁴¹ and increased with pillar diameter (Fig. 3g).”

Please also note the following addition to the Material and Methods section:

“For g-ratio analysis, myelin thickness was measured with ImageJ using this formula: $G = D/(D+2t)$ with D = pillar diameter and t = myelin thickness.”

The stiffness experiments are perhaps the most interesting for me, but I suppose I need to be convinced that this will improve relevant hits for screening. The drug effect is perhaps disappointing. Neither trial in human was more than hopeful, and so a screen with drugs on these appropriately stiff pillars that produced other novel hits, perhaps other than or more than clemastine/benzotropine, would help convince me whether it is worth the effort of making these softer pillars.

Our platform demonstrated strong potential for phenotypic drug screening to assess compounds that enhance myelin formation (Fig. 5). Using MBP-based detection of myelin wrapping, we validated the assay with two well-established pro-myelinating agents, clemastine and benztropine (Fig. 5a-c), known to promote myelination in both in vitro and in vivo models. We also included wiskostatin, a cytoskeletal inhibitor⁸, to evaluate the platform's sensitivity to drugs modulating the mechanical aspects of myelin wrapping (Fig. 5g-i).

Wiskostatin induced a dose-dependent decrease in myelination, confirming the assay's ability to detect subtle cytoskeleton-related effects.

Importantly, we examined the influence of substrate stiffness on drug response. Our data confirmed that increased stiffness alone can enhance oligodendrocyte (OL) differentiation and myelin formation⁹, potentially overshadowing or masking the effects of candidate compounds (Fig. 5a-c). This finding highlights a critical limitation of overly rigid in vitro systems. In contrast, our soft, axon-mimetic micropillars better replicate the mechanical environment of CNS axons and may reduce false positives, thus improving the reliability of pro-myelinating compound identification.

To further explore context-dependent drug responses, we tested GSK239512 and simvastatin (Fig.5d-f), compounds with conflicting 2D data that ultimately failed in clinical trials. Our results confirmed previous in vitro findings for GSK239512's pro-myelinating effects¹⁰. In contrast, simvastatin, although associated with neuroprotective effects in MS patients, also enhanced myelination in our 3D setting, in contrast to reports from 2D culture systems where it impaired OL process formation, possibly via inhibition of Ras and Rho signaling¹¹. This divergence underscores the importance of microenvironmental context, particularly stiffness and 3D architecture, in modulating drug responses. It suggests that some compounds previously dismissed based on 2D data may exhibit beneficial effects in more physiologically relevant settings.

Collectively, our results emphasize the importance of modelling both geometry and mechanics to accurately assess therapeutic potential. The ability of our platform to distinguish genuine drug effects from stiffness-driven artifacts is critical for improving the predictive value of in vitro drug screening. Future work will involve testing patient-derived OLs and additional compounds with complex or ambiguous clinical trajectories, further validating the translational relevance of our assay.

Please note that these new experiments have been added to the Results (lines 320-344) and Discussion (lines 429-440) sections and are displayed in Figure 5 and Extended Data Figure 6 of the revised manuscript.

Reviewer #3:

This is a very clever manuscript, presenting a method to generate a reliable platform for the study of myelination. The need for an experimental system allowing for myelination in vitro assay is undisputed. Several attempts to use nanofibers were plagued by difficulty in execution and imaging and scalability. One of the seminal papers by Mei et al. in 2014 provided an excited solution by proposing a high throughput screening using engineered micropillars, which they termed BIMA. The current study provides an advance to that system, in that they describe the construction of a mold (much simpler design in terms of execution and feasibility), with the possibility to create hydrogel-based micropillars made of acrylamide and therefore tunable to different stiffness levels and thereby allowing the investigation of biochemical and mechanical signals. Another strength of the manuscript is the assessment of the cell: pillar ratio, which allows to ask questions related to number of axons myelinated by a single oligodendrocyte.

Based on these considerations, it is believed that the current manuscript advances the fields and is impactful in providing a simpler approach to the scientific community which could enhance reproducibility and be more readily available for applications.

Major concerns which need to be thoroughly addressed include:

1. The manuscript emphasizes the importance of mechanostimulation and clearly describes that one major advantage of the hydrogel-based micropillars is the possibility to test stiffnesses that more closely resemble the axonal elastic modulus. Yet, lines 175 clearly states that for all the experiments a stiffness of 20 kPa (rather than 5 kPa) was used for all the experiments. This is problematic and it is important to show that the myelination assay is suitable for hydrogel with a stiffness of 5kPa.

We thank Reviewer #3 for this remark. We have repeated most of the experiments with pillars of 5 kPa, including all drug testing, as presented in this document in Figure 5.

We acknowledge that it was an essential point as we examined the influence of substrate stiffness on drug response. Our data confirmed that increased stiffness alone can enhance oligodendrocyte (OL) differentiation and myelin formation⁹, potentially overshadowing or masking the effects of candidate compounds. This finding highlights a critical limitation of overly rigid in vitro systems. In contrast, our soft, axon-mimetic micropillars better replicate the mechanical environment of CNS axons and may reduce false positives, thus improving the reliability of pro-myelinating compound identification.

Figure 5: Micropillar assay enables stiffness-dependent compound screening.

a-c) OLS on soft (5 kPa) and intermediate (20 kPa) pillars were treated with DMSO, benztropine (1 μM), or clemastine (10 μM), stained for MBP as representative images (**a**), scale bar = 20 μm, and quantified for cell number (**b**, left), percentage of fully wrapped pillars (score 3) (**b**, middle), number of wrapped pillars per cell (**b**, right), and wrapping score distribution (0–3) across conditions (**c**). **d–f)** OLS on soft (5 kPa) pillars were treated with DMSO, GSK239512 (10 μM), or simvastatin (10 μM), stained for MBP (**d**, scale bar = 20 μm), and analysed as in **b** and **c** (**e,f**). **g–i)** OLS on soft (5 kPa) pillars were treated with DMSO or wiskostatin (10 or 50 μM), stained for MBP (**g**, scale bar = 15 μm), and quantified as in **b** and **c** (**h,i**). For all conditions, rat OLS were cultured on micropillars (5 μm diameter, 22 μm height, 10 μm spacing). Data are mean ± S.E.M. from triplicates; statistical significance by one-way ANOVA with Tukey’s test, *p < 0.05, **p < 0.01, and ***p < 0.001, ****p < 0.0001.

Please note that these new experiments have been added to the Results (lines 320-344) and Discussion (lines 429-440) sections and are displayed in Figure 5 and Extended Data Figure 6 of the revised manuscript.

2. The cell preparation used is quite confusing. O4 is an antibody which may recognize late progenitors and newly formed. In the manuscript, line 155 describes the cells as isolated from neonatal rat brains and lines 207-208 mentions that all the cells were Ki67Negative (and thereby non proliferative) in all conditions. Yet, in figure 3b there is a clear effect on the number of cells cultured on the hydrogel pillar platform. This is quite hard to follow. A much better characterization of the cell preparation is necessary, possibly with a transcriptional profiling and purity of the preparation. In addition, the results obtained in cells isolated from neonatal brains should be compared with those obtained in cells isolated from the adult brain.

We understand the potential for confusion based on how we phrased these results in the original manuscript. It may not have been explicitly stated but the objective of the Ki67 staining was to confirm that we had effectively differentiated the oligodendrocyte progenitor cells (OPCs) to oligodendrocytes (OLs) and that the cells seeded onto the micropillar platforms were non-proliferative OLs. To avoid confusion, we have removed any mention of the O4+ cells being Ki67-negative in the revised manuscript.

The reason for the increase in the number of cells on the micropillar surfaces compared to the flat gels was almost certainly due to an increase in the surface area for cell attachment in the z-plane, i.e. OLs had a 3D environment to attach to and grow on the micropillar structures. Similar to the densely packed 3D cytoarchitecture of the brain parenchyma, OLs grown on micropillar structures survived better in this clustered 3D environment compared to a flat gel surface.

Using flow cytometry, we show that 60% of the cells are O4+ following the magnetic bead selection protocol (Extended Data Fig.2a-c). As suggested, we have also conducted transcriptomic analysis of the OLs cultured for 5 days post-differentiation on both flat and micropillar hydrogel conditions. Our results show that the OL population exhibits high expression of oligodendrocyte-specific gene signatures, with very low expression of genes associated with OPCs or other brain cell types which are 80 times less expressed (Extended Data Fig. 2d).

Extended Data Figure 2: OPC purification and OL population characterisation.

a-b) Cell population from rat neonate brain analysed by flow cytometry. Detailed of the gating focusing on live cells O4- and O4+ before **(a)** and after **(b)** MACS selection. Data shown are from 3 independent experiments (Trial 1 to 3). **c)** Histograms of the relative expression of O4 before and after MACS selection (left) and quantification (right), results are expressed in percentages of O4 positive cells in both populations. **d)** Expression of cell type-specific signature genes for neural progenitor cells (NPC), oligodendrocyte progenitor cells (OPC), oligodendrocytes (OL), astrocyte progenitor cells (APC), astrocytes (AC), neuronal progenitors (NP) and neurons (NC) in neonate rat oligodendrocytes cultured on hydrogels. Data from flow cytometry are mean \pm S.E.M. from 3 independent experiments.

Please note, the following addition in the Material and Methods' section:

“Flow cytometry

After the immunomagnetic isolation of cells, 2 different fractions (each of 100,000 cells) were analysed. The original fraction, collected prior to the isolation procedure, contained diverse

brain cell types, whereas the positive fraction comprised only those retained within the column following MACS. Cells were incubated with anti O4-Phycoerythrin (PE) (Miltenyi Biotec; 130-117-507) for 30 min in the fridge and washed. To measure cell viability, 1 μ l of 100 μ g/ml of propidium iodide (PI) (Sigma; P4864) was added to the cell suspension and incubated for 1-2 min before analysis with a BD Accuri™ C6 Flow Cytometer. FlowJo software was used to analyse the data.”

Additionally, to characterise deeper our population of OLs obtained after culture in our platform, we performed a comparative analysis of transcriptomic data from OLs differentiated in pillars and flat hydrogel. Gene Set Enrichment Analysis (GSEA) of all significantly upregulated genes (Fig. 3a, left and Extended Data Table 3) confirmed enrichment of signalling pathways related to ECM remodelling (ECM, collagen-containing ECM, cell junction organisation) and glial lineage progression (neurogenesis, cell differentiation). Conversely, GSEA of downregulated genes (Fig. 3a, right and Extended Data Table 4) showed enrichment for pathways associated with cell division (kinetochore, mitotic spindle, cell cycle) and high cellular dynamism (ATP binding/hydrolysis activity, microtubule/cytoskeletal motor activity). Analysis of the top 30 DEGs (Extended Data Fig. 3d and Extended Data Table 2) revealed that most were significantly upregulated in OLs cultured on micropillar hydrogels. These genes were associated with key biological processes, including cellular metabolism (Pfkf1, CPkm, Cox4i2, Aldh1l2, Mgarp), ECM remodelling (Bcan, Fbln2) and neural development and differentiation (Wnt4, Vegfa, Nkx6-2). Gene ontology (GO) analysis further indicated enrichment for cell division and drug response pathways (Extended Data Fig. 3e and Extended Data Table 5). Key genes associated with OL maturation and myelination were also upregulated, including Mog, Mag, Mbp, and Gpr62. Additional genes involved in cytoskeletal and lipid remodelling, critical for myelin sheath formation, such as Ank3, Stbn1/4, Tubb3, Arpc2, and Fa2h, were also elevated. This included increased expression of Nfas and decreased expression of Lamb1 and Lama, which have previously been associated with OL maturation (Fig. 3b and Extended Data Table 2). Transcriptomic clustering further revealed that OLs cultured on micropillars aligned with human mature OL profiles, whereas those on flat hydrogels resembled OPCs and immature OLs (Extended Data Fig. 3f).

Figure 3a-b) Rat OLs were cultured on 50kPa micropillars (MP) with an interpillar distance of 10 μ m and 5 μ m diameter or on 50 kPa flat 2D hydrogels. Bioinformatic analysis of RNAseq data was performed and differentially expressed genes (DEGs) were clustered by their gene ontology (GO) and enrichment scores tested using Fisher’s exact test. Significantly upregulated GO terms, with an adjusted P-value <0.05 and Log2FC>1, are presented in **(a, left)** and downregulated (Log2FC<1) GO terms presented in **(b, right)**.

Extended Data Fig. 3f) Heatmap showing the Euclidean distances between samples, calculated from the regularised log transformation (rlog transformed) for the three technical replicates of bulk transcriptomic (flat1, 2, 3 and MP1, 2, 3) and SCTransformed for single cell transcriptomic dataset of OPC, immature OL (ImmOD) and mature OL (OD from the neocortex) of two neurotypical individuals (CTRL8352 and 8353) (GSE218022) ²⁶.

Please note that the manuscript has been revised accordingly in the Results (lines 190-216), Material and Methods (lines 693-725) and Discussion sections (lines 442-452) and results are displayed in Figure 3, Extended Data Figure 3 and Extended Data Tables 2-5 of the revised manuscript.

3. Throughout the manuscript it is mentioned that the current platform could be generalized to *in vitro* assay of drugs prior to clinical trials, yet no human cells are being tested. The use of human iPSCs in this system would be essential to prove the point and make the manuscript very exciting.

We agree, and since first submission we have indeed seeded OLs derived from foetal human neural progenitor cells (hNPC-OLs) and human-induced pluripotent stem cells (hPSC-OLs) onto our platform (Fig. 6). Our results confirm the successful differentiation of human OLs into mature myelinating OLs as shown by confocal imaging of MBP expression for both types and multilayered compact myelin wraps for PSCs (Fig. 6).

These additional data are presented in the figure 6 of the revised manuscript and highlighted in yellow in the text.

Figure 6: Validation of micropillar myelination by human oligodendrocytes.

a-e) Human foetal OLs cultured on micropillars. **a)** Schematic and timeline of foetal human OPC culture. **b)** Maximum Z intensity projection images of pillars wrapped by human foetal OLs after 9 days of culture (Cyan is nuclei stained by Hoechst, green is pillars stained by FITC and yellow is MBP). Scale bar = 20 μm . **c)** Average cell count per field (15x15 micropillars). **d)** Percentage of fully wrapped pillars (score 3) (left), number of fully wrapped pillars (score 3) per cell (middle), and percentage of pillars with wrapping scores of 0-3 (right). Data are mean \pm S.E.M. with $n = 3$ fields. **e)** TEM images of human foetal OLs wrapped around micropillars after 14 days of culture. Scale bars = 5 μm (left), 500 nm (middle) and 200 nm (right). **f-k)** hPSC-derived OLs cultured on micropillar assay. **f)** Schematic of hPSC-derived O4+ cell culture and differentiation. **g-i)** Representative images and analyses. Scale bar in **(g)** = 20 μm . **j)** SEM images of hPSC-derived OLs and micropillars after 9 days of culture. Scale bars = 10 μm (left) and 2 μm (right). **k)** TEM images of hPSC-derived OLs and micropillars after 14 days of culture. Scale bar = 0.2 μm . Human OLs were cultured on stiff micropillar arrays with $D = 5 \mu\text{m}$, $d = 6 \mu\text{m}$ and $h = 17 \mu\text{m}$. Data are mean \pm S.E.M. from triplicates.

Please note that these new experiments have been added to the Results (lines 346-383), Material and Methods (lines 589-623) and Discussion (lines 470-479) sections of the revised manuscript.

References:

1. Ciccone, G. & Salmeron-Sanchez, M. Tuning the matrix: Recent advances in mechanobiology unveiled through polyacrylamide hydrogels. *Curr Opin Biomed Eng* **35**, 100604 (2025).
2. Mei, L. & Xiong, W. C. Neuregulin 1 in neural development, synaptic plasticity and schizophrenia. *Nature Reviews Neuroscience* *2008 9:6* **9**, 437–452 (2008).
3. Sobottka, B., Ziegler, U., Kaech, A., Becher, B. & Goebels, N. CNS live imaging reveals a new mechanism of myelination: The liquid croissant model. *Glia* **59**, 1841–1849 (2011).
4. Ong, W. *et al.* Biomimicking Fiber Platform with Tunable Stiffness to Study Mechanotransduction Reveals Stiffness Enhances Oligodendrocyte Differentiation but Impedes Myelination through YAP-Dependent Regulation. *Small* **16**, (2020).
5. Espinosa-Hoyos, D. *et al.* Engineered 3D-printed artificial axons. *Sci Rep* **8**, (2018).
6. Jagielska, A. *et al.* Artificial axons as a biomimetic 3D myelination platform for the discovery and validation of promyelinating compounds. *Scientific Reports* *2023 13:1* **13**, 1–13 (2023).
7. Carvalho, E. D. *et al.* Engineered micropillars to unveil oligodendrocyte responses to physical cues. *bioRxiv* 2025.03.16.643578 (2025) doi:10.1101/2025.03.16.643578.
8. Bacon, C., Lakics, V., Machesky, L. & Rumsby, M. N-WASP regulates extension of filopodia and processes by oligodendrocyte progenitors, oligodendrocytes, and Schwann cells - Implications for axon ensheathment at myelination. *Glia* **55**, 844–858 (2007).
9. Yang, M., Martin, C. J. L., Kowsari, K., Jagielska, A. & Vliet, K. J. Van. Myelin ensheathment and drug responses of oligodendrocytes are modulated by stiffness of artificial axons. *PLoS One* **20**, e0290521 (2025).
10. Chen, Y. *et al.* Histamine Receptor 3 negatively regulates oligodendrocyte differentiation and remyelination. *PLoS One* **12**, e0189380 (2017).
11. Miron, V. E. *et al.* Statin Therapy Inhibits Remyelination in the Central Nervous System. *Am J Pathol* **174**, 1880–1890 (2009).
12. Mei, F. *et al.* Micropillar arrays as a high-throughput screening platform for therapeutics in multiple sclerosis. *Nat Med* **20**, 954–960 (2014).
13. Bechler, M. E., Byrne, L. & Ffrench-Constant, C. CNS Myelin Sheath Lengths Are an Intrinsic Property of Oligodendrocytes. *Current Biology* **25**, 2411–2416 (2015).
14. Ong, W. *et al.* Biomimicking Fiber Platform with Tunable Stiffness to Study Mechanotransduction Reveals Stiffness Enhances Oligodendrocyte Differentiation but Impedes Myelination through YAP-Dependent Regulation. *Small* **16**, (2020).
15. Espinosa-Hoyos, D. *et al.* Engineered 3D-printed artificial axons. *Sci Rep* **8**, (2018).
16. Mei, L. & Xiong, W. C. Neuregulin 1 in neural development, synaptic plasticity and schizophrenia. *Nature Reviews Neuroscience* *2008 9:6* **9**, 437–452 (2008).
17. Nguyen, M. K. *et al.* Covalently tethering siRNA to hydrogels for localized, controlled release and gene silencing. *Sci Adv* **5**, 801–829 (2019).
18. Park, K., Choi, H., Kang, K., Shin, M. & Son, D. Soft Stretchable Conductive Carboxymethylcellulose Hydrogels for Wearable Sensors. *Gels* **8**, 92 (2022).
19. Dauth, S. *et al.* Extracellular matrix protein expression is brain region dependent. *Journal of Comparative Neurology* **524**, 1309–1336 (2016).

20. Linka, K. *et al.* Unraveling the Local Relation Between Tissue Composition and Human Brain Mechanics Through Machine Learning. *Front Bioeng Biotechnol* **9**, 704738 (2021).
21. Silbereis, J. C., Pochareddy, S., Zhu, Y., Li, M. & Sestan, N. The Cellular and Molecular Landscapes of the Developing Human Central Nervous System. *Neuron* **89**, 248–268 (2016).
22. Kang, M. & Yao, Y. Laminin regulates oligodendrocyte development and myelination. *Glia* **70**, 414–429 (2022).
23. Zhang, Y. *et al.* Modeling of the axon membrane skeleton structure and implications for its mechanical properties. *PLoS Comput Biol* **13**, (2017).
24. Zhang, Y. *et al.* Modeling of the axon membrane skeleton structure and implications for its mechanical properties. *PLoS Comput Biol* **13**, (2017).
25. Rosenberg, S. S., Kelland, E. E., Tokar, E., De La Torre, A. R. & Chan, J. R. *The Geometric and Spatial Constraints of the Microenvironment Induce Oligodendrocyte Differentiation*. www.pnas.org/cgi/content/full/ (2008).
26. Chung, C. *et al.* Comprehensive multi-omic profiling of somatic mutations in malformations of cortical development. *Nature Genetics* **2023** 55:2 **55**, 209–220 (2023).

Response to reviewers' comments

Reviewer #1:

The authors should be commended for such an extensive revision. The manuscript is far improved and the results are far more compelling as an advance from previous studies. There are only two comments that would benefit the manuscript:

1. I believe that one key reference is missing that supports the rationale for this approach--the use of hydrogel-based structures. There is a study demonstrating that oligodendrocytes can myelinate fixed (PFA) axons. In this study, I recall that myelination is extensive--far more than on any nanofibers, especially with compact myelin. This suggests that the rigidity of the structure is important for this process.

We thank Reviewer #1 for this remark. This study (Rosenberg et al.) was already present in the manuscript, but we added this specific point in the Introduction section.

“Mechanobiological cues from neurons and the extracellular matrix (ECM) are key regulators of myelination, as OL mechanosensitivity to stiffness influences process extension and wrapping^{6,7}. Importantly, OLs can also produce compact myelin on paraformaldehyde (PFA)-fixed axons, suggesting that myelination can occur independent of axonal signals when the mechanical environment is permissive⁸. In parallel, neuronal activity can direct myelination by modulating OL selection of axons⁹.”

2. Secondly, it would benefit the table to include the Lee et al., study, as this was the first demonstration of myelination of fibers.

We thank Reviewer #1 for this important point. We added Lee et al. study in the Extended Data table 1 as shown below.

Study	Material	Shape	Height/Length	Stiffness Range	Diameter Range	Thinnest feature at softest condition	Interdistance Spacing	Human OL Compatible	Compact Myelin Formation	High-throughput Compatible	Omics-Compatible	Fabrication Technique	Length of Culture	PMID
This work (Lasli et al.)	Polyacrylamide hydrogel micropillars	Cylindric	20-25 μm	0.5-50 kPa	3 to 30 μm	5 μm at 0.5 kPa	5 to 15 μm	Yes (foetal & iPSC-derived)	Yes (confirmed by TEM)	Potentially (array-based design)	Yes	Standard photolithography for reusable mould	7-14 days	/
Lee et al. (2012)	Poly-L-lactic acid & polystyrene nanofibres	Cylindric	> 100 μm	> 1 GPa	0.2-4 μm	N/A	N/A	Not tested	Weak compaction and infrequent (TEM)	Low (fibre entanglement)	Not shown	Electrospinning	15 days	22796663

Reviewer #2:

Thank you for the revisions of this paper. I have always liked the concept and design, and now the authors have increased the robustness of their work, by including extra data, being precise with their language but also by comparing their data more with previously published tools with some similarities - the table is rather helpful.

In my view, I think this paper is now ready for publication.

We thank Reviewer #2 for the positive outcome.

Reviewer #3:

The authors have substantially improved both the content of the paper (by adding a number of important experiments) and the readability of the text.

I believe that the inclusion of the human data, the transcriptomic data, the more careful quantification of ultrastructural data, address previous concerns regarding novelty and I consider the manuscript as suitable for publication.

We thank Reviewer #3 for the positive outcome.